

# A field theory representation of sum of powers of principal minors and physical applications

**Morteza Nattagh Najafi[1], Abolfazl Ramezanpour[2,3] and Mohammad Ali Rajabpour[4]**

**1** Department of Physics, University of Mohaghegh Ardabili, P.O. Box 179, Ardabil, Iran
**2** Department of Physics, College of Science, Shiraz University, Shiraz 71454, Iran
**3** Medical Systems Biophysics and Bioengineering, Leiden Academic Centre
for Drug Research, Faculty of Science, Leiden University, Leiden, The Netherlands
**4** Instituto de Física, Universidade Federal Fluminense, Av. Gal. Milton Tavares de Souza,
s/n, Campus da Praia Vermelha São Domingos, 24210-346 Niterói - RJ, Brasil

⋆ morteza.nattagh@gmail.com , † aramezanpour@gmail.com ,
‡ mohammadali.rajabpour@gmail.com

## Abstract

We introduce a novel field theory representation for the Sum of Powers of Principal Minors (SPPM), a mathematical construct with profound implications in quantum mechanics and statistical physics. We begin by establishing a Berezin integral formulation of the SPPM problem, showcasing its versatility through various symmetries including $SU(n)$, its subgroups, and particle-hole symmetry. This representation not only facilitates new analytical approaches but also offers deeper insights into the symmetries of complex quantum systems. For instance, it enables the representation of the Hubbard model's partition function in terms of the SPPM problem. We further develop three mean field techniques to approximate SPPM, each providing unique perspectives and utilities: the first method focuses on the evolution of symmetries post-mean field approximation, the second, based on the bosonic representation, enhances our understanding of the stability of mean field results, and the third employs a variational approach to establish a lower bound for SPPM. These methods converge to identical consistency relations and values for SPPM, illustrating their robustness. The practical applications of our theoretical advancements are demonstrated through two compelling case studies. First, we exactly solve the SPPM problem for the Laplacian matrix of a chain, a symmetric tridiagonal matrix, allowing for precise benchmarking of mean-field theory results. Second, we present the first analytical calculation of the Shannon-Rényi entropy for the transverse field Ising chain, revealing critical insights into phase transitions and symmetry breaking in the ferromagnetic phase. This work not only bridges theoretical gaps in understanding principal minors within quantum systems but also sets the stage for future explorations in more complex quantum and statistical physics models.

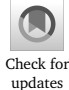

## Contents

# 1 Introduction

Matrices find widespread applications in a variety of scientific fields, such as quantum mechanics and machine learning. In quantum mechanics, matrices serve as a representation of physical observables and transformations in the Hilbert space, offering a potent mathematical framework to describe and comprehend the behavior of quantum systems. In machine learning, matrices are commonly employed to represent datasets, linear models, and for matrix factorization methods that facilitate feature extraction and dimensionality reduction. While many properties of matrices, including determinant, trace, rank, eigenvalues, and eigenvectors can be calculated in polynomial time, other quantities such as permanent [1], the sum of powers of principal minors(SPPM) [2] and mixed discriminants [2], require exponential time complexity making them relatively difficult to compute. The sum of powers of principal minors has been found to be useful in a variety of interesting applications. For example, in theoretical physics, it appears in the calculation of the Shannon-Rényi entropy of fermionic systems and quantum spin chains [3]. In machine learning, the sum of powers of principal minors is used in determinental point processes [4]. Additionally, the sum of powers of principal minors plays a role in mixed discriminants [2], which are objects used in convex geometry to study mixed volumes [5]. Additionally, we demonstrate in this paper that the partition function of the renowned Hubbard model can be expressed as a SPPM problem.

So far, exact numerical calculations have been the only viable approach for computing the Shannon-Rényi entropy of quantum chains, including the transverse field Ising (TFI) chain and free fermions. Calculations of this kind are limited by the curse of exponential growth, resulting in the largest matrix sizes being limited to around 42 [6, 7]. Moreover, there is a growing interest in the machine learning community to efficiently compute the sum of powers of principal minors of symmetric matrices. With this in consideration, for the first time we introduce a method that is analytically controllable and can be used to compute this quantity for a broad spectrum of matrices perturbatively.

The primary focus of this paper is the introduction of a Berezin integral representation for SPPM problem with integer powers $n$. This representation bears significant importance in multiple aspects. It resembles a field theory representation and exhibits a range of intriguing symmetries including $SU(n)$ symmetry, its subgroups like $U(1)$ and axial $U(1)$, symmetric group symmetry, and particle-hole symmetry. Furthermore, this representation facilitates the creation of new representations, such as the bosonic integral representation, which can be utilized for additional analytical computations.

Following the introduction of various representations of SPPM involving Grassmann or scalar variables, we introduce three mean field methods for approximating the sum. The initial approach in mean field methodology involves simplifying the interaction component of the action within the Berezin-Grassmann framework by substituting it with quadratic terms and then establishing the corresponding consistency relations. The use of the Berezin integral representation sheds light on the examination of symmetries and their behavior post-mean field, particularly regarding spontaneous symmetry breaking. The second method which is based on the bosonic representation and the steepest-descent approximation is more appropriate for verifying the stability of mean field outcomes, rendering it a more viable option for higher-order perturbation theory. The third mean field technique employs a variational method, where Jensen's inequality is utilized to derive a lower bound for the sum of powers of principal minors. All three mean field approaches converge to identical consistency relations and yield the same value for the SPPM. We conduct a thorough analysis of all techniques and classify the majority of intriguing scenarios. Within the Berezin integral representation, we contend that near the critical point, intriguing scaling relationships can emerge, akin to those observed in critical phenomena.

We will examine two compelling examples in our study. The initial example is a symmetric tridiagonal matrix known as the Laplacian matrix of a chain. All the principal minors of this matrix are non-negative. This system is interpreted as the weighted partition function of rooted spanning forests. A rooted spanning forest is a type of subgraph within the original graph, encompassing all its vertices and forming a forest. Each tree within this forest has a specifically designated root vertex at which it is rooted. We solve the SPPM problem for any value of the power, which can be interpreted as the inverse temperature. We determine the free energy, average energy, entropy, and the relevant principal minors exactly. Since we can solve this problem exactly, we can use it to compare and evaluate the mean-field theory results in a controlled manner.

Our second example involves calculating the Shannon-Rényi entropy of the TFI chain. The corresponding matrix in this case is a dense antisymmetric matrix, and an exact solution may not be possible. However, we have managed to calculate this quantity analytically at the mean field level for the first time. Our analytical results demonstrate the phase transition at $h = 1$ and reveal both the universal and non-universal contributions to the Rényi entropy, which were previously obtained solely through numerical methods. Additionally, our Berezin integral representation provides a fresh perspective on the scaling relations around the critical point of the TFI chain. Furthermore, we demonstrate that some symmetries are spontaneously broken in the ferromagnetic phase. We should emphasize that the Shannon-Rényi entropy *differs* from the von Neumann-Rényi entanglement entropy, which can be efficiently computed here using the correlation matrix method, see [8,9]. Simply put, the Shannon-Rényi entropy concerns the diagonal elements of the density matrix, while the von Neumann Rényi entropy focuses on the eigenvalues of the density matrix. From an experimental perspective, Shannon-Rényi entropy is a more practical quantity because it only requires projective measurements in a local basis, like the spins at different sites. This is a standard procedure with today's technology.

The structure of the paper is as follows: The upcoming section initially defines the main problem and then highlights several significant physical problems that are analogous to the SPPM problem. Specifically, it demonstrates how the Hubbard model's partition function can be related to the SPPM. Section 3 outlines the paper's key findings. Section 4 begins with a field theory formulation of the SPPM problem and then introduces various new ways to represent the SPPM. It notably discusses the potential application of Sourlas duality for approximating the sum. Section 5 categorizes the symmetries in the field theory representation for both general and specific (symmetric and antisymmetric) matrices. Section 6 focuses on the mean-field approximation of the SPPM, presenting three different mean-field methods. The first method

is particularly useful in understanding the symmetries of the problem, the second aids in grasping the stability of mean-field results, and the third offers a lower bound for the SPPM. These methods are analyzed in both original and dual spaces. Section 7 provides an exact solution for the SPPM problem for arbitrary powers in the Laplacian of a chain, which is then used to validate the mean-field theory results. Section 8 is dedicated to examining the Shannon-Rényi entropy of the ground state of the TFI chain, including the first analytical calculation of this entropy using the mean-field techniques. It especially highlights how transitioning to the dual space yields highly accurate results in the ferromagnetic domain. The paper concludes in section 9 with discussions and prospects for future research. Numerous appendices accompany the paper, detailing the calculations of the results discussed in the main text.

## 2 Definitions and motivations

In this section we first introduce the problem of sum of the powers of the principal minors (SPPM) of a matrix, and then discuss the main motivations and applications. Consider $\mathbf{A} = \left(a_{i,j}\right)_{i,j=1}^{l}$ as an $l \times l$ square matrix. First we define $[l] = \{1, 2, ..., l\}$, and the sets $I, J \subseteq [l]$. Then the sets $I^c$ and $J^c$ can be defined as the complements of $I$ and $J$ in $[l]$. We then define the matrix $\mathbf{A}_{I,J}$ as a submatrix of $\mathbf{A}$ corresponding to the rows $I$ and the columns $J$, all kept in their original order. From now on we also use the notation $\mathbf{A}_I \equiv \mathbf{A}_{II}$. The quantity of interest is $M^{(n)}(\mathbf{A})$ defined as the summation of $n$th power of all the principal minors (SPPM) of $\mathbf{A}$, i.e.

$$M^{(n)}(\mathbf{A}) \equiv \sum_I \left[\det \mathbf{A}_I\right]^n , \qquad (1)$$

where $\sum_I$ represents the summation over all possible subsets $I$ (sometimes called *all configurations* in this paper). In this work we mostly consider integer $n$s. The number of principal minors is $2^l$. The SPPM for $n = 1$ is simply related to the characteristic polynomial. In the rest of this section we discuss briefly a few problems that can be mapped to the SPPM problem.

### 2.1 Rényi entropy of eigenstates of quadratic fermionic systems

Consider a free fermionic system described by a quadratic Hamiltonian

$$H_{\text{free}} = \mathbf{c}^\dagger \mathbf{M}\mathbf{c} + \frac{1}{2}\mathbf{c}^\dagger \mathbf{N}\mathbf{c}^\dagger + \frac{1}{2}\mathbf{c}\mathbf{N}^T\mathbf{c} - \frac{1}{2}\text{Tr}\mathbf{M}, \qquad (2)$$

where $\mathbf{c} \equiv (c_1, c_2, ..., c_L)$ is a vector made of the fermionic annihilation operators and $\mathbf{c}^\dagger$ is defined similarly by the creation operators. The $\mathbf{M}$ and $\mathbf{N}$ are in general Hermitian and antisymmetric matrices, respectively. In this paper however we work with real Hamiltonians. The correlation matrix $\mathbf{G}$ for an eigenstate is defined using two Jordan fermionic operators $\gamma_j \equiv c_j^\dagger + c_j$, and $\bar{\gamma}_j \equiv i\left(c_j^\dagger - c_j\right)$ as follows:

$$iG_{jk} = \text{Tr}\left[\rho_l \bar{\gamma}_j \gamma_k\right], \qquad (3)$$

where $\rho_l$ is the (reduced) density matrix corresponding to the (sub)system. Formation probabilities are defined as the probability of finding a particular configuration of fermions in the (sub)system. These probabilities for the subsystem can be calculated as follows: one first writes the reduced density matrix of the eigenstate (usually the ground state) in fermionic coherent basis as follows [9]:

$$\begin{aligned}
\rho_l(\xi, \xi') &= <\xi|\rho_l|\xi'> \\
&= \det\left[\frac{1}{2}(\mathbf{I} - \mathbf{G})e^{\frac{1}{2}(\bar{\xi}-\xi')^T \mathbf{F}(\bar{\xi}+\xi')}\right],
\end{aligned} \qquad (4)$$

where we defined $c_j|\xi>= \xi_j|\xi>$, $\mathbf{I}$ is the identity matrix and

$$\mathbf{F} \equiv (\mathbf{I} + \mathbf{G})(\mathbf{I} - \mathbf{G})^{-1}. \tag{5}$$

Then one can calculate the formation probability of an arbitrary configuration $C$ by putting $\xi_j = 0$ when there is no fermion at site $j$ and integrate over remaining $\xi$ variables. This procedure leads to the following general formula for formation probabilities [3]:

$$P(C) = \det\left[\frac{1}{2}(\mathbf{I} - \mathbf{G})\right] \det \mathbf{F}_C, \tag{6}$$

where $\mathbf{F}_C$ is the submatrix of $\mathbf{F}$ corresponding to the configuration $C$. It is the submatrix of $\mathbf{F}$ with all rows and columns in $J \equiv \left\{j : c_j^\dagger |C\rangle = 0\right\}$ remained intact. Note that the summation over all principal minors of $F$ is $\det(\mathbf{I} + \mathbf{F})$, guaranteeing that $\sum_C P(C) = 1$. Although we found the above relation for a subsystem with mixed state it can be also applied for the full system as far as the $\mathbf{F}$ matrix exists.

The Shannon-Rényi entropy of the (sub)system is defined as

$$R_n(l) \equiv \frac{1}{1-n} \ln \sum_C P(C)^n, \tag{7}$$

where $n$ is the Rényi index. Note that in the limit $n \to 1$ we recover the well-known Shannon entropy, i.e. $R_1(l) = -\sum_C P(C) \ln P(C)$. Using Eq. 6, one can express the Shannon-Rényi entropy of the fermionic system in terms of SPPM as:

$$R_n(l) = \frac{n}{1-n} \ln \det\left[\frac{1}{2}(\mathbf{I} - \mathbf{G})\right] + \frac{1}{1-n} \ln M^{(n)}(\mathbf{F}). \tag{8}$$

Note that the above arguments are also valid for quantum spin chains that can be mapped to free fermions via Jordan-Wigner transformation such as the TFI chain.

We emphasize once again that the Shannon-Rényi entropy concerns the diagonal elements of the density matrix in the computational basis. On the other hand, the so called von-Neumann-Rényi entropy is defined as

$$S_n(l) = \frac{1}{1-n} \ln\left(\mathrm{tr}\rho_l^n\right). \tag{9}$$

For free fermions, the latter entropy can be efficiently computed in polynomial time using the correlation matrix method and has been extensively studied, see [8,9].

The Shannon-Rényi entropy of the ground state of a quantum chain normally shows a volume-law behavior and presents some information about the phase transition and its universality class [6, 10–14].

Instead of calculating the Shannon-Rényi entropy of the full system one can also use marginal probabilities and calculate the entropy for the subsystems. These entropies as their full system counterparts also show a volume law behavior, however, at the phase transition point there is a logarithmic subleading term the coefficient of which shows an interesting *universal* behavior. Significantly, for specific values of the Rényi index, the coefficient associated with the logarithm is influenced by the central charge, a key parameter in two-dimensional conformal field theories [3, 7, 15–22]. The formation probabilities have also been studied in depth for subsystems of certain free fermions [3, 23–27]. For results on the full system see [11, 28]. These probabilities have been also investigated in experiments [29, 30].

Most notably, the Shannon-Rényi entropy of quantum critical spin chains, such as the XXZ and TFI chains, has recently been measured experimentally in various bases for critical systems using the IBM quantum computer [30]. Furthermore, for the first time, the central charge of

the underlying conformal field theory (CFT) was successfully estimated using the Shannon-Rényi entropy in the same experiment [30].

Although one can calculate analytically certain formation probabilities in the thermodynamic limit, the results for the Shannon-Rényi entropy has been entirely numerical. Our paper is the first step to tackle this problem analytically using various mean field schemes for the ground state of the TFI chain.

## 2.2 Shannon-Rényi entropy of finite temperature quadratic fermionic systems

Consider the free fermionic system in a finite temperature $\beta$. Then the density matrix of the system is:

$$\rho_\beta = \frac{e^{-\beta H_{\text{free}}}}{Z_\beta}.$$ (10)

The normalization factor is $Z_\beta = \det[\mathbf{I} + \mathbf{T}]^{\frac{1}{2}}$ where

$$\mathbf{T} = \exp\left[-\beta \begin{pmatrix} \mathbf{M} & \mathbf{N} \\ \mathbf{-N} & \mathbf{-M} \end{pmatrix}\right] \equiv \begin{pmatrix} \mathbf{T}_{11} & \mathbf{T}_{12} \\ \mathbf{T}_{21} & \mathbf{T}_{22} \end{pmatrix}.$$ (11)

Because of the Wick's theorem one can again write all the correlation functions with respect to the correlation matrix $iG^\beta_{jk} = \text{Tr}\left[\rho_\beta \bar\gamma_j \gamma_k\right]$. After finding the correlation matrix one can write back the density matrix in the coherent basis with respect to the $\mathbf{G}^\beta$ matrix as the Eq. 4. This shows that the Shannon-Rényi entropy of finite temperature density matrix of free fermions is an SPPM problem. A more direct approach is to write the density matrix (10) in coherent bases. This can be done first by using the Balian-Brezin decomposition formula to decompose the exponential to three seperate exponentials and then using the well-known formulas of fermionic coherent states [31]. This procedure leads to a formula similar to Eq. 4, i.e.

$$P(C) = \frac{1}{Z_\beta} \det \mathbf{F}^\beta_C,$$ (12)

with the $\mathbf{F}^\beta$ matrix that can be derived from the following equation:

$$\mathbf{F}^\beta = \mathbf{T}_{12}(\mathbf{T}_{22})^{-1} + \mathbf{T}_{22}^{-1}.$$ (13)

This method gives the same result as the one based on first finding the $\mathbf{G}^\beta$ matrix and then writing the density matrix with respect to this matrix.

Having found $\mathbf{F}^\beta$ one can calculate the Shannon-Rényi entropy by following the same procedure as the previous section:

$$R^\beta_n(l) = -\frac{n}{1-n} \ln Z_\beta + \frac{1}{1-n} \ln M^{(n)}(\mathbf{F}^\beta),$$ (14)

relating the Shannon-Rényi entropy of finite temperature density matrix of free fermions to an SPPM problem.

## 2.3 Determinantal point processes

Determinantal Point Processes (DPP) are a family of probabilistic models that have a repulsive behavior [32], and are useful in machine learning where returning a diverse set of objects is important [4]. DPP has many applications in random matrix theory [33] and also many other areas of pure and applied mathematics [34]. For a list of examples of DPP such as descents in random sequences, non-intersecting random walks, edges in random spanning trees, eigenvalues of random matrices, Aztec diamond tilings and relevant references see [4].

DPP can be defined easily using the concept of formation probabilities for quadratic fermionic Hamiltonians. Without loosing any generality consider the set $\mathcal{C} = \{1, 2, ..., l\}$ and the probability measure $\mathcal{P}$ on the $2^{\mathcal{C}}$ subsets of the set $\mathcal{C}$. Here the probability of having subset $\mathbf{C}$ is given by

$$P(\mathbf{C} = C) = \frac{1}{\det[\mathbf{I} + \mathbf{F}]} \det \mathbf{F}_C , \tag{15}$$

where $\mathbf{F}$ and $\mathbf{F}_C$ are defined in accordance with Eqs. 5 and 6. In general, one can start with positive semi-definite matrix $\mathbf{F}$ and define the above probability distribution which in DPP community is called an L-ensemble [35].

Determinantal point processes are usually defined for symmetric $\mathbf{F}$'s by considering marginal probabilities; we say that a random subset $\mathbf{C}$ is drawn according to a DPP if for every $A \subseteq \mathcal{C}$ we have

$$P(A \subseteq \mathbf{C}) = \det \mathbf{K}_A . \tag{16}$$

Here $\mathbf{K} = \mathbf{I} - (\mathbf{I} + \mathbf{F})^{-1}$ and $A$ is the set of rows and columns that should be kept. Equation (16) with symmetric $\mathbf{K}$ is the starting point of DPP studies. Note that the definition given by Eq. 16 is more general than that of Eq. 15. Moreover, the marginal probabilities $P(A \subseteq \mathbf{C})$ need not sum to 1.

SPPM and its generalizations are useful in the study of the Hellinger distance between two DPPs [4] and also in normalization of some deformed DPPs when one is interested in distributions that are more peaked around high-quality, diverse sets than the corresponding DPPs [36, 37].

## 2.4 Partition function of the Hubbard model

Another interesting problem that can be mapped to SPPM is the Hubbard model. For simplicity we take the one dimensional case. The model is described by the following many-body Hamiltonian

$$H_{\text{Hub}} = -t \sum_{<ij>,\sigma} c_{i,\sigma}^{\dagger} c_{j,\sigma} + U \sum_{i=1}^{L} n_{i\uparrow} n_{i,\downarrow} , \tag{17}$$

where $c_{i,\sigma}^{\dagger}$ and $c_{i,\sigma}$ are fermionic creation and annihilation operators respectively with spin $\sigma$ at site $i$ and $n_{i,\sigma} = c_{i,\sigma}^{\dagger} c_{i,\sigma}$ is the corresponding number operator. $t$ and $U$ are the hopping energy and Coulomb on-site interaction energy, respectively. The partition function of this system is as follows:

$$Z^{\text{Hub}}(L) = \text{Tr}\left[ e^{-\beta(H_{\text{Hub}} - \mu N_F)} \right] , \tag{18}$$

where $N_F \equiv \sum_{i\sigma} n_{i\sigma}$ is the fermion number operator. To relate this to a minor problem, we use the standard Berezin integral representation of fermoinic partition function [38]. We write the partition function as a functional integral over Grassmann variables using the identity

$$Z^{\text{Hub}}(L) = \lim_{N \to \infty} Z_N^{\text{Hub}}(L) \tag{19a}$$

$$\equiv \lim_{N \to \infty} \text{Tr}\left[ \left( e^{-\epsilon(H_{\text{Hub}} - \mu N_F)} \right)^N \right] , \tag{19b}$$

where $\epsilon \equiv \frac{\beta}{N}$, and $N$ is the number of imaginary time slices. Considering Grassmann variables at each site $i$ and time slice $k$ by $\bar{\psi}_{ik,\sigma}$ and $\psi_{ik,\sigma}$, one can show that [38]

$$Z_N^{\text{Hub}}(L) = \int_{\psi\bar{\psi}} \exp\left[ -S^{\text{Hub}}\left( \bar{\psi}, \psi \right) \right] , \tag{20}$$

where $S^{\text{Hub}}\left(\bar{\psi}, \psi\right)$ is given in Eq. C.2. Using the notation $\vec{\psi}_\sigma \equiv \left\{\left\{\psi_{ik,\sigma}\right\}_{i=1}^{L}\right\}_{k=1}^{N}$ the size of which is $l \equiv LN$, and also $\Psi \equiv \begin{pmatrix} \vec{\psi}_\uparrow \\ \vec{\psi}_\downarrow \end{pmatrix}$ (and the same for $\bar{\Psi}$) after the dual transformation (Sec. 4.2 for details) we have

$$Z_N^{\text{Hub}}(L) = \left(1 + m^2\right)^{-l} \det\left(\mathbf{A}_{\text{Hub}} + \frac{m}{m^2 + 1}\mathbf{I}\right)^2 M^{(2)}\left(m\mathbf{I} - \left(\mathbf{A}_{\text{Hub}} + \frac{m}{m^2 + 1}\mathbf{I}\right)^{-1}\right), \qquad (21)$$

where $m^2 = -1 \pm \frac{1}{\sqrt{-U\epsilon}}$, and $\mathbf{A}_{\text{Hub}}$ is a $t$- and $\mu$-dependent matrix given by Eq. C.8, and $\mathbf{I}$ is an identity matrix, the dimension of both being $NL \times NL$.

# 3 Summary of the main results

In this section we briefly review some of the main contributions of this paper. As we mentioned in the previous section some interesting physical and applied mathematics problems can be mapped to the SPPM problem. It was already realized in [3] that the Shannon-Rényi entropy of the ground states of free fermionic Hamiltonians is related to the SPPM problem. The application of the same problem in machine learning community was already appreciated in [4]. To our knowledge, the mapping of the partition function of certain interacting fermionic systems such as the Hubbard model to the SPPM has not been noticed before. The most important contribution of this paper is writing a Berezin integral representation, i.e. Eq 23 for the SPPM problem for generic matrices in Sec. 4. This representation is useful to explicitly map the problem of the partition function of the Hubbard model to the SPPM using the Sourlas duality [39] which we slightly generalize in Sec. 4. In addition one can apply Hubbard-Stratonovich kind of transformations to this Grassmann representation to find other formulas that might be useful for tackling the SPPM problem for certain matrices.

The Berezin integral representation, which resembles a path integral, shows some remarkable symmetries such as $SU(n)$ symmetry and its subgroups such as, $U(1)$ and axial $U(1)$, the symmetric group symmetry, particle-hole symmetry and a few more that we discuss in depth in Sec. 5. Most of these symmetries exist for generic matrices and are responsible for many identities for the two point functions. In the case of symmetric and anti-symmetric matrices there are extra symmetries that we also classify in the same section.

The SPPM for generic matrices is an NP hard problem and one needs to tackle the problem using approximate methods. For $n = 2$ we develop three mean field (MF) methods to get an estimate for the SPPM in Sec. 6. The first method is based on directly writing the interaction term of the Grassmann representation with respect to quadratic terms. We find the consistency relations and discuss how the symmetries discussed in Sec. 5 can be broken in various circumstances. These are summarized in Tables 3, 4 and 5. We also discuss how one can verify the stability of the solutions using a bosonic version of the SPPM and saddle point approximation. We improve the MF results by going to the dual space (from now on we call the space resulting from the Sourlas transformation as "dual space"). The third method is based on Jensen inequality which proves that the MF results give a lower bound to the SPPM. This method can be also considered as a variational method and can be a starting point of application of numerical optimization methods in estimating SPPM.

To show that the MF approximation is useful we discuss two examples. The first example is the discrete Laplacian in one dimension which we use as a benchmark. We first solve the SPPM for arbitrary $n$s exactly and discuss the statistical mechanics of the rooted spanning forests on a chain. Then we tackle the problem using MF approximation and show that the MF gives a good approximation of the SPPM especially in the dual space. The second example which is our main example is the Shannon-Rényi entropy of the ground state of the Ising

chain. This problem has been studied previously using numerical calculations in [40]. We apply the MF approximation and find the solution for the consistency relations. Remarkably we find that the consistency relations have a trivial solution in the paramagnetic phase and non-trivial solutions appear just below the critical magnetic field. We show that the MF gives a very good approximation of the Shannon-Rényi entropy in the entire phase diagram. Most importantly, the MF approximation not only is able to detect the phase transition point exactly it also provides very good estimates of the universal quantities. Our MF formula is apparently the first analytic result for the Shannon-Rényi entropy in the fermionic systems.

# 4 A Berezin-Grassmann integral representation

In this section we present a Berezin-Grassmann representation for the sum of principal minors of an arbitrary matrix, for a summary of useful Berezin integrals over Grassmann variables and notations see Appendix (A). We start with $n = 1$. Using the Grassmann representation of determinant one can easily check that

$$M^{(1)}(\mathbf{A}) = \sum_I \int_{\chi,\bar{\chi}} \left( \prod_{i \in I} \bar{\chi}_i \chi_i \right) w_{\mathbf{A}}^{(1)}(\bar{\chi}, \chi) \tag{22a}$$

$$= \int_{\chi,\bar{\chi}} w_{\mathbf{A}}^{(1)}(\bar{\chi}, \chi) \exp \left[ \sum_{j=1}^l (\bar{\chi}_j \chi_j) \right] \tag{22b}$$

$$= \det [\mathbf{I} + \mathbf{A}], \tag{22c}$$

where $\{\bar{\chi}_i, \chi_i\}_{i=1}^l$ are independent Grassmann variables with the property $\chi_j \chi_k + \chi_k \chi_j = 0$ and $\chi_j^2 = 0$ (the same holds for $\bar{\chi}_j$) for all $j, k = 1, ..., l$, $\chi$ is an $l$-vector $(\chi_1, \chi_2, ..., \chi_l)$ (the same for $\bar{\chi}$), $w_{\mathbf{A}}^{(1)}(\bar{\chi}, \chi) \equiv \exp \left[ \sum_{jk=1}^l \bar{\chi}_j a_{jk} \chi_k \right]$, and $\int_{\bar{\chi}, \chi} \equiv \int \left( \prod_{i=1}^l d\bar{\chi}_i d\chi_i \right)$ is a Berezin integral of Grassmann variables. The second line is easily understood by expanding the exponential term and using the properties of Berezin integrals of Grassmann variables. The generalization of the above formula to $n > 1$ needs considering $n$ copies/replicas of Grassmann variables, i.e. $\left\{ \left\{ \bar{\chi}_j^{(r)}, \chi_j^{(r)} \right\}_{j=1}^l \right\}_{r=1}^n$ with a generalized integrand. The generalization of this equation is expressed as follows:

$$M^{(n)}(\mathbf{A}) = \int_{\bar{\chi},\chi} w_{\mathbf{A}}^{(n)} \{\bar{\chi}, \chi\}, \tag{23}$$

where the definitions are generalized to

$$\int_{\bar{\chi},\chi} \equiv \int \left( \prod_{j=1}^l \prod_{r=1}^n d\bar{\chi}_j^{(r)} d\chi_j^{(r)} \right) \quad \text{and} \quad \chi^T \equiv \left( \chi^{(1)}, \chi^{(2)}, ..., \chi^{(n)} \right)$$

(note that each $\chi^{(r)}$ is an $l$-vector $\left( \chi_1^{(r)}, \chi_2^{(r)}, ..., \chi_l^{(r)} \right)^T$, all definitions applies also for $\bar{\chi}^{(r)}$), and

$$w_{\mathbf{A}}^{(n)} \{\bar{\chi}, \chi\} \equiv w_{\mathbf{A}}^{(1)} \left( \bar{\chi}^{(1)}, \chi^{(1)} \right) w_{\mathbf{A}}^{(1)} \left( \bar{\chi}^{(2)}, \chi^{(2)} \right) ... w_{\mathbf{A}}^{(1)} \left( \bar{\chi}^{(n)}, \chi^{(n)} \right)$$

$$\times \exp \left[ \frac{1}{n!} \sum_{j=1}^l \left( \sum_{r=1}^n \bar{\chi}_j^{(r)} \chi_j^{(r)} \right)^n \right]. \tag{24}$$

One can cast this equation into the following more compact and appropriate form

$$w_{\mathbb{A}}^{(n)}\{\bar{\chi},\chi\} \equiv w_{\mathbf{A}}^{(n)}\{\bar{\chi},\chi\} = \exp\left[\bar{\chi}\mathbb{A}_n\chi + \frac{1}{n!}\sum_{j=1}^{l}\left(\sum_{r=1}^{n}\bar{\chi}_j^{(r)}\chi_j^{(r)}\right)^n\right], \tag{25}$$

where we have defined an $(nl) \times (nl)$ block-diagonal matrix

$$\mathbb{A}_n \equiv \mathrm{diag}_n\{\mathbf{A},\mathbf{A},...,\mathbf{A}\}. \tag{26}$$

We note that the only non-vanishing contribution to the second term in the exponential of Eq. 25 is

$$\frac{1}{n!}\left(\sum_{r=1}^{n}\bar{\chi}_j^{(r)}\chi_j^{(r)}\right)^n \to \prod_{r=1}^{n}\bar{\chi}_j^{(r)}\chi_j^{(r)}, \tag{27}$$

then, by expanding the exponential term and using the well-known Berezin integrals (Appendix A) one can easily prove the equation.

Putting another way, we are dealing with a system with a partition function given by an effective action $\mathcal{S}_{\mathbf{A}}^{(n)}\{\bar{\chi},\chi\} \equiv \log w_{\mathbb{A}}^{(n)}\{\bar{\chi},\chi\}$, i.e.

$$\mathcal{S}_{\mathbf{A}}^{(n)}\{\bar{\chi},\chi\} = \bar{\chi}\mathbb{A}_n\chi + \frac{1}{n!}\sum_{j=1}^{l}\left(\sum_{r=1}^{n}\bar{\chi}_j^{(r)}\chi_j^{(r)}\right)^n. \tag{28}$$

We call the space of copies as the *replica space*, which is $n$-dimensional, while $j = 1,...,l$ enumerates the matrix elements, which we call *matrix space*. The above analogy is especially true if $\mathbf{A}_n$ is a semi-positive definite matrix. It is worth mentioning that, using the change of variables $\chi_j^{(r)} \to i\bar{\chi}_j^{(r)}$ and $\bar{\chi}_j^{(r)} \to i\chi_j^{(r)}$ one can easily show that for arbitrary $n$ we have

$$M^{(n)}(\mathbf{A}^T) = M^{(n)}(\mathbf{A}). \tag{29}$$

Also for even $n$ values, using $\bar{\chi}_j^{(r)} \to -\bar{\chi}_j^{(r)}$ and $\chi_j^{(r)} \to \chi_j^{(r)}$ one finds

$$M^{(n)}(-\mathbf{A}) = M^{(n)}(\mathbf{A}), \tag{30}$$

which is rather expected from the basic properties of principal minors.

## 4.1 Sum of the squares of the principal minors

In this subsection we summarize some results regarding sum of the squares of the principal minors, i.e. $M^{(2)}(\mathbf{A})$. This is the case which we comprehensively investigate using the mean field technique. For $n = 2$ we have

$$\begin{aligned} M^{(2)}(\mathbf{A}) &= \int_{\bar{\chi}\chi}\exp\left[\mathcal{S}_{\mathbf{A}}^{(2)}\{\bar{\chi},\chi\}\right] \\ &= \int_{\bar{\chi}\chi}\exp\left[\bar{\chi}\mathbb{A}\chi + \frac{1}{2}\sum_{j=1}^{l}\left(\bar{\chi}_j^{(1)}\chi_j^{(1)} + \bar{\chi}_j^{(2)}\chi_j^{(2)}\right)^2\right], \end{aligned} \tag{31}$$

where $\mathbb{A} \equiv \begin{pmatrix} \mathbf{A} & 0 \\ 0 & \mathbf{A} \end{pmatrix}$, $\chi \equiv \begin{pmatrix} \chi^{(1)} \\ \chi^{(2)} \end{pmatrix}$, and $\bar{\chi} \equiv \left(\bar{\chi}^{(1)}, \bar{\chi}^{(2)}\right)$. Note that the second term in the exponential can be also written as $\sum_{j=1}^{l}\bar{\chi}_j^{(1)}\chi_j^{(1)}\bar{\chi}_j^{(2)}\chi_j^{(2)}$, which is reminiscent of the interaction term in the Hubbard model (see the followings).

Using the discrete Hubbard-Stratanovich (HS) trick one can write (31) as:

$$M^{(2)}(\mathbf{A}) = \frac{1}{2^l} \sum_S [\det \mathbf{A}_S]^2, \tag{32}$$

where $\mathbf{A}_S = \mathbf{A} + \mathbf{S}$ in which $\mathbf{S}$ is a diagonal matrix with the non-zero elements $\{s_1, s_2, ..., s_l\}$ and $s_j = \pm 1$. The above formula can be easily generalized to the $Z_q$ case as follows:

$$M^{(2)}(\mathbf{A}) = \frac{1}{q^l} \sum_S [\det \mathbf{A}_S \det \mathbf{A}_{S^*}], \tag{33}$$

where now $s_j = e^{\frac{2\pi i p_j}{q}}$ with $p_j \in \{1, 2, ..., q\}$.

A randomized version of the above formula exits, see [2], in which one can assume that $s_j$'s are independent symmetric random variables with first moment equal to zero, and second moment equal to 1. Then we have

$$M^{(2)}(\mathbf{A}) = \mathbb{E}[(\det \mathbf{A}_S)^2], \tag{34}$$

where $\mathbb{E}$ is the expectation value. The above formulas can be useful in numerical calculations for certain matrices. Note that if instead of discrete HS one uses the continuous version then we will have

$$M^{(2)}(\mathbf{A}) = \left(\frac{1}{\sqrt{2\pi}}\right)^l \int_{-\infty}^{\infty} \prod_{j=1}^{l} ds_j \, e^{-\frac{1}{2}\sum_k s_k^2} [\det(\mathbf{A}_S)]^2, \tag{35}$$

which is a especial case of Eq. (34). The above version of HS trick is a good starting point for tackling the problem of SPPM using the saddle point approximation.

Another variant can be derived by first change of variables on the Grassmann variables and then use a generalized version of the HS trick. However, in this case one needs to introduce $l^2$ variables. The usefulness of different versions depends on the matrix and application.

## 4.2 Strong-weak coupling duality

Consider $\chi, \bar{\chi}, \phi$ and $\bar{\phi}$ as two-component Grassmann variables with the components $\chi^{(1)}$, $\chi^{(2)}$ and $\phi^{(1)}, \phi^{(2)}$ respectively, with the same definition for the fields with bar (from now on all variables on the left of expression with "bar" contain also a transpose). Using properties of Berezin integration over Grassmann variables (Appendix A and B) one can show that (see Eq. B.14) [39]:

$$\int_{\chi, \bar{\chi}} \exp\left\{\sum_{jk} \bar{\chi}_j \mathbb{A}_{jk} \chi_k + \frac{\lambda}{2} \sum_j (\bar{\chi}_j \chi_j)^2\right\} = \prod_j \left[(u_j + m_j^2)^{-1}\right] \det[\mathbb{A}'] \tag{36}$$

$$\times \int_{\bar{\phi}, \phi} \exp\left\{\sum_{jk} \bar{\phi}_j \mathbb{N}_{jk} \phi_k + \frac{1}{2} \sum_j u_j (\bar{\phi}_j \phi_j)^2\right\},$$

where $u_j$ and $m_j$ are some arbitrary numbers that are related by the relations

$$\mathbb{N}_{jk} \equiv m_j \delta_{jk} - \left[(\mathbb{A}')^{-1}\right]_{jk}, \qquad \mathbb{A}'_{jk} \equiv \mathbb{A}_{jk} + \delta_{jk} \frac{m_j}{u_j + m_j^2}, \qquad \lambda = \frac{u_j}{(u_j + m_j^2)^2}. \tag{37}$$

Note that the above relation is more general than the one introduced in Ref [39]. When all the $m_j$ and $u_j$'s are equal we recover the result in [39]. Using the above duality one can show

$$M^{(2)}(\mathbf{A}) = \prod_j \left[(u_j + m_j^2)^{-1}\right] \det[\mathbb{A}'] \int_{\bar{\phi}, \phi} \exp\left[\sum_{jk} \bar{\phi}_j \mathbb{N}_{jk} \phi_k + \frac{1}{2} \sum_j u_j (\bar{\phi}_j \phi_j)^2\right], \tag{38}$$

which serves as the weak-strong duality transformation. To see this, note that $\lambda$ is the inverse of $u$ in the large $u$ limit. Using Eq. B.14, we see that

$$M^{(2)}(\mathbf{A}) = c_l M_u^{(2)}(\mathbf{N}), \tag{39}$$

where $M_u^{(2)}$ and $c_l$ are given by Eq. B.14, and $\mathbb{N}$ is given by Eq. 37. Here we have defined

$$\mathbf{N} \equiv \mathbf{I}_u - (\mathbf{A} + \mathbf{D}_u)^{-1}, \tag{40}$$

where

$$\mathbf{D}_u \equiv \text{diag} \left\{ \frac{m_j}{u_j + m_j^2} \right\}_{j=1}^l, \quad \text{and} \quad \mathbf{I}_u \equiv \text{diag} \left\{ m_j \right\}_{j=1}^l, \tag{41}$$

so that

$$\mathbb{N} = \begin{pmatrix} \mathbf{N} & \mathbf{0} \\ \mathbf{0} & \mathbf{N} \end{pmatrix}. \tag{42}$$

It is also worth mentioning that we may start with the positive semi-definite matrix $\mathbf{A}$, however, in dual space the matrix $\mathbf{N}$ may not be a positive semi-definite matrix.

The above duality might be useful when one deals with a problem with large $\lambda$ values, and the equation helps to transform it to a problem with small *interaction* coupling to be treated perturbatively. The especial case of the duality is when $m = 0$ and $u = 1$, i.e. $M^{(2)}(\mathbf{A}) = (\det \mathbf{A})^2 \times M^{(2)}(\mathbf{A}^{-1})$ which can be derived also out of the relation between the principal minors of the inverse of a matrix with the principal minors of the matrix itself. It can be also proved easily using the Hubbard-Stratonovich trick. For $\lambda = 1$ which is an SPPM problem, we have $m_j = \pm \sqrt{\sqrt{u_j} - u_j}$, or equivalently $u_\pm = \frac{1}{2} - m^2 \pm \sqrt{\frac{1}{4} - m^2}$, see Appendix B for details. Note that in the dual space mean field theory results are the most effective when we take negative branch $m_j = -\sqrt{\sqrt{u_j} - u_j}$.

Equation 38 can be exploited to expand Eq. 31 for weak couplings $u_j$. By expanding the interacting part of the exponential term one finds up to $O(u^3)$

$$\begin{aligned} M^{(2)}(\mathbf{A}) &= \left( \prod_j \left[ (u_j + m_j^2)^{-1} \right] \right) \det[\mathbb{A}'] \left( \det[\mathbb{N}] + \sum_{k_1} u_{k_1} \det\left[ \mathbb{N}_{I_{k_1}^c J_{k_1}^c} \right] \right. \\ &\qquad\qquad \left. + \sum_{k_1 > k_2} u_{k_1} u_{k_2} \det\left[ \mathbb{N}_{I_{k_1,k_2}^c J_{k_1,k_2}^c} \right] + ... \right) \\ &= \left( \prod_j \left[ (u_j + m_j^2)^{-1} \right] \right) \det[\mathbb{A}'] \det[\mathbb{N}] \left( 1 + \sum_{k_1} u_{k_1} G^{(1)} \{k_1\} \right. \\ &\qquad\qquad \left. + \sum_{k_1 > k_2} u_{k_1} u_{k_2} G^{(2)} \{k_1, k_2\} + ... \right), \end{aligned} \tag{43}$$

where $I_{\{k_i\}_{i=1}^s} = [l] \setminus \{k_i\}_{i=1}^s$, $J_{\{k_i\}_{i=1}^s} = [l] \setminus \{k_i\}_{i=1}^s$ are sets $[l]$ minus $\{k_i\}_{i=1}^s$ (for both copies), and $I^c$ and $J^c$ are their complements. The second line of this relation contains summation over $s$-point functions defined as

$$\begin{aligned} G^{(s)} \{k_i\}_{i=1}^s &\equiv \det\left[ \mathbb{N}_{I_{\{k_i\}_{i=1}^s}, J_{\{k_i\}_{i=1}^s}}^{-T} \right] \\ &= \left( \det\left[ \mathbf{N}_{I_{\{k_i\}_{i=1}^s}, J_{\{k_i\}_{i=1}^s}}^{-T} \right] \right)^2 \\ &= \left[ \sum_{\sigma \in S_s} \left( \text{sgn}(\sigma) \prod_{i=1}^s \mathcal{G}^{(1)}(k_i, k_{\sigma(i)}) \right) \right]^2, \end{aligned} \tag{44}$$

where $S_n$ is the set of all permutations, $\mathrm{sgn}(\sigma)$ is $+$ $(-)$ for even (odd) permutations, $\mathcal{G}^{(1)}(k, k') \equiv \left(\mathbf{N}^{-T}\right)_{k,k'}$, and $G^{(1)}(k) = \mathcal{G}^{(1)}(k, k)$. This relation is a Wick expansion. Using this, we can design a diagramatic interpretation for Eq. 43 as follows: To each term in $r$th level we attribute a graph composed of $r$ nodes, and corresponding to each $\mathcal{G}^{(1)}(k_m, k_n)$ we make an edge connecting the nodes $k_m$ and $k_n$. Each $\mathcal{G}^{(1)}(k_m, k_m)$ is a local loop at the node $m$. Suppose that a given diagram $i$ is composed of several connected components $C_i$, for which there are $n_i$ identical copies, leading a symmetry factor $n_i!$. The contribution of such a term in the diagram is $\frac{C_i^{n_i}}{n_i!}$, so that Eq. 43 becomes

$$
\begin{aligned}
M^{(2)}(\mathbf{A}) &= c_l \left(\det \mathbb{N}\right) \prod_i \sum_{i=0}^{\infty} \frac{C_i^{n_i}}{n_i!} \\
&= c_l \left(\det \mathbb{N}\right) e^{\sum_i C_i} = e^{-F^{(2)}(\mathbf{A})}.
\end{aligned}
\tag{45}
$$

This relation shows that for the free energy $F^{(2)}(\mathbf{A})$, the summation is over connected components. This result is a consequence of the linked cluster expansion theorem [41]. One should use this perturbative expansion with caution since the principal minors of $\mathbb{N}$ and the combinatorial number of terms involved in each level grows exponentially with the level of expansion.

An interesting application of the duality is when one uses the mean field method in the dual space. Two examples will be discussed later in the context of the Laplacian matrix and the Rényi entropy of the TFI chain. We see that for both examples there are values of $u$ which the mean field is more stable in the dual space than the original space. The duality with the free parameters $\{u_j\}$'s provide remarkable possibilities to look for the best value of them in which the mean field approximation is most effective.

## 5 Symmetries of the system

In this section we explore various symmetries of the system. In particular, we study the implications of these symmetries on the two point functions. We define the "*expectation value*" of an "*observable*" $O\{\bar{\chi}, \chi\}$ as follows:

$$
\langle O\{\bar{\chi}, \chi\} \rangle \equiv \frac{\int_{\bar{\chi}, \chi} O\{\bar{\chi}, \chi\} w_{\mathbb{A}}^{(n)}\{\bar{\chi}, \chi\}}{\int_{\bar{\chi}, \chi} w_{\mathbb{A}}^{(n)}\{\bar{\chi}, \chi\}} .
\tag{46}
$$

An example is the following $2d$-point functions, which can be exactly written with respect to the minors as:

$$
\left\langle \prod_{r \in S} \bar{\chi}_j^{(r)} \chi_j^{(r)} \right\rangle = \frac{\sum_I \left(\det \mathbf{A}_{I,j}\right)^d \left(\det \mathbf{A}_I\right)^{n-d}}{M^{(n)}(\mathbf{A})} ,
\tag{47}
$$

where $S$ is a subset of $\{1, 2, ..., n\}$ with $d$ elements. In this relation the sum is over all subsets $I$, and $\mathbf{A}_I$ is the matrix $\mathbf{A}$ with removed rows and columns corresponding to $I$, and $\mathbf{A}_{I,j}$ is the same with an additional removed $j$th row and column. Generally, the properties of $\langle O\{\bar{\chi}, \chi\} \rangle$ depend on the symmetries of $\mathbb{A}$. In the following subsections we summarize some symmetry-related properties of the system.

## 5.1  $U(n)$ symmetry

Consider the following unitary transformation

$$
\begin{pmatrix} \chi^{(1)} \\ \chi^{(2)} \\ . \\ . \\ \chi^{(n)} \end{pmatrix} \to U_n \begin{pmatrix} \chi^{(1)} \\ \chi^{(2)} \\ . \\ . \\ \chi^{(n)} \end{pmatrix}, \qquad \begin{pmatrix} \bar{\chi}^{(1)} \\ \bar{\chi}^{(2)} \\ . \\ . \\ \bar{\chi}^{(n)} \end{pmatrix} \to U_n^* \begin{pmatrix} \bar{\chi}^{(1)} \\ \bar{\chi}^{(2)} \\ . \\ . \\ \bar{\chi}^{(n)} \end{pmatrix},
\tag{48}
$$

where $U_n$ is a $n \times n$ unitary matrix, and $U_n^*$ is its complex conjugate. It is straightforward to demonstrate the invariance of the effective action given by Eq. 28 under this transformation. To illustrate this, we observe that $U_n$ exclusively acts within the "replica space" and behaves as a unit matrix in the "matrix space", resulting in the commutation of $\mathbb{A}$ with $U_n$. Additionally, $\sum_{m=1}^{n} \bar{\chi}^{(m)} \chi^{(m)}$ remains unchanged under this transformation.

To grasp the implications of this transformation, we begin by considering the especial case of $n = 2$ and then return to the general $n$ values. In this specific case, the representation of $U_2$ is as follows:

$$
U_2 = e^{i\gamma} \begin{pmatrix} a & b \\ -b^* & a^* \end{pmatrix},
\tag{49}
$$

where $\gamma$ is a $U(1)$ parameter, and $a$ and $b$ are some complex parameters with an extra condition $|a|^2 + |b|^2 = 1$ (showing the $SU(2)$ part), leaving totally four independent parameters in the transformations. This symmetry transformation gives rise to the strong restrictions for two point functions. As an example we have

$$
\begin{aligned}
\left\langle \bar{\chi}_j^{(1)} \chi_k^{(1)} \right\rangle \to \left\langle \bar{\chi}_j^{(1)} \chi_k^{(1)} \right\rangle &+ X_1 \left( \left\langle \bar{\chi}_j^{(2)} \chi_k^{(2)} \right\rangle - \left\langle \bar{\chi}_j^{(1)} \chi_k^{(1)} \right\rangle \right) \\
&+ X_2 \left\langle \bar{\chi}_j^{(1)} \chi_k^{(2)} \right\rangle + X_2^* \left\langle \bar{\chi}_j^{(2)} \chi_k^{(1)} \right\rangle,
\end{aligned}
\tag{50}
$$

where $X_1 \equiv |b|^2$, $X_2 \equiv a^* b$ and $X_2^*$ are independent variables. Given that there are three independent parameters for the $SU(2)$ part, one finds that the three last coefficients should be zero, implying that

$$
\left\langle \bar{\chi}_j^{(1)} \chi_k^{(1)} \right\rangle = \left\langle \bar{\chi}_j^{(2)} \chi_k^{(2)} \right\rangle,
\tag{51a}
$$

$$
\left\langle \bar{\chi}_j^{(1)} \chi_k^{(2)} \right\rangle = \left\langle \bar{\chi}_j^{(2)} \chi_k^{(1)} \right\rangle = 0.
\tag{51b}
$$

A subgroup of $U(2)$ is obtained when we choose $\gamma = \frac{1}{2}(\alpha_1 + \alpha_2)$, $a = e^{\frac{i}{2}(\alpha_1 - \alpha_2)}$, $b = 0$ giving rise to

$$
U_2 \to U(1) \otimes U(1) \equiv \begin{pmatrix} e^{i\alpha_1} & 0 \\ 0 & e^{i\alpha_2} \end{pmatrix}.
\tag{52}
$$

This symmetry implies that (the same for $\bar{\chi}$'s)

$$
\left\langle \chi_j^{(1)} \chi_k^{(2)} \right\rangle = \left\langle \chi_j^{(1)} \chi_k^{(1)} \right\rangle = \left\langle \chi_j^{(2)} \chi_k^{(2)} \right\rangle = 0.
\tag{53}
$$

The $SU(2)$ subgroup ($\gamma = 0$) which rotates the replica can be represented in terms of the new parameters in the form $R_2(\phi, \hat{n}) \equiv U_2(\phi, \hat{n}, \gamma = 0)$, where

$$
R_2(\phi, \hat{n}) = \exp\left[ -i\frac{\phi}{2} \hat{n}.\vec{\sigma} \right] = \mathbb{I} \cos\frac{\phi}{2} - i\hat{n}.\vec{\sigma} \sin\frac{\phi}{2}.
\tag{54}
$$

Here $\phi$ is a real parameter and $\hat{n} \equiv (n_x, n_y, n_z)$ is a real unit vector, and $\sigma_i$'s ($i = x, y, z$) are Pauli matrices. Using this, one can easily show that

$$
\begin{aligned}
\text{Re}[a] &= \cos\frac{\phi}{2}, & \text{Im}[a] &= -n_z \sin\frac{\phi}{2}, \\
\text{Re}[b] &= -n_y \sin\frac{\phi}{2}, & \text{Im}[b] &= -n_x \sin\frac{\phi}{2}.
\end{aligned}
\tag{55}
$$

For example, in the case $n_x = n_y = 0$, we have

$$
R_2(\phi, n_z = 1) = e^{-i\frac{\phi}{2}\sigma_z} = \begin{pmatrix} e^{-i\frac{\phi}{2}} & 0 \\ 0 & e^{i\frac{\phi}{2}} \end{pmatrix},
\tag{56}
$$

under which $\left\langle \bar{\chi}_j^{(1)} \chi_k^{(2)} \right\rangle \to e^{i\phi} \left\langle \bar{\chi}_j^{(1)} \chi_k^{(2)} \right\rangle$. This transformation, which is called the *axial $U(1)$* implies that $\left\langle \bar{\chi}_j^{(1)} \chi_k^{(2)} \right\rangle = 0$ (for all $j$ and $k$ values). Note also that the transformations

$$
R_2(3\pi, n_x = 1) = i\sigma_x, \tag{57a}
$$
$$
R_2(3\pi, n_y = 1) = i\sigma_y, \tag{57b}
$$

interchange the replica. Both imply $\left\langle \bar{\chi}_j^{(1)} \chi_k^{(1)} \right\rangle \leftrightarrow \left\langle \bar{\chi}_j^{(2)} \chi_k^{(2)} \right\rangle$, and also

$$
\left\langle \bar{\chi}_j^{(1)} \chi_k^{(2)} \right\rangle \leftrightarrow \pm \left\langle \bar{\chi}_j^{(2)} \chi_k^{(1)} \right\rangle,
$$

where in the latter $+(-)$ stands for $R_2(3\pi, n_x = 1)$ ($R_2(3\pi, n_y = 1)$), in agreement with Eq. 51.

The above results can be directly generalized to all $n$ values as follows:

$$
\left\langle \bar{\chi}_j^{(r)} \chi_k^{(r)} \right\rangle = \left\langle \bar{\chi}_j^{(r')} \chi_k^{(r')} \right\rangle, \qquad \text{for all } r, r', j, k, \tag{58a}
$$
$$
\left\langle \bar{\chi}_j^{(r)} \chi_k^{(r')} \right\rangle = \left\langle \bar{\chi}_j^{(r')} \chi_k^{(r)} \right\rangle = 0, \qquad \text{for all } r \neq r', j, k, \tag{58b}
$$
$$
\left\langle \chi_j^{(r)} \chi_k^{(r')} \right\rangle = \left\langle \bar{\chi}_j^{(r)} \bar{\chi}_k^{(r')} \right\rangle = 0, \qquad \text{for all } r, r', j, k. \tag{58c}
$$

We will derive them using some especial subgroups of the $U(n)$ group.

## 5.2 $U^n(1)$ symmetry

A subgroup of $U(n)$ is the generalization of Eq. 52, i.e. $U^n(1) \equiv U(1) \otimes U(1) \otimes ... \otimes U(1)$, which is arguably the most obvious symmetry of $\mathcal{S}_A^{(n)}$. It means $\chi_j^{(r)} \to e^{i\alpha_r} \chi_j^{(r)}$, $\bar{\chi}_j^{(r)} \to e^{-i\alpha_r} \bar{\chi}_j^{(r)}$, $r = 1, ..., n$, where $\alpha_r$'s are arbitrary real numbers. A direct consequence of this symmetry is

$$
\left\langle \bar{\chi}_j^{(r)} \chi_k^{(r')} \right\rangle = 0, \tag{59}
$$

for $r \neq r'$. It is worth exploring the properties of some subgroups of the group $U^n(1)$ and the resulting identities for the two-point functions in the following subsections.

### 5.2.1 $U(1)$ and axial $U(1)$ symmetry

An important subgroup of $U^{(n)}(1)$, namely $U(1)$ group, corresponds to the case where all $\alpha_r$'s are identical, i.e. $\left( \bar{\chi}_j^{(r)}, \chi_j^{(r)} \right) \to \left( e^{-i\alpha} \bar{\chi}_j^{(r)}, e^{i\alpha} \chi_j^{(r)} \right)$. A system with this symmetry is expected to have the property

$$
\left\langle \chi_j^{(r)} \chi_k^{(r')} \right\rangle = \left\langle \bar{\chi}_j^{(r)} \bar{\chi}_k^{(r')} \right\rangle = 0, \tag{60}
$$

for all $r, r' = 1, 2, ..., n$.

Another subgroup of $U^n(1)$ is the so-called *axial $U(1)$* group, defined by the transformation Eq. 56 with $\alpha = -2\phi$ (in the case $n = 2$). More explicitly it transforms the pair $\left( \bar{\chi}_j^{(1)}, \chi_j^{(1)} \right) \to \left( e^{-i\alpha} \bar{\chi}_j^{(1)}, e^{i\alpha} \chi_j^{(1)} \right)$, and $\left( \bar{\chi}_j^{(2)}, \chi_j^{(2)} \right) \to \left( e^{i\alpha} \bar{\chi}_j^{(2)}, e^{-i\alpha} \chi_j^{(2)} \right)$. This transformation in extended easily to any arbitrary $n$.

### 5.2.2 The parity and sectorial parity (SP) symmetry

The parity symmetry $\bar{\chi}, \chi \to -\bar{\chi}, -\chi$ implies that the expectation value for all strings $\langle X \rangle$ is zero ($X \equiv$ a string of $\chi$s and $\bar{\chi}$s) if the total number of $\chi$'s and $\bar{\chi}$'s in the string is odd.

We also define a *sectorial parity (SP) transformation* $\bar{\chi}_j^{(r)}, \chi_j^{(r)} \to \text{sgn}(r)\bar{\chi}_j^{(r)}, \text{sgn}(r)\chi_j^{(r)}$, where $\text{sgn}(r) = \pm 1$. It corresponds to $U^{(n)}(1)$ with $\alpha_r = \pi$ or $2\pi$. To be more specific, consider the transformation $(\bar{\chi}_j^{(r)}, \chi_j^{(r)}) \to (-\bar{\chi}_j^{(r)}, -\chi_j^{(r)})$, and $(\bar{\chi}_j^{(r')}, \chi_j^{(r')} \to \bar{\chi}_j^{(r')}, \chi_j^{(r')})$, implying that for $r \neq r'$

$$\left\langle \bar{\chi}_j^{(r)} \chi_k^{(r')} \right\rangle = 0, \tag{61}$$

and also

$$\left\langle \bar{\chi}_j^{(r)} \bar{\chi}_k^{(r')} \right\rangle = \left\langle \chi_j^{(r)} \chi_k^{(r')} \right\rangle = 0. \tag{62}$$

### 5.3 The symmetric group (SG)

Equation 23 is invariant under symmetric group which is the permutations of all copies/replicas. This symmetry implies that all multi-point functions are unchanged under exchanging the replicas (upon interchanging $r \leftrightarrow r'$). In particular, it forces the two-point functions to satisfy the following equations

$$\left\langle \bar{\chi}_j^{(r)} \chi_k^{(r')} \right\rangle = \left\langle \bar{\chi}_j^{(r')} \chi_k^{(r)} \right\rangle, \tag{63a}$$

$$\left\langle \bar{\chi}_j^{(r)} \chi_k^{(r)} \right\rangle = \left\langle \bar{\chi}_j^{(r')} \chi_k^{(r')} \right\rangle, \tag{63b}$$

$$\left\langle \chi_j^{(r)} \chi_k^{(r')} \right\rangle = \left\langle \chi_j^{(r')} \chi_k^{(r)} \right\rangle, \tag{63c}$$

for all $r$ and $r'$ values. The last line specially implies that $\left\langle \chi_j^{(r)} \chi_j^{(r')} \right\rangle = 0$.

### 5.4 $SU^n(2)$ and chiral particle-hole (CPH) symmetry for symmetric matrices

Consider the transformation

$$\begin{pmatrix} \bar{\chi}^{(r)} \\ \chi^{(r)} \end{pmatrix} \to \mathcal{SU}_2^{(r)} \begin{pmatrix} \bar{\chi}^{(r)} \\ \chi^{(r)} \end{pmatrix}, \tag{64}$$

where $\mathcal{SU}_2^{(r)}$ is given by Eq. 49, with $\gamma = 0$ and the new parameters which depend on $r$. Given that $\mathbb{A} = \text{diag}\{\mathbf{A}, ..., \mathbf{A}\}$, if $\mathbf{A}$ is symmetric, then one can easily show by a simple expansion that $\bar{\chi}_j^{(r)} \mathbf{A}_{jk} \chi_k^{(r)}$ is invariant under the transformation Eq. 64, and consequently $\mathcal{S}_{\mathbf{A}}^{(n)}$ is invariant. Then, using the fact that this transformation is local in the replica space, the big symmetry group is represented by

$$\mathcal{SU}_2^n \equiv \mathcal{SU}_2^{(1)} \otimes \mathcal{SU}_2^{(2)} \otimes ... \otimes \mathcal{SU}_2^{(n)}. \tag{65}$$

To inspect the consequences of this symmetry group, we consider an especial element $\mathcal{SU}_2^{(r)} \equiv \mathbb{I}^{(1)} \otimes ... \otimes \mathcal{SU}_2^{(r)} \otimes ... \otimes \mathbb{I}^{(n)}$, where $\mathcal{SU}_2^{(r)} = R_2(\phi, n_z = 1)$. This element gives rise to the transformations

$$\left\langle \bar{\chi}_j^{(r)} \bar{\chi}_k^{(r')} \right\rangle \to e^{-i\frac{\phi}{2}(1+\delta_{rr'})} \left\langle \bar{\chi}_j^{(r)} \bar{\chi}_k^{(r')} \right\rangle, \tag{66a}$$

$$\left\langle \chi_j^{(r)} \chi_k^{(r')} \right\rangle \to e^{i\frac{\phi}{2}(1+\delta_{rr'})} \left\langle \chi_j^{(r)} \chi_k^{(r')} \right\rangle, \tag{66b}$$

$$\left\langle \bar{\chi}_j^{(r)} \chi_k^{(r')} \right\rangle \to e^{-i\frac{\phi}{2}(1-\delta_{rr'})} \left\langle \bar{\chi}_j^{(r)} \chi_k^{(r')} \right\rangle, \tag{66c}$$

implying that

$$\left\langle \bar{\chi}_j^{(r)} \bar{\chi}_k^{(r')} \right\rangle = \left\langle \chi_j^{(r)} \chi_k^{(r')} \right\rangle = 0, \qquad \text{for all } r, r', \tag{67a}$$

$$\left\langle \bar{\chi}_j^{(r)} \chi_k^{(r')} \right\rangle = 0, \qquad \text{for all } r \neq r'. \tag{67b}$$

On the other hand, by taking the $\mathcal{SU}_2^{(r)} = U_2(a = 0, b = e^{i\alpha})$, we find that

$$\left\langle \bar{\chi}_j^{(r)} \chi_k^{(r)} \right\rangle = \left\langle \bar{\chi}_k^{(r)} \chi_j^{(r)} \right\rangle. \tag{68}$$

Another especial symmetry for the symmetric matrices is associated with the transformation $\mathcal{SU}_2 = R_2(\pi, n_y = 1)$, which implies $\bar{\chi}_j \to -\chi_j$ and $\chi_j \to +\bar{\chi}_j$. We call it *chiral particle-hole (CPH) symmetry*. This implies that if $\mathbf{A}$ is a symmetric matrix, then we have

$$\left\langle \bar{\chi}_j^{(r)} \chi_k^{(r')} \right\rangle = \left\langle \bar{\chi}_k^{(r')} \chi_j^{(r)} \right\rangle, \tag{69a}$$

$$\left\langle \bar{\chi}_j^{(r)} \bar{\chi}_k^{(r')} \right\rangle = \left\langle \chi_k^{(r')} \chi_j^{(r)} \right\rangle, \tag{69b}$$

for *all* $r$ and $r'$ values. The first line of this equation is just the same as the relation in the first line of Eq. 63 obtained from the *SG* symmetry.

## 5.5 Particle-hole (PH) symmetry for antisymmetric matrices

We define the particle-hole (PH) transformation as the operation of exchanging *particles* and *holes*, i.e. we exchange $\chi_j^{(r)}$ with $\bar{\chi}_j^{(r)}$ for all values of $r$. Under this transformation the interaction term is invariant for even $n$, while the term $\bar{\chi}_j^{(r)} A_{jk} \chi_k^{(r)}$ goes to $-\bar{\chi}_k^{(r)} A_{jk} \chi_j^{(r)} = \bar{\chi}^{(r)} \left(-\mathbf{A}^T\right) \chi^{(r)}$, showing that the effective action is invariant only for antisymmetric matrices. One finds for antisymmetric matrices

$$\left\langle \bar{\chi}_j^{(r)} \chi_j^{(r)} \right\rangle = 0, \tag{70a}$$

$$\left\langle \bar{\chi}_j^{(r)} \chi_k^{(r')} \right\rangle = -\left\langle \bar{\chi}_k^{(r')} \chi_j^{(r)} \right\rangle, \tag{70b}$$

$$\left\langle \bar{\chi}_j^{(r)} \bar{\chi}_k^{(r')} \right\rangle = \left\langle \chi_j^{(r)} \chi_k^{(r')} \right\rangle. \tag{70c}$$

It is worth mentioning that, the second line of this equation, when combined with the first line for $j = k$ in Eq. 63 (SG symmetry) for antisymmetric matrices gives rise to the relation

$$\left\langle \bar{\chi}_j^{(r)} \chi_j^{(r')} \right\rangle = \left\langle \bar{\chi}_j^{(r')} \chi_j^{(r)} \right\rangle = 0. \tag{71}$$

For odd values of $n$ the following argument based on a sectorial PH transformation leads to $\left\langle \bar{\chi}_j^{(r)} \chi_j^{(r)} \right\rangle = 0$ for all $r$ values. A sectorial PH transformation is defined as a PH transformation applied to all $r$ replica except one (arbitrarily chosen) replica $r'$, so that $\bar{\chi}_j^{(r)} \to \chi_j^{(r)}$ and $\chi_j^{(r)} \to \bar{\chi}_j^{(r)}$ for $r \in \{1, ..., r'-1, r'+1, ..., n\}$, while $\bar{\chi}_j^{(r')} \to \bar{\chi}_j^{(r')}$ and $\chi_j^{(r')} \to \chi_j^{(r')}$. By employing the same reasoning as presented earlier, it can be readily demonstrated that Eq. 70 holds true, this time for all $r$ values except $r'$. By varying $r'$, it becomes evident that Eq. 70 remains valid for all $r$ values.

## 5.6 Diagonally equivalent symmetry

Consider a diagonal matrix with non-zero diagonal elements $\mathbf{D}$, then using the Berezin-Grassmann representation it is easy to show that

$$M^{(n)}(\mathbf{A}_D) = M^{(n)}(\mathbf{A}), \tag{72}$$

where $\mathbf{A}_D = \mathbf{D}\mathbf{A}\mathbf{D}^{-1}$. We say that $\mathbf{A}$ and $\mathbf{A}_D$ are diagonally equivalent.

Table 1: The symmetry identities resulting from the symmetries.

| | Identity |
|---|---|
| (I) | $\left\langle \bar{\chi}_j^{(r)} \chi_k^{(r')} \right\rangle = \left\langle \bar{\chi}_j^{(r')} \chi_k^{(r)} \right\rangle = 0$, all $r \neq r'$, all $j, k$ |
| (II) | $\left\langle \bar{\chi}_j^{(r)} \chi_k^{(r)} \right\rangle = \left\langle \bar{\chi}_j^{(r')} \chi_k^{(r')} \right\rangle = 0$, all $r, r', j, k$ |
| (III) | $\left\langle \bar{\chi}_j^{(r)} \chi_k^{(r')} \right\rangle = \left\langle \bar{\chi}_j^{(r')} \chi_k^{(r)} \right\rangle$, all $r, r', j, k$ |
| (IV) | $\left\langle \chi_j^{(r)} \chi_k^{(r')} \right\rangle = \left\langle \chi_j^{(r')} \chi_k^{(r)} \right\rangle$, $\left\langle \chi_j^{(r)} \chi_j^{(r')} \right\rangle = 0$, all $r, r', j, k$, same for $\bar{\chi}$ |
| (V) | $\left\langle \chi_j^{(r)} \chi_k^{(r')} \right\rangle = \left\langle \bar{\chi}_j^{(r)} \bar{\chi}_k^{(r')} \right\rangle = 0$, all $r, r', j, k$ |
| (VI) | $\left\langle \bar{\chi}_j^{(r)} \chi_k^{(r)} \right\rangle = \left\langle \bar{\chi}_k^{(r)} \chi_j^{(r)} \right\rangle$, all $r, j, k$ |
| (VII) | $\left\langle \bar{\chi}_j^{(r)} \bar{\chi}_k^{(r')} \right\rangle = \left\langle \chi_k^{(r')} \chi_j^{(r)} \right\rangle$, all $r, r', j, k$ |
| (VIII) | $\left\langle \bar{\chi}_j^{(r)} \chi_k^{(r')} \right\rangle = \left\langle \bar{\chi}_k^{(r')} \chi_j^{(r)} \right\rangle$, all $r, r', j, k$ |
| (IX) | $\left\langle \bar{\chi}_j^{(r)} \chi_k^{(r')} \right\rangle = -\left\langle \bar{\chi}_k^{(r')} \chi_j^{(r)} \right\rangle$, $\left\langle \bar{\chi}_j^{(r)} \chi_j^{(r)} \right\rangle = 0$, all $r \neq r'$, all $j, k$ |

## 5.7 Permutation symmetry

The principal minors of the matrices $\mathbf{A}$ and $\mathbf{S}_l \mathbf{A} \mathbf{S}_l^T$ are the same, where $\mathbf{S}_l$ is a permutation matrix. This interchanges the *space* index of the grassmann variables, i.e. $\chi_j^{(r)} \to \chi_{s(j)}^{(r)}$ where $s$ is a member of $\mathbf{S}_l$.

## 5.8 A generalization

Equation 1 can be generalized to [37]

$$M^{(n)}(\mathbf{A}^{(1)}, \mathbf{A}^{(2)}, ..., \mathbf{A}^{(n)}) \equiv \sum_I \det \mathbf{A}_I^{(1)} \det \mathbf{A}_I^{(2)} ... \det \mathbf{A}_I^{(n)} , \tag{73}$$

whose associated block-diagonal matrix is a generalization of Eq. 26

$$\mathbb{A}_n \equiv \text{diag}_n \left\{ \mathbf{A}^{(1)}, \mathbf{A}^{(2)}, ..., \mathbf{A}^{(n)} \right\} , \tag{74}$$

where $\mathbf{A}^{(j)}$'s are arbitrary matrices. For the case $\mathbf{A}^{(j)}$'s are different, $U(n)$ is not generally a symmetry of the system, while $U^n(1)$ and $U(1)$ and axial $U(1)$ and SP symmetries remain valid. The latter symmetries imply, for example, that $\left\langle \bar{\chi}_j^{(r)} \chi_j^{(r')} \right\rangle = 0$ for any $r \neq r'$. When $\mathbf{A}^{(j)}$'s are symmetric matrices, $SU^n(2)$ and CPH are the symmetries of the system, while when $\mathbf{A}^{(j)}$'s are antisymmetric, PH symmetry remains valid. For the case $n = 2$ and $\mathbb{A} = \text{diag}_2 \left\{ \mathbf{A}^{(1)}, \mathbf{A}^{(2)} \right\}$, one can observe by inspection that $\sigma_x \mathbb{A} \sigma_x = \sigma_y \mathbb{A} \sigma_y = \tilde{\mathbb{A}} \equiv \text{diag}_2 \left\{ \mathbf{A}^{(2)}, \mathbf{A}^{(1)} \right\}$, while $\sigma_z \mathbb{A} \sigma_z = \mathbb{A}$, and the second term of Eq. 28 remains unchanged, where $\sigma_{x,y,z}$ are Pauli matrices. Using this, one can show that

$$\left\langle f\left(\bar{\chi}^{(1)} \chi^{(1)}\right) \right\rangle_{\mathbb{A}} = \left\langle f\left(\bar{\chi}^{(2)} \chi^{(2)}\right) \right\rangle_{\tilde{\mathbb{A}}} , \tag{75a}$$

$$\left\langle f\left(\bar{\chi}^{(1)} \chi^1 \bar{\chi}^{(2)} \chi^2\right) \right\rangle_{\mathbb{A}} = \left\langle f\left(\bar{\chi}^{(1)} \chi^1 \bar{\chi}^{(2)} \chi^2\right) \right\rangle_{\tilde{\mathbb{A}}} , \tag{75b}$$

where $f$ is an arbitrary function.

## 5.9 Symmetries of the dual representation

In this section we study the symmetries of the system in the dual space for the case $n = 2$, which is given by Eq. 38. For a general $\mathbf{A}$ and arbitrary local parameters $u_j$ and $\lambda_j$ satisfying 37, one

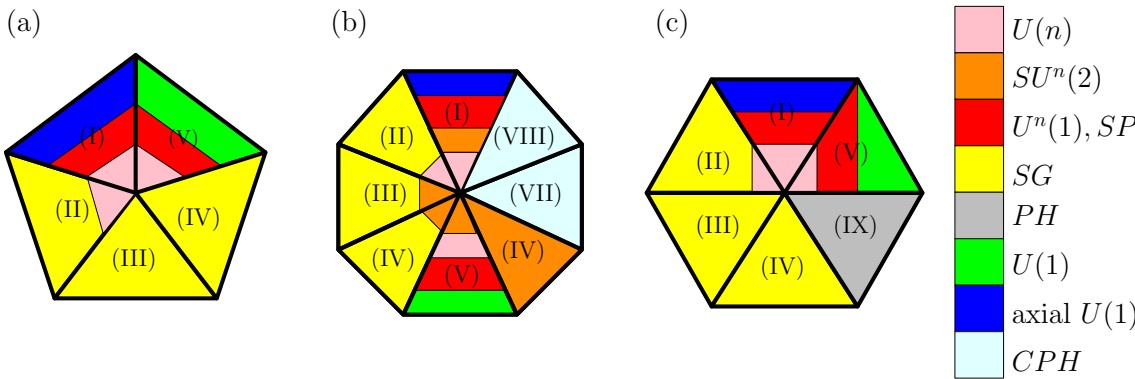

Figure 1: A schematic representation of the consequences of the symmetries for two-point functions. The colors show the symmetries as indicated in the color bar. The symmetry identities listed in Table 1 are considered to be the pieces of polygons. The colors in each piece show that the associated symmetry leads to the symmetry identity that is numbered in Table 1. The pentagon (a) represents the results for *general* matrices, where we have $U(n)$, $U^n(1)$, $SP$, $SG$, $U(1)$ and *axial* $U(1)$ symmetries. The octagon (b) and the hexagon (c) show the results for symmetric and antisymmetric matrices respectively. Note that for symmetric matrices, in addition to the symmetries of general matrices, we have $SU^n(2)$ and $CPH$ symmetries, while for the antisymmetric matrices the $PH$ is included.

expects that $U(2), U^2(1), U(1)$, SG, SP and axial $U(1)$ remain the symmetries of the system. Consequently, the symmetry equations (I-V) hold for the two-point functions in the dual space. Most importantly

$$\left\langle \bar{\phi}_j^{(1)} \phi_k^{(2)} \right\rangle = \left\langle \bar{\phi}_j^{(2)} \phi_k^{(1)} \right\rangle = 0 \,, \tag{76a}$$

$$\left\langle \phi_j^{(r)} \phi_k^{(r')} \right\rangle = \left\langle \bar{\phi}_j^{(r)} \bar{\phi}_k^{(r')} \right\rangle = 0 \,, \tag{76b}$$

for any $r, r' = 1, 2$. Additionally, when $u_j = u, \forall j$, for a circulant matrix **A** the correlation functions are also translational invariant in the dual space.

To summarize this section, in Table. 1 and Fig. 1 we see the effect of the symmteries on two-point functions for three cases: (a) generic matrices, (b) symmetric matrices, and (c) antisymmetric matrices. For a symmetric **A**, we see that **N** in the dual space (Eq. 38 and 37) is symmetric, which leads to the same symmetry identities as **A** (I-VIII identities in Table 1). Note that when the matrix **A** is antisymmetric, **N** is not necessarily antisymmetric, and consequently the PH symmetry may or may not be the symmetry of the system.

# 6 Mean-field theory

In this section, we utilize the MF theory to approximate Eq. 31 when $n = 2$ (generalization to larger values of $n$ is conceptually straightforward). This approach involves employing standard MF schemes like neglecting the fluctuations and using self-consistent equations for two-point functions of the Grassmann variables $\bar{\chi}_j^{(1)}, \chi_j^{(1)}, \bar{\chi}_j^{(2)}$, and $\chi_j^{(2)}$. The self-consistent two point

functions are defined as follows:

$$\Delta_j^{(\bar{1}1)} \equiv \left\langle \bar{\chi}_j^{(1)} \chi_j^{(1)} \right\rangle, \qquad \Delta_j^{(\bar{2}2)} \equiv \left\langle \bar{\chi}_j^{(2)} \chi_j^{(2)} \right\rangle,$$
$$\Delta_j^{(\bar{1}2)} \equiv \left\langle \bar{\chi}_j^{(1)} \chi_j^{(2)} \right\rangle, \qquad \Delta_j^{(\bar{2}1)} \equiv \left\langle \bar{\chi}_j^{(2)} \chi_j^{(1)} \right\rangle, \qquad (77)$$
$$\Delta_j^{(12)} \equiv \left\langle \chi_j^{(1)} \chi_j^{(2)} \right\rangle, \qquad \Delta_j^{(\bar{1}\bar{2})} \equiv \left\langle \bar{\chi}_j^{(1)} \bar{\chi}_j^{(2)} \right\rangle,$$

where $j$ is a spatial coordinate. Throughout this paper, we keep the notation "$\Delta$" only for self-consistent two point functions in MF theory. In a system where all the symmetries are preserved (lacking spontaneous symmetry breaking), all the two-point functions, except $\Delta_j^{(\bar{1}1)}$ and $\Delta_j^{(\bar{2}2)}$, are necessarily identical to zero dictated by the $U(2)$ or $U^2(1)$ symmetries (refer to Sec. 5). However, in a system with broken symmetries (like the ferromagnetic phase of the Ising model, as we will explore later), all the two-point functions have a chance to be non-zero. Using the standard MF scheme (ignoring the fluctuations) one approximates the interaction (four-Grassmann) term as follows

$$\bar{\chi}_j^{(1)} \chi_j^{(1)} \bar{\chi}_j^{(2)} \chi_j^{(2)} \approx \Delta_j^{(\bar{2}2)} \bar{\chi}_j^{(1)} \chi_j^{(1)} + \Delta_j^{(\bar{1}1)} \bar{\chi}_j^{(2)} \chi_j^{(2)} - \Delta_j^{(\bar{2}1)} \bar{\chi}_j^{(1)} \chi_j^{(2)} - \Delta_j^{(\bar{1}2)} \bar{\chi}_j^{(2)} \chi_j^{(1)} - \Delta_j^{(\bar{1}\bar{2})} \chi_j^{(1)} \chi_j^{(2)}$$
$$- \Delta_j^{(12)} \bar{\chi}_j^{(1)} \bar{\chi}_j^{(2)} - \Delta_j^{(\bar{1}1)} \Delta_j^{(\bar{2}2)} + \Delta_j^{(\bar{1}2)} \Delta_j^{(\bar{2}1)} + \Delta_j^{(\bar{1}\bar{2})} \Delta_j^{(12)}. \qquad (78)$$

The above approximation is tantamount to ignoring the fluctuations defined by

$$\Delta(j) \equiv \left\langle \bar{\chi}_j^{(1)} \chi_j^{(1)} \bar{\chi}_j^{(2)} \chi_j^{(2)} \right\rangle - \left\langle \bar{\chi}_j^{(1)} \chi_j^{(1)} \right\rangle \left\langle \bar{\chi}_j^{(2)} \chi_j^{(2)} \right\rangle$$
$$+ \left\langle \bar{\chi}_j^{(1)} \chi_j^{(2)} \right\rangle \left\langle \bar{\chi}_j^{(2)} \chi_j^{(1)} \right\rangle + \left\langle \bar{\chi}_j^{(1)} \bar{\chi}_j^{(2)} \right\rangle \left\langle \chi_j^{(1)} \chi_j^{(2)} \right\rangle, \qquad (79)$$

being identically zero for the MF theory. Under this approximation $\mathcal{S}_{\mathbf{A}}^{(2)}\{\bar{\chi}, \chi\}$ (Eq. 31) becomes quadratic as follows

$$\mathcal{S}_{\mathbf{A}}^{(2)}\{\bar{\chi}, \chi\} \approx S_{\mathrm{MF}}\{\bar{\chi}, \chi\} - \sum_{r=1}^{l} \Delta_4(j), \qquad (80)$$

where $\Delta_4(j) \equiv \Delta_j^{(\bar{1}1)} \Delta_j^{(\bar{2}2)} - \Delta_j^{(\bar{1}2)} \Delta_j^{(\bar{2}1)} - \Delta_j^{(\bar{1}\bar{2})} \Delta_j^{(12)}$, and

$$S_{\mathrm{MF}}\{\bar{\chi}, \chi\} \equiv \bar{\chi} \mathbb{O} \chi - \frac{1}{2} \chi \mathbb{I}_3 \chi - \frac{1}{2} \bar{\chi} \bar{\mathbb{I}}_3 \bar{\chi}, \qquad (81)$$

where

$$\mathbb{O} = \begin{pmatrix} \mathbf{A} + \tilde{\mathbf{I}}_{\bar{2}2} & -\tilde{\mathbf{I}}_{\bar{2}1} \\ -\tilde{\mathbf{I}}_{\bar{1}2} & \mathbf{A} + \tilde{\mathbf{I}}_{\bar{1}1} \end{pmatrix}, \qquad \mathbb{I}_3 = \begin{pmatrix} \mathbf{0} & \tilde{\mathbf{I}}_{\bar{1}\bar{2}} \\ -\tilde{\mathbf{I}}_{\bar{1}\bar{2}} & \mathbf{0} \end{pmatrix}, \qquad \bar{\mathbb{I}}_3 = \begin{pmatrix} \mathbf{0} & \tilde{\mathbf{I}}_{12} \\ -\tilde{\mathbf{I}}_{12} & \mathbf{0} \end{pmatrix}, \qquad (82)$$

and

$$\tilde{\mathbf{I}}_{\bar{2}2} = \mathrm{diag}\left\{ \Delta_j^{(\bar{2}2)} \right\}_{j=1}^{l}, \qquad \tilde{\mathbf{I}}_{\bar{1}1} = \mathrm{diag}\left\{ \Delta_j^{(\bar{1}1)} \right\}_{j=1}^{l},$$
$$\tilde{\mathbf{I}}_{\bar{2}1} = \mathrm{diag}\left\{ \Delta_j^{(\bar{2}1)} \right\}_{j=1}^{l}, \qquad \tilde{\mathbf{I}}_{\bar{1}2} = \mathrm{diag}\left\{ \Delta_j^{(\bar{1}2)} \right\}_{j=1}^{l}, \qquad (83)$$
$$\tilde{\mathbf{I}}_{\bar{1}\bar{2}} = \mathrm{diag}\left\{ \Delta_j^{(\bar{1}\bar{2})} \right\}_{j=1}^{l}, \qquad \tilde{\mathbf{I}}_{12} = \mathrm{diag}\left\{ \Delta_j^{(12)} \right\}_{j=1}^{l}.$$

The self-consistent two-point ($\Delta$) functions are expressible in terms of $S_{\mathrm{MF}}\{\bar{\chi}, \chi\}$ as follows:

$$\Delta_j^{(XY)} = \frac{\int_{\bar{\chi}, \chi} X_j Y_j e^{S_{\mathrm{MF}}\{\bar{\chi}, \chi\}}}{\int_{\bar{\chi}, \chi} e^{S_{\mathrm{MF}}\{\bar{\chi}, \chi\}}}, \qquad (84)$$

where $X_j, Y_j \equiv \bar{\chi}_j^{(1)}, \bar{\chi}_j^{(2)}, \chi_j^{(1)}$ and $\chi_j^{(2)}$, which are abbreviated by $\bar{1}, \bar{2}, 1$ and 2, respectively. Once all the two-point functions are obtained using the above equations, one can find $M^{(2)}$ by substituting them into Eq. 31 as follows:

$$M^{(2)}(\mathbf{A}) \approx M_{\text{MF}}^{(2)} \equiv e^{-\sum_{j=1}^{l} \Delta_4(j)} \int_{\bar{\chi},\chi} e^{S_{\text{MF}}\{\bar{\chi},\chi\}} . \tag{85}$$

Using the Pfaffian identities, one obtains

$$M_{\text{MF}}^{(2)} = e^{-\sum_{j=1}^{l} \Delta_4(j)} \text{Pf}\left[\mathbb{S}_{\text{MF}}^{\text{Pf}}(\mathbf{A})\right], \tag{86}$$

where Pf stands for Pfaffian (see Eq. D.19), and

$$\mathbb{S}_{\text{MF}}^{\text{Pf}}(\mathbf{A}) = \begin{bmatrix} \bar{\mathbb{I}}_3 & -\mathbb{O} \\ \mathbb{O}^T & \mathbb{I}_3 \end{bmatrix}. \tag{87}$$

In order to make the mean-field (MF) calculations feasible we adopt a symmetric case where we assume $\Delta_j^{(\bar{2}2)} = \Delta_j^{(\bar{1}1)}$ and $\Delta_j^{(\bar{2}1)} = \Delta_j^{(\bar{1}2)}$, and $\Delta_j^{(\bar{1}\bar{2})} = \Delta_j^{(12)}$ (expected for a system with SG and PH symmetries). Additionally, from now on we focus on *periodic systems*, in which all the $\Delta$ functions are anticipated to be independent of the coordinate $j$ owing to the system's translational invariance. Therefore, the independent self-consistent functions are $\Delta_j^{(\bar{1}1)} = \Delta_j^{(\bar{2}2)} \equiv \Delta_1$, and $\Delta_j^{(\bar{1}2)} = \Delta_j^{(\bar{2}1)} \equiv \Delta_2$, and $\Delta_j^{(\bar{1}\bar{2})} = \Delta_j^{(12)} \equiv \Delta_3$. Therefore, in this case we have

$$\Delta_4 \equiv \frac{1}{l} \sum_j \Delta_4(j) = \Delta_1^2 - \Delta_2^2 - \Delta_3^2, \tag{88}$$

and

$$S_{\text{MF}}\{\bar{\chi},\chi\} \equiv \bar{\chi} \mathbb{O}^s \chi - \Delta_3 \sum_{j=1}^{l} \left(\chi_j^{(1)} \chi_j^{(2)} + \bar{\chi}_j^{(1)} \bar{\chi}_j^{(2)}\right), \tag{89}$$

and

$$\mathbb{O}^s \equiv \mathbb{O}|_{\Delta_j=\Delta} = \begin{pmatrix} \mathbf{A} + \Delta_1 \mathbf{I} & -\Delta_2 \mathbf{I} \\ -\Delta_2 \mathbf{I} & \mathbf{A} + \Delta_1 \mathbf{I} \end{pmatrix}. \tag{90}$$

One can block-diagonalize $\mathbb{O}^s$ using a unitary transformation. For the details of this transformation, see Appendix D. The details of the calculations using Pfaffian formulation can be found in Appendix D.2. Specifically, the self-consistent two point functions are expressed in terms of the inverse matrix Eq. D.20. The result is the following set of self-consistent MF equations (see Eq. D.14)

$$\Delta_1 = \left[\frac{\mathbf{A}_s + \Delta_1 \mathbf{I}}{\mathbf{x} + 2\Delta_2 \mathbf{A}_a}\right]_{j,j},$$

$$\Delta_2 = \left[\frac{-\mathbf{A}_a + \Delta_2 \mathbf{I}}{\mathbf{x} + 2\Delta_2 \mathbf{A}_a}\right]_{j,j}, \tag{91}$$

$$\Delta_3 = \left[\frac{\Delta_3 \mathbf{I}}{\mathbf{x} + 2\Delta_2 \mathbf{A}_a}\right]_{j,j},$$

where

$$\mathbf{x} \equiv \left(\mathbf{A}^T + \Delta_1 \mathbf{I}\right)\left(\mathbf{A} + \Delta_1 \mathbf{I}\right) - \left(\Delta_2^2 + \Delta_3^2\right)\mathbf{I}$$
$$= \mathbf{A}\mathbf{A}^T + 2\Delta_1 \mathbf{A}_s + \Delta_4 \mathbf{I}, \tag{92}$$

and $\mathbf{A}_{s,a} \equiv \frac{1}{2}\left(\mathbf{A} \pm \mathbf{A}^T\right)$. The resulting $\Delta$-functions should be incorporated into Eq. 85 to find $M^{(2)}(\mathbf{A})$, leading to (see also Eq. D.3)

$$M_{\text{MF}}^{(2)}(\mathbf{A}) = e^{-l\Delta_4} \det \mathbb{S}_{\text{MF}}(\mathbf{A}), \tag{93}$$

where

$$\mathbb{S}_{\mathrm{MF}}(\mathbf{A}) = \begin{pmatrix} \mathbf{A} + (\Delta_1 - \Delta_2)\mathbf{I} & \Delta_3\mathbf{I} \\ -\Delta_3\mathbf{I} & -\mathbf{A}^T - (\Delta_1 + \Delta_2)\mathbf{I} \end{pmatrix}. \tag{94}$$

It is worth mentioning that one can obtain the same results using the Pfaffian to do the Berezin integrals of Grassmann variables, see Sec. D.2. Using Schur complement method for block matrices one can easily show that for even size matrices (see Eq. D.10)

$$\det \mathbb{S}_{\mathrm{MF}}(\mathbf{A}) = \det(\mathbf{x} + 2\Delta_2 \mathbf{A}_a). \tag{95}$$

Thanks to the circulant property of the matrices, the MF equations can be written in an analytical form (see [42] for a review on cirlulant matrices). We define $A_{q_k} \equiv \sum_n \mathbf{A}_{mn} e^{iq_k(n-m)}$ where $k \in \{1, 2, ..., l\}$, $q_k = \frac{2\pi}{l}(k + k_0)$, and $k_0 = 0$ $(\frac{1}{2})$ for circulant (anticirculant) matrices. We also define the Fourier component of $\mathbf{x}$ as

$$x(q_k) = (A_{-q_k} + \Delta_1)(A_{q_k} + \Delta_1) - \Delta_2^2 - \Delta_3^2. \tag{96}$$

Using these quantities, one converts Eqs. 91 to

$$\begin{aligned}
\Delta_1 &= \frac{1}{l} \sum_{k=1}^{l} \frac{A_s(q_k) + \Delta_1}{x(q_k) + 2\Delta_2 A_a(q_k)}, \\
\Delta_2 &= \frac{1}{l} \sum_{k=1}^{l} \frac{-A_a(q_k) + \Delta_2}{x(q_k) + 2\Delta_2 A_a(q_k)}, \\
\Delta_3 &= \frac{1}{l} \sum_{k=1}^{l} \frac{\Delta_3}{x(q_k) + 2\Delta_2 A_a(q_k)},
\end{aligned} \tag{97}$$

where $A_{s,a}(q_k) = \frac{1}{2}\left(A_{q_k} \pm A_{-q_k}\right)$. A closed formulae is obtained for SPPM in the continuum limit using the Euler-Maclaurin formula for $l \to \infty$. In this limit, the $\Delta$ functions, which are in general $l$-dependent tend to a thermodynamic asymptotic limit. Supposing that these functions converge fast enough to this asymptotic thermodynamic limit, and for the case $x(q) + 2\Delta_2 A_a(q)$ is regular (non-zero, and finite and free of discontinuities) for all $q$s, we obtain the following expression for SPPM using Eqs. 93 and Eq. 95:

$$\begin{aligned}
\ln M_{\mathrm{MF}}^{(2)}(\mathbf{A}) &= \ln \det \mathbb{S}_{\mathrm{MF}} - l\Delta_4 \\
&= \sum_{k=1}^{l} \ln\left[x(q_k) + 2\Delta_2 A_a(q_k)\right] - l\Delta_4 \\
&\to \frac{l}{2\pi} \int_{-\pi}^{\pi} dq \ln\left[x(q) + 2\Delta_2 A_a(q)\right] - l\Delta_4 + \ln\left[x(0) + 2\Delta_2 A_a(0)\right] + O(l^{-1}).
\end{aligned} \tag{98}$$

Note that the last relation of Eq. 97 for non-zero $\Delta_3$ means

$$\frac{\partial}{\partial \Delta_4} M_{\mathrm{MF}}^{(2)}(\mathbf{A}) = 0, \tag{99}$$

i.e. this function is stationary with respect to $\Delta_4$. Another important property of the MF theory for antisymmetric matrices is concerning the case $\Delta_2 = 0$ and $\Delta_3 \neq 0$. One can easily see that the only equation to be satisfied is (see Eq. 91)

$$\frac{1}{l} \sum_{k=1}^{l} x(q_k)^{-1} = 1, \tag{100}$$

from which we see that everything (as well as $M^{(2)}(\mathbf{A})$) is a sole function of $\Delta_4$ (note that for antisymmetric matrices $A_{-q} = -A_q$ which means $x_q = -A_q^2 + \Delta_4$). Using the antisymmetric property of $\mathbf{A}$, one can easily show that $\Delta_2 = 0$ is always a solution.

## 6.1 Scaling properties of the MF equations

In this section we explore various scaling properties of the MF solution. It should be noted that $\Delta_1 = \Delta_2 = \Delta_3 = 0$ is a trivial solution for antisymmetric matrices. However, for non-trivial solutions, we investigate the third relation in Eq. 97 in the scaling limit when $\Delta_3 \neq 0$. We consider the continuum limit $l \to \infty$ of the consistency relations, having in mind that for SPPM $l$ is finite. In the large $l$ limit, normally $\Delta$ functions converge exponentially fast to their asymptotic thermodynamic limit, so that Euler-Maclaurin integrals give reliable approximations for $\Delta$ functions for finite $l$, while the convergence is expected to be slow (power-law) in the critical regions. In the continuum limit (to order $l^{-1}$) we have

$$\int_{-\pi}^{\pi} \frac{dq}{2\pi} \frac{1}{x(q) + 2\Delta_2 A_a(q)} = 1. \tag{101}$$

For antisymmetric matrices we have $A(-q) = -A(q)$, this equation for the case $\Delta_2 = 0$ becomes

$$\int_{-\pi}^{\pi} \frac{dq}{2\pi} \frac{1}{\Delta_4 - A(q)^2} = 1. \tag{102}$$

Solving the above equation in general is a challenge. Nevertheless, its scaling properties provide valuable insights into the conditions that lead to non-zero values of $\Delta_4$, as well as how these quantities scale with the system size and external parameters. This can be done through the analysis of the zeros and singular points of $A(q)$. Denoting the list of singular points of $\mathbf{A}$ as $\{q_i\}$, we have $A(q)|_{q \to q_i} \propto (q - q_i)^{\tau_i}$, where $i$ varies over the zeros and singular points of $A(q)$ and $\tau_i$ is its corresponding exponent. Since $A(q)$ is antisymmetric one can easily see that $-q_i$ should also be a root or a singular point. Note that $\Delta_4 = 0$, although is a trivial solution of Eq. 97, it is not a solution of Eq. 102. The onset of spontaneous symmetry breaking is given by the condition that $\Delta_4$ is nonzero, where the main contribution in the integral comes from the singular points, around which $A(q)$ shows scaling behaviors (for a system in the vicinity of the critical point, the main contribution is expected to come from this singular/zero point). Suppose that this singular/zero point is zero, so that $A(q) = a_0^{-1} q^{\tau}$ for small $q$s. Here $a_0^{-1}$ is a proportionality parameter, which depends on the external parameters in the system (for the Ising model, $a_0 = 1 - h$, where $h$ is the magnetic field, see Sec. 8). To satisfy Eq. 102, the integral should not blow up in the vicinity of the singular points, when $\Delta_{1,3}$ are close to zero (the transition point), i.e.

$$\int_{q \sim 0} \frac{dq}{\Delta_4 - a_0^{-2} q^{2\tau}} < \infty. \tag{103}$$

Given the fact that the main contribution to the integrals comes from the singular point, we extend the integral to the whole range $q \in [-\pi, \pi]$. Doing so, one can show that Eq. 103 results in

$$\Delta_4 - \Delta_4(a_0 = 0) \propto a_0^{\zeta}, \qquad \zeta \equiv \frac{2}{2\tau - 1}. \tag{104}$$

In the case where $A(q)$ is symmetric, the outcome exhibits significant differences. When we impose the condition $\Delta_3 \neq 0$, the following equations should be satisfied:

$$\int_{-\pi}^{\pi} \frac{dq}{2\pi} \frac{1}{(A(q) + \Delta_1)^2 - \Delta_3^2} = 1, \tag{105a}$$

$$\int_{-\pi}^{\pi} \frac{dq}{2\pi} \frac{A(q)}{(A(q) + \Delta_1)^2 - \Delta_3^2} = 0. \tag{105b}$$

Table 2: Various passibilities for the MF equations and the possible solutions. (a) and (b) show the results for antisymmetric matrices, while (c) and (d) show the results for symmetric matrices.

| $\mathbf{A} = \mathbf{A}_a$ , $\Delta_1$ or $\Delta_3$ or both $\neq 0$ | $\mathbf{A} = \mathbf{A}_a$ , $\Delta_1 = \Delta_3 = 0$ |
|---|---|
| $\displaystyle\int_{-\pi}^{\pi} \frac{dq}{2\pi} \frac{A(q)}{x(q) + 2\Delta_2 A(q)} = 0$ (a) | $\Delta_2 = \displaystyle\int_{-\pi}^{\pi} \frac{dq}{2\pi} \frac{-A(q) + \Delta_2}{x(q) + 2\Delta_2 A(q)}$ (b) |
| $\Rightarrow \boxed{\Delta_2 = 0}$ , $\displaystyle\int_{-\pi}^{\pi} \frac{dq}{2\pi} \frac{1}{x(q)} = 1$ or a more complicated solution | $\Rightarrow \boxed{\Delta_2 = 0}$ or a more complicated solution |
| $\mathbf{A} = \mathbf{A}_s$ , $\Delta_2$ or $\Delta_3$ or both $\neq 0$ | $\mathbf{A} = \mathbf{A}_s$ , $\Delta_2 = \Delta_3 = 0$ |
| $\displaystyle\int_{-\pi}^{\pi} \frac{dq}{2\pi} \frac{A(q)}{x(q)} = 0$ $\displaystyle\int_{-\pi}^{\pi} \frac{dq}{2\pi} \frac{1}{x(q)} = 1$ (c) | $\Delta_1 = \displaystyle\int_{-\pi}^{\pi} \frac{dq}{2\pi} \frac{1}{A(q) + \Delta_1}$ (d) |

Table 3: In this table we show the consequences of different symmetries on the two point functions for $n = 2$. The last column is for antisymmetric matrices. Each cell of the table has been identified by two values: 0 when the two point function is zero and $-$ when the symmetry has no effect.

| | SG | $U(2)$ | $SU^2(2)$ | $U^2(1)$ | $U(1)$ | SP | Ax. $U(1)$ | PH | CPH |
|---|---|---|---|---|---|---|---|---|---|
| $\Delta_1$ | $-$ | $-$ | $-$ | $-$ | $-$ | $-$ | $-$ | 0 | $-$ |
| $\Delta_2$ | $-$ | 0 | 0 | 0 | $-$ | 0 | 0 | $-$ | $-$ |
| $\Delta_3$ | 0 | 0 | 0 | 0 | 0 | 0 | $-$ | $-$ | $-$ |

For the case $\Delta_3 = 0$ the only equation to be satisfied is the first relation in Eq. 97, i.e.

$$\Delta_1 = \int_{-\pi}^{\pi} \frac{dq}{2\pi} \frac{1}{A(q) + \Delta_1} , \tag{106}$$

for which the same arguments as the antisymmetric case applies. This serves as an evidence that the two point function $\Delta_3$ should be zero for the case $A(q)$ is symmetric. In Table. 2 we summarize various possibilities regarding the MF solutions for symmetric and antisymmetric matrices.

## 6.2 Spontaneous symmetry breaking

This subsection gives some information concerning the two point functions based on the symmetry arguments explored in Sec. 5. For a system with $U(2)$ and $U^2(1)$ symmetry $\Delta_2 = \Delta_3 = 0$ as we already mentioned in Sec. 5, while $U(1)$ as a subgroup of $U(2)$ and $U^2(1)$ forces only $\Delta_3$ to be zero. Table 3 shows some relations obtained by symmetries for two point functions. For antisymmetric matrices $\Delta_1$ is zero because of PH symmetry. As a well-known scenario, the above symmetries can be *spontaneously broken* by changing the external parameters of the system. The TFI chain is an example where some symmetries are spontaneously broken below a critical magnetic field (see the following section). Table 3 helps to characterize various phases

Table 4: Four possibilities for $(\Delta_2, \Delta_3)$ versus the symmetries of the system for a general matrix. The first column shows the value of the pair $(\Delta_2, \Delta_3)$, and "0" ($\emptyset$) shows that the corresponding two-point function is zero (non-zero). Five different symmetries are considered. $\checkmark$ shows that the symmetry holds, and $\times$ shows that the symmetry is broken. The symbol $-$ shows that the corresponding symmetry can be either $\checkmark$ or $\times$.

| $(\Delta_2, \Delta_3)$ | SG | $U(2)$ | $U^2(1)$ | $U(1)$ | SP | Ax. $U(1)$ |
|---|---|---|---|---|---|---|
| $G_1$: $(0, 0)$ | $\checkmark$ | $\checkmark$ | $\checkmark$ | $\checkmark$ | $\checkmark$ | $\checkmark$ |
| $G_2$: $(\emptyset, 0)$ | $\checkmark$ | $\times$ | $\times$ | $-$ | $\times$ | $\times$ |
| | $-$ | $\times$ | $\times$ | $\checkmark$ | $\times$ | $\times$ |
| $G_3$: $(0, \emptyset)$ | $\times$ | $\times$ | $\times$ | $\times$ | $\times$ | $\checkmark$ |
| $G_4$: $(\emptyset, \emptyset)$ | $\times$ | $\times$ | $\times$ | $\times$ | $\times$ | $\times$ |

Table 5: Four possibilities for $(\Delta_2, \Delta_3)$ in terms of symmetries of the symmetric matrices ($n = 2$). The symbols are like the Table 4.

| $(\Delta_2, \Delta_3)$ | SG | $U(2)$ | $SU(2)$ | $U^2(1)$ | $U(1)$ | SP | Ax $U(1)$ |
|---|---|---|---|---|---|---|---|
| $S_1$: $(0, 0)$ | $\checkmark$ | $\checkmark$ | $\checkmark$ | $\checkmark$ | $\checkmark$ | $\checkmark$ | $\checkmark$ |
| $S_2$: $(\emptyset, 0)$ | $\checkmark$ | $\times$ | $\times$ | $\times$ | $-$ | $\times$ | $\times$ |
| | $-$ | $\times$ | $\times$ | $\times$ | $\checkmark$ | $\times$ | $\times$ |
| $S_3$: $(0, \emptyset)$ | $\times$ | $\times$ | $\times$ | $\times$ | $\times$ | $\times$ | $\checkmark$ |
| $S_4$: $(\emptyset, \emptyset)$ | $\times$ | $\times$ | $\times$ | $\times$ | $\times$ | $\times$ | $\times$ |

resulting from this spontaneous symmetry breaking. In general one or more symmetries of the system can break, leading some two-point functions of the system to be nonzero. Four different cases are shown for general matrices in terms of the symmetries in Table 4 (different regimes are shown by $G_i$, $i = 1, ..., 4$). In a system with full symmetry ($G_1$) both $\Delta_2$ and $\Delta_3$ are zero, while in the opposite case ($G_4$) both of them can be non-zero.

Four different cases associated with $\Delta_2$ and $\Delta_3$ for the symmetric matrices is shown in Table. 5, where the states are labeled by $S_i$, $i = 1, ..., 4$. In this case we have additionally $SU(2)$ symmetry which should be necessarily broken for having non-zero $\Delta_2$ or $\Delta_3$. The discrete Laplacian is an important example for this case, however, it does not show any broken symmetry.

For the anti-symmetric matrices, the effect of the symmetries are even stronger, where restrictions on $\Delta_1$ is set by the symmetries. Therefore, one can decide about *all* the two-point functions in this case, which additionally have PH symmetry. The details are shown in Table 6. A system with full symmetry (the case $A_1$) results in *all* two-point functions to be identically zero. In the opposite extreme case (*all* the symmetries are spontaneously broken, i.e. the case $A_8$), the two-point functions can be non-zero. The intermediate cases are explored in the table.

## 6.3 MF equations in the dual representation

In this section we develop a MF theory for the dual representation Eq. 38. The details of the calculations are presented in Appendix D.3. Analyses presented in the previous section are applicable for $M_u^{(2)}(\mathbf{A})$ with some changes. The MF equations are written in terms of $\mathbf{N}$ given by Eqs. 40 and 42, where $u$ is the duality parameter. More precisely, with a simple re-definition

Table 6: Eight possibilities for $(\Delta_1, \Delta_2, \Delta_3)$ in terms of the symmetries of the anti-symmetric matrices ($n = 2$). The symbols are like the Table 4.

| $(\Delta_1, \Delta_2, \Delta_3)$ | SG | $U(2)$ | $U^2(1)$ | $U(1)$ | SP | Ax $U(1)$ | PH |
|---|---|---|---|---|---|---|---|
| $A_1$: $(0,0,0)$ | ✓ | ✓ | ✓ | ✓ | ✓ | ✓ | ✓ |
| $A_2$: $(\emptyset,0,0)$ | ✓ | ✓ | ✓ | ✓ | ✓ | ✓ | × |
| $A_3$: $(0,\emptyset,0)$ | ✓ | × | × | — | × | × | ✓ |
| | — | × | × | ✓ | × | × | ✓ |
| $A_4$: $(0,0,\emptyset)$ | × | × | × | × | × | ✓ | ✓ |
| $A_5$: $(\emptyset,\emptyset,0)$ | ✓ | × | × | — | × | × | × |
| | — | × | × | ✓ | × | × | × |
| $A_6$: $(0,\emptyset,\emptyset)$ | × | × | × | × | × | × | ✓ |
| $A_7$: $(\emptyset,0,\emptyset)$ | × | × | × | × | × | ✓ | × |
| $A_8$: $(\emptyset,\emptyset,\emptyset)$ | × | × | × | × | × | × | × |

$\Delta_i' \equiv u\Delta_i$ ($i = 1, 2, 3$), and $\Delta_4' \equiv \frac{1}{u}\left[\Delta_1'^2 - \Delta_2'^2 - \Delta_3'^2\right]$, one can obtain similar equations as the direct MF equations 91. Mainly, the expansion given in Eq. D.32 is obtained, resulting to the following self-consistent equations:

$$
\begin{aligned}
\Delta_1' &= u\left[\frac{\mathbf{N}_s + \Delta_1'\mathbf{I}}{\mathbf{x}^{(\mathbf{N})} + 2\Delta_2'\mathbf{N}_a}\right]_{j,j}, \\
\Delta_2' &= u\left[\frac{-\mathbf{N}_a + \Delta_2'\mathbf{I}}{\mathbf{x}^{(\mathbf{N})} + 2\Delta_2'\mathbf{N}_a}\right]_{j,j}, \\
\Delta_3' &= u\left[\frac{\Delta_3'\mathbf{I}}{\mathbf{x}^{(\mathbf{N})} + 2\Delta_2'\mathbf{N}_a}\right]_{j,j},
\end{aligned}
\tag{107}
$$

where $\mathbf{x}^{(\mathbf{N})}$ is given by (see Eq. D.34)

$$
\mathbf{x}^{(N)} = \mathbf{N}\mathbf{N}^T + 2\Delta_1'\mathbf{N}_s + u\Delta_4',
\tag{108}
$$

and $\mathbf{N}_{a,s} \equiv \frac{1}{2}\left(\mathbf{N} \pm \mathbf{N}^T\right)$. One then finds

$$
M^{(2)}(\mathbf{A}) = c_l e^{-l\Delta_4'} \det \mathbb{S}_{\mathrm{MF}}^u(\mathbf{N}),
\tag{109}
$$

where $\mathbb{S}_{\mathrm{MF}}^u(\mathbf{N})$ and $c_l$ are given in Eqs. D.36 and B.14, respectively. One can employ the Euler-Maclaurin formula to cast this equation into the following form (for even $l$, see Eq. D.38 for the details)

$$
\begin{aligned}
\ln M_{u,\mathrm{MF}}^{(2)}(\mathbf{A}) + l\Delta_4' &= \sum_{k=1}^{l} \ln\left[\frac{A_{q_k}'A_{-q_k}'}{u + m^2}\left(x^{(N)}(q_k) + 2\Delta_2'N_a(q_k)\right)\right] \\
&\to \frac{l}{2\pi}\int_{-\pi}^{\pi} dq \ln\left[\frac{A_q'A_{-q}'}{u + m^2}\left(x^{(N)}(q) + 2\Delta_2'N_a(q)\right)\right] + O(1),
\end{aligned}
\tag{110}
$$

where

$$
x^{(N)}(q) \equiv (N_q + \Delta_1')(N_{-q} + \Delta_1') - \Delta_2'^2 - \Delta_3'^2,
\tag{111}
$$

is the Fourier component of one row of $\mathbf{x}^{(N)}$, and $N_q \equiv m - \frac{1}{A_q'}$ is the Fourier component of $\mathbf{N}$, $A_q' \equiv A_q + \frac{m}{u+m^2}$.

## 6.4 Saddle point approximation

In this section we present an alternative way which serves as a MF scheme with an ability to go beyond the lowest order. This method which is based on the saddle point (SP) approximation, enables us to test the efficiency of the MF results by tracking how the MF results vary as the order of saddle point expansion changes. We consider the case $\Delta_j^{(1)} = \Delta_j^{(\bar{1}1)} = \Delta_j^{(\bar{2}2)}$, and $\Delta_j^{(2)} \equiv \Delta_j^{(\bar{1}2)} = \Delta_j^{(\bar{2}1)}$, and $\Delta_j^{(3)} = \Delta_j^{(\bar{1}\bar{2})} = \Delta_j^{(12)}$ ($j \in [1,l]$ is the space index). We define $y_j \equiv (y_j^{(1)}, y_j^{(2)}, y_j^{(3)})^T$, where $y_j^{(1)} \equiv \Delta_j^{(1)}$, $y_j^{(2)} \equiv i\Delta_j^{(2)}$, $y_j^{(3)} \equiv i\Delta_j^{(3)}$. This enables us to use the notation

$$
\begin{aligned}
y_j^T y_j &= \left(y_j^{(1)}\right)^2 + \left(y_j^{(2)}\right)^2 + \left(y_j^{(3)}\right)^2 \\
&= \left(\Delta_j^{(1)}\right)^2 - \left(\Delta_j^{(2)}\right)^2 - \left(\Delta_j^{(3)}\right)^2 \equiv \Delta_4(j).
\end{aligned}
\tag{112}
$$

We define also the corresponding three-currents $J_j^\chi \equiv (J_j^{(1)}, J_j^{(2)}, J_j^{(3)})$ as follows

$$
J_j^{(1)} = \bar{\chi}_j^{(1)} \chi_j^{(1)} + \bar{\chi}_j^{(2)} \chi_j^{(2)},
\tag{113a}
$$

$$
J_j^{(2)} = i\left(\bar{\chi}_j^{(1)} \chi_j^{(2)} + \bar{\chi}_j^{(2)} \chi_j^{(1)}\right),
\tag{113b}
$$

$$
J_j^{(3)} = i\left(\bar{\chi}_j^{(1)} \bar{\chi}_j^{(2)} + \chi_j^{(1)} \chi_j^{(2)}\right),
\tag{113c}
$$

so that $\left\langle J_j^{(k)} \right\rangle = 2y_j^{(k)}$, $k = 1, 2, 3$. We then define the vectors $y \equiv (y_1, ..., y_l)$ and $J_\chi \equiv (J_1^\chi, ..., J_l^\chi)$. Using these vectors we define the action

$$
\begin{aligned}
\mathfrak{L}(\bar{\chi}, \chi, y, J_\chi) &\equiv \bar{\chi} \mathbb{A} \chi - \frac{1}{2} y^T \mathcal{B} y + y^T J_\chi \\
&= \bar{\chi} \mathbb{A} \chi - \sum_j \left[\left(\Delta_j^{(1)}\right)^2 - \left(\Delta_j^{(2)}\right)^2 - \left(\Delta_j^{(3)}\right)^2\right] \\
&\quad + \sum_j \Delta_j^{(1)}(\bar{\chi}_j^{(1)} \chi_j^{(1)} + \bar{\chi}_j^{(2)} \chi_j^{(2)}) \\
&\quad - \sum_j \Delta_j^{(2)} \left(\bar{\chi}_j^{(1)} \chi_j^{(2)} + \bar{\chi}_j^{(2)} \chi_j^{(1)}\right) \\
&\quad - \sum_j \Delta_j^{(3)}(\bar{\chi}_j^{(1)} \bar{\chi}_j^{(2)} + \chi_j^{(1)} \chi_j^{(2)}),
\end{aligned}
\tag{114}
$$

and $\mathcal{B} = 2\mathbb{I}_{2l}$. This action corresponds to the effective actions 80 and 89 in the MF analysis. In the case

$$
\Delta_j^{(2)} = 0,
\tag{115}
$$

we omit $J_j^{(2)}$, so that $y_j \to (y_j^{(1)}, y_j^{(3)})$, and $J_j \to (J_j^{(1)}, J_j^{(3)})$. To set up the problem for the SP approximation, we first need to insert auxiliary variables in the MF action, so that the integration over these variables gives the original partition function. More precisely, the exponential of the interaction term is written in terms of an integration over a new quadratic term with additional auxiliary variable, which plays the role of the self-consistent correlation. Now we use the following identity

$$
\int \mathcal{D}\{y\} \exp\left[-\frac{1}{2} y^T \mathcal{B} y + y^T J_\chi\right] = [\det \mathcal{B}]^{-\frac{1}{2}} \exp\left[\frac{1}{2} J_\chi^T \mathcal{B}^{-1} J_\chi\right],
\tag{116}
$$

with $\mathcal{D}\{y\} \equiv \prod_k \prod_j \frac{dy_j^{(k)}}{(2\pi)^{\frac{3}{2}}}$ and also

$$
\frac{1}{2} J_\chi^T \mathcal{B}^{-1} J_\chi = \sum_j \bar{\chi}_j^{(1)} \chi_j^{(1)} \bar{\chi}_j^{(2)} \chi_j^{(2)},
\tag{117}
$$

(which is the interaction term in Eq. 31). Using this identity, one can easily prove that

$$M^{(2)}(\mathbf{A}) \equiv e^{-F^{(2)}(\mathbf{A})} = [\det \mathcal{B}]^{\frac{1}{2}} \int \mathcal{D}\{y\}\, \mathcal{M}^{(2)}(\mathbf{A}, y), \tag{118}$$

where

$$\mathcal{M}^{(2)}(\mathbf{A}, y) \equiv e^{-\mathcal{F}^{(2)}(\mathbf{A}, y)} \equiv \int_{\bar{\chi}\chi} e^{\mathfrak{L}(\bar{\chi}, \chi, y, J_{\chi})}. \tag{119}$$

Using the properties of the Berezin integrals, we obtain

$$\mathcal{M}^{(2)}(\mathbf{A}, y) = e^{-\sum_{j=1}^{l} \Delta_4(j)} \mathrm{Pf}\left[\mathbb{S}^{\mathrm{Pf}}(\mathbf{A})\right], \tag{120}$$

where

$$\mathbb{S}^{\mathrm{Pf}}(\mathbf{A}) = \begin{bmatrix} \bar{\mathbb{I}}_3 & -\mathbb{O} \\ \mathbb{O}^T & \mathbb{I}_3 \end{bmatrix}, \tag{121}$$

and $\mathbb{O}$, $\bar{\mathbb{I}}_3$, and $\mathbb{I}_3$ are given in Eq. 82, this time for free parameters $\Delta_1, \Delta_3$. The main contribution of the integrand of Eq. 118 comes from the stationary point, i.e. by minimizing $\mathcal{F}^{(2)}(\mathbf{A}, y)$ with respect to $y_j^{(k)}$s ($k = 1, 2, 3$ and $j \in [1, l]$):

$$\partial_{y_j^{(k)}} \mathcal{F}^{(2)}(\mathbf{A}, y)\Big|_{y=\bar{y}} = 0, \tag{122}$$

or equivalently,

$$\partial_{y_j^{(k)}} \ln\left[e^{\frac{1}{2}y^T \mathcal{B} y} \mathcal{M}^{(2)}(\mathbf{A}, y)\right]\Big|_{y=\bar{y}} = \sum_{k'} \mathcal{B}_{kk'} \bar{y}_j^{(k')}, \tag{123}$$

which is our original MF self-consistent relations Eqs. 84. Here we see explicitly that the stationary approximation of the bosonic representation of SPPM corresponds to the MF approximation. The idea is to take a further step from the stationary approximation, and expand $\mathcal{F}^{(2)}(\mathbf{A}, y)$ to the second order. To the second order, defining $F_{\mathrm{MF}}^{(2)}(\mathbf{A}) \equiv \mathcal{F}^{(2)}(\mathbf{A}, \bar{y})$ as the MF approximation of $\mathcal{F}^{(2)}$ we have (note that the first derivatives are zero by definition)

$$\mathcal{F}_{\mathrm{SP}}^{(2)}(\mathbf{A}, y) \equiv F_{\mathrm{MF}}^{(2)}(\mathbf{A}) + \frac{1}{2} \sum_j \sum_{k,k'} \left(y_j^{(k)} - \bar{y}_j^{(k)}\right) \beta_{kk'}^{(j)} \left(y_j^{(k')} - \bar{y}_j^{(k')}\right), \tag{124}$$

where $\beta_{kk'}^{(j)} \equiv \partial_{y_j^{(k)}} \partial_{y_j^{(k')}} \mathcal{F}^{(2)}(\mathbf{A}, y)|_{y=\bar{y}}$. In situations where $\mathcal{F}^{(2)}$ has a single minimum, this approximation can be applied across a significant range of variables, and it can be inserted into Eq. 118 for a whole range of integration. In more intricate situations, attention must be paid to the influence of different minima of $\mathcal{F}$; for a more detailed review, refer to Appendix J. Then integrating over $y$ variable gives

$$M_{\mathrm{SP}}^{(2)}(\mathbf{A}) \approx \frac{e^{-F_{\mathrm{MF}}^{(2)}(\mathbf{A})}}{\sqrt{\det \mathcal{C}}}, \tag{125}$$

where $\mathcal{C} \equiv \mathcal{B}^{-1}\beta$. Note that $\beta$ and $\mathcal{C}$ are block diagonal matrices. Representing the elements of $\mathcal{C}$ by $c_{kk'}^{(j)}$, using Eq. 119 we find

$$c_{kk'}^{(j)} = \delta_{kk'} - h_{kk'}^{(j)}, \tag{126}$$

where

$$h_{kk'}^{(j)} = \frac{1}{2}\left[\left\langle J_j^{(k)} J_j^{(k')}\right\rangle - \left\langle J_j^{(k)}\right\rangle \left\langle J_j^{(k')}\right\rangle\right], \tag{127}$$

is the correlation function of the currents. Now we exploit the fact that in the MF approximation $\left\langle \bar{\chi}_j^{(1)} \chi_j^{(1)} \bar{\chi}_j^{(2)} \chi_j^{(2)} \right\rangle \approx \left( \Delta_j^{(1)} \right)^2 - \left( \Delta_j^{(3)} \right)^2$, which results in

$$-h_{11}^{(j)} = h_{22}^{(j)} = \left( \Delta_j^{(1)} \right)^2 + \left( \Delta_j^{(3)} \right)^2 , \tag{128a}$$

$$h_{12}^{(j)} = h_{21}^{(j)} = 2i\Delta_j^{(1)} \Delta_j^{(3)} . \tag{128b}$$

Now if we consider the periodic system, so that $\Delta_j^{(1)} = \Delta_1$ and $\Delta_j^{(3)} = \Delta_3$ for all $j$ values, we obtain

$$M_{\text{SP}}^{(2)}(\mathbf{A}) \approx \frac{e^{-F_{\text{MF}}^{(2)}(\mathbf{A})}}{(1-(\Delta_1^2-\Delta_3^2)^2)^{\frac{l}{2}}} . \tag{129}$$

This equation can be written in terms of free energy density $f_{\text{SP}} \equiv F_{\text{SP}}/l$, and $f_{\text{MF}} \equiv F_{\text{MF}}/l$ ($F_{\text{SP}} \equiv -\ln M_{\text{SP}}^{(2)}(\mathbf{A})$) as follows:

$$f_{\text{SP}}/f_{\text{MF}} = 1 + \frac{1}{2f_{\text{MF}}} \ln\left[ 1 - (\Delta_1^2 - \Delta_3^2)^2 \right] . \tag{130}$$

This equation shows that when the second term on the right hand side is small enough, the saddle point approximation is close to the mean field result. This is a signature of the stability of the mean field results.

In the dual space, the MF equations are similar to the ordinary one, except we just need to substitute $\Delta_1$ and $\Delta_3$ by $\Delta_1' = u\Delta_1$ and $\Delta_3' = u\Delta_3$, and $\mathbf{A}$ by $\mathbf{N}$. The result is (for the details see Appendix H)

$$f_{\text{SP}}^u/f_{\text{MF}}^u = 1 + \frac{1}{2f_{\text{MF}}^u} \ln\left[ 1 - \frac{1}{u^2}\left( \Delta_1'^{\,2} - \Delta_3'^{\,2} \right)^2 \right] , \tag{131}$$

where $f_{\text{SP}}^u$ and $f_{\text{MF}}^u$ correspond to $f_{\text{SP}}$ and $f_{\text{MF}}$ respectively in the dual space. For the case $\frac{1}{u^2}\left( \Delta_1'^{\,2} - \Delta_3'^{\,2} \right)^2 \to 0$ one finds $f_{\text{SP}}^u \to f_{\text{MF}}^u$. As a general result we see that the lower the second term, the more stable the MF solutions. More precisely, the stable MF results correspond to the situation where the argument of the logarithm is close to one so that $1 - \frac{1}{u^2}\left( \Delta_1'^{\,2} - \Delta_3'^{\,2} \right)^2 \ll \exp\left[ f_{\text{MF}}^u \right]$.

## 6.5 Variational method: A lower bound

In this section we present our third MF scheme based on a variational method. In addtion to giving us the MF equations, this method provides a lower bound for the SPPM. This enables us to design other forms of trial representations for SPPM and to obtain other approximations. We start with Eq. 31 (with the action $\mathcal{S}_{\text{A}}^{(2)}\{\bar{\chi}, \chi\}$) and show that it is higher than any trial value obtained from a trial action $\mathcal{S}_{\text{trial}}\{y, \bar{\chi}, \chi\}$:

$$\mathcal{M}_{\text{trial}}(y) = \int_{\bar{\chi}\chi} \exp\left[ \mathcal{S}_{\text{trial}}\{y, \bar{\chi}, \chi\} \right] , \tag{132}$$

where $y \equiv \{y_1, ..., y_s\}$ is the set of parameters in the trial action. To prove this, we define

$$\langle ... \rangle_{\text{trial}} \equiv \frac{\int_{\bar{\chi},\chi}(...)e^{\mathcal{S}_{\text{trial}}\{y,\bar{\chi},\chi\}}}{\mathcal{M}_{\text{trial}}(y)} , \tag{133}$$

where $\langle ... \rangle_{\text{trial}}$ shows the average of any function of $\bar{\chi}$ and $\chi$ with respect to the trial action. One can retrieve the original SPPM using the identity

$$M^{(2)}(\mathbf{A}) = \mathcal{M}_{\text{trial}}(y) \left\langle e^{\mathcal{S}_{\text{A}}^{(2)}\{\bar{\chi},\chi\} - \mathcal{S}_{\text{trial}}\{y,\bar{\chi},\chi\}} \right\rangle_{\text{trial}} . \tag{134}$$

The main idea is to use the Jensen's inequality (see Appendix I)

$$\left\langle e^{\mathcal{S}_{\mathrm{A}}^{(2)}\{\bar{\chi},\chi\} - \mathcal{S}_{\mathrm{trial}}\{y,\bar{\chi},\chi\}} \right\rangle_{\mathrm{trial}} \geq e^{\left\langle \mathcal{S}_{\mathrm{A}}^{(2)}\{\bar{\chi},\chi\} - \mathcal{S}_{\mathrm{trial}}\{y,\bar{\chi},\chi\} \right\rangle_{\mathrm{trial}}}, \tag{135}$$

which leads to

$$\tilde{\mathcal{M}}_{\mathrm{trial}}(y) \leq M^{(2)}(\mathbf{A}), \tag{136}$$

where

$$\tilde{\mathcal{M}}_{\mathrm{trial}}(y) \equiv \mathcal{M}_{\mathrm{trial}}(y) e^{\left\langle \mathcal{S}_{\mathrm{A}}^{(2)}\{\bar{\chi},\chi\} - \mathcal{S}_{\mathrm{trial}}\{y,\bar{\chi},\chi\} \right\rangle_{\mathrm{trial}}}. \tag{137}$$

This identity gives a lower bound for SPPM, and can be used to find an approximation by maximizing $\tilde{\mathcal{M}}_{\mathrm{trial}}(y)$ with respect to all $y_i$'s, i.e.

$$\frac{\partial \tilde{\mathcal{M}}_{\mathrm{trial}}(y)}{\partial y_i} = 0, \ i = 1, ..., s. \tag{138}$$

This equation, along with Eq. 136 are the main equations of the variational method for estimating the value of $M^{(2)}(\mathbf{A})$. To make connection with the MF theory, we set

$$\mathcal{S}_{\mathrm{trial}}\{y,\bar{\chi},\chi\} \rightarrow \mathfrak{L}(\bar{\chi},\chi,y,J_\chi), \tag{139}$$

where $y = \{\Delta_1, \Delta_2, \Delta_3\}$, and $\mathfrak{L}(\bar{\chi},\chi,y,J_\chi)$ is given in Eq. 114 and corresponds to $S_{\mathrm{MF}}\{\bar{\chi},\chi\} - \sum_{j=1}^{l} \Delta_4(j)$ in the MF equations. We define $\mathcal{M}(\mathbf{A},y)$ according to Eq. 132 which is equivalent to Eq. 119. Then, using the fact

$$\left\langle \mathcal{S}_{\mathrm{A}}^{(2)}\{\bar{\chi},\chi\} - S_{\mathrm{MF}}\{\bar{\chi},\chi\} \right\rangle_{\mathrm{MF}} = -\sum_{j=1}^{l} \Delta_4(j), \tag{140}$$

and using Eq. 137, we find that $\tilde{\mathcal{M}}(\mathbf{A},y) = \mathcal{M}(\mathbf{A},y)$. By maximizing $\mathcal{M}(\mathbf{A},y)$ with respect to $y_i$'s (Eq. 138) one obtains the governing equations on $\Delta$ functions, which are the original MF equations 97 and 122. To summarize, we find the following identity

$$M^{(2)}(\mathbf{A}) \geq \mathcal{M}(\mathbf{A},y), \tag{141}$$

so that by maximizing the right hand side with respect to $\Delta_i$'s ($i = 1, 2, 3$) (using Eq. 138) one obtains $\Delta$'s (Eq. 97) and $M_{\mathrm{MF}}(\mathbf{A})$.

The same analysis is also applicable in the dual space, such that the right hand side of the Eq. 141 depends on $u$. To get the best result one should maximize $\mathcal{M}_u(\mathbf{A},y)$ with respect to $u$. This gives an alternative criterion to find the best $u$ value.

## 7 Discrete Laplacian

In this subsection we consider the matrix of discrete Laplacian (DL) defined as $[A_{\mathrm{DL}}]_{ii} = 2$ ($i = 1, ..., L$), $[A_{\mathrm{DL}}]_{i,i+1} = -1$ ($i = 1, ..., L-1$), $[A_{\mathrm{DL}}]_{i-1,i} = -1$ ($i = 2, ..., L$), $[A_{\mathrm{DL}}]_{1,L} = [A_{\mathrm{DL}}]_{L,1} = -1$. For this matrix we first provide an exact solution for the SPPM for arbitrary values of the powers of the principal minors. This helps us to interpret the system as a statistical model with an unusual Hamiltonian and the powers as the inverse of the temperature. Since the principal minors of the matrix of Laplacian can be interpreted as spanning forests one can consider the problem that we tackle here also as the statistical mechanics of the spanning forests. After providing a full exact solution of the model we show how the MF technique provided in the previous section can be used to get an approximate solution in the especial case $n = 2$. This section is used as a benchmark for the power of the MF method introduced in the previous section.

## 7.1 An exact solution

In this sub-section we present an exact solution for the discrete Laplacian, which is provided in Eq. 154 and its generalization Eq. 161. The partition function is enough to obtain for instance the average energy and entropy in Eqs. 156 and 157.

For a closed chain of size $L$ we define the partition function as follows:

$$Z(n, L) \equiv M^{(n)}(\mathbf{A}_{DL}) = \sum_I [\det \mathbf{A}_{DL}(I)]^n = \sum_I e^{-nE_I}. \tag{142}$$

Here, the energy function $E_I = -\ln \det \mathbf{A}_{DL}(I)$ depends on the number of clusters formed by the present (up) indices in the index configuration $I$. More precisely, $E_I = 0$ when $I$ includes all indices, otherwise,

$$E_I = -\sum_{l=1}^{L-1} m_l^+ \ln(l+1) = E\left(\{m_l^+\}\right), \tag{143}$$

where $m_l^+$ denotes the number of clusters of size $l$. Similarly, we define $m_l^-$ for the number of clusters of size $l$ formed by the absent (down) indices. Thus, we have,

$$L = \sum_l l m_l^+ + \sum_l l m_l^-. \tag{144}$$

Let us also define the total number of up and down clusters,

$$m^+ = \sum_l m_l^+, \qquad m^- = \sum_l m_l^-. \tag{145}$$

Note that in a closed chain $m^+ = m^- > 0$, except for the single configuration with all indices present in $I$. Now the partition function can be rewritten as

$$Z(n, L) = 1 + \sum_{m=1}^{\infty} \sum_{\{m_l^+, m_l^-\}} \delta_{m^{\pm}, m} \delta_{\sum_l l(m_l^+ + m_l^-), L} \mathcal{N}_{closed}\left(\{m_l^+\}, \{m_l^-\}\right) e^{-nE(\{m_l^+\})}, \tag{146}$$

where $\mathcal{N}_{closed}(\{m_l^+\}, \{m_l^-\})$ gives the number of index configurations $I$ with the specified number of up and down clusters. In Appendix E we obtain an exact expression for the number of such configurations,

$$\mathcal{N}_{closed}(\{m_l^+\}, \{m_l^-\}) = \frac{2L}{m^+ + m^-} \frac{m^+! m^-!}{\prod_l (m_l^+!) \prod_l (m_l^-!)}. \tag{147}$$

The partition function in terms of the spectral entropy $s(e)$ and free energy density $f$ reads as follows

$$Z(n, L) = \int e^{L(s(e') - ne')} de' = e^{-nLf}. \tag{148}$$

In the thermodynamic limit $L \to \infty$, by the saddle-point approximation, one finds

$$f = e - Ts, \tag{149}$$

where $T \equiv \frac{1}{n}$, $e \equiv \langle E \rangle / L$ is the energy density, and $s \equiv -\left(\frac{\partial}{\partial T} f\right)_L$ is the entropy density. The average of any observable $O$ is defined as

$$\langle O \rangle \equiv \frac{\sum_I O_I \exp(-nE_I)}{Z(n, L)}. \tag{150}$$

To compute the partition function $Z(n, L)$, we define the following generating function

$$G(n, \mu) = \sum_L e^{-\mu L} Z(n, L), \tag{151}$$

with $\mu$ as a chemical potential. Notably, in the thermodynamic limit, we have

$$G(n, \mu) \approx e^{-\mu \langle L \rangle} Z(n, \langle L \rangle) \equiv e^{-n \langle L \rangle g}. \tag{152}$$

Here $\langle L \rangle$ is the average $L$ where $e^{-\mu L} Z(n, L)$ shows a pronounced peak, and $g$ is the associated free energy density. Therefore, the free energies $f$ and $g$ are related as follows

$$g = f + \mu T = e - Ts + \mu T. \tag{153}$$

In Appendix E, we start from the partition function $Z(n, L)$ (Eq. 146) and obtain an exact expression for the above generating function,

$$G(n, \mu) = \frac{1}{1 - e^{-\mu}} + \frac{\partial}{\partial \mu} \ln \left( 1 - \frac{e^{-\mu}}{1 - e^{-\mu}} \left( \sum_l e^{-\mu l + n \ln(l+1)} \right) \right). \tag{154}$$

This expression for the generating function can be computed numerically to obtain the free energy $g$ and then $f$ from Eq. 153. Note that $G(n, \mu)$ is well defined only for $\mu > \mu_{th}$ where the threshold chemical potential depends on $n$. For a given $n$, we find numerically this threshold value and compute the interesting quantities for $\mu$ larger but very close to $\mu_{th}$. From the definition of the generating function, the average size of the system in the thermodynamic limit, see Eq. 150, is given by

$$\langle L \rangle = -\frac{\partial \ln G(n, \mu)}{\partial \mu}, \tag{155}$$

which is a decreasing function of $\mu$. And, $\langle L \rangle$ goes to infinity as $\mu \to \mu_{th}$. From the above equation we obtain the chemical potential as a function of $n$ and $\langle L \rangle$. In the same way, we compute the average energy

$$\langle E \rangle = -\frac{\partial \ln G(n, \mu)}{\partial n}. \tag{156}$$

This is the energy of relevant index configurations given $n$ and $\mu$. Recall that $E_I = -\ln \det \mathbf{A}_{DL}(I)$. In addition, we are able to find the entropy of index configurations given the energy density $e = \langle E \rangle / L$. This is done by another Legendre transformation,

$$s(e) = -n(f - e). \tag{157}$$

In words, given $n$ and $\mu$, we compute numerically the free energy $g$ and then $f, e, s$ for $\mu \to \mu_{th}$, i.e., for $\langle L \rangle \gg 1$. To construct $s(e)$, one has to do the above computation for different values of $n$ to find the entropy $s$ and energy density $e$ of the relevant index configurations as a function of $n$. Figure 2 displays the results obtained in this way for large system sizes. For comparison, we also report the exact results for a small chain of size $L = 20$. In fact the results of exact enumerations for small sizes converge very quickly to those of larger sizes obtained by the above solution. From the figure we also see that the entropy of minimum energy configurations is nonzero up to $n = 100$.

The average number of present clusters $\langle m_l^+ \rangle$ can be computed in a similar way by introducing another Lagrange multiplier $\lambda_l$ which is coupled to $m_l^+$,

$$Z(n, L, \{\lambda_l\}) = \sum_I e^{-nE_I + \sum_l \lambda_l m_l^+}, \tag{158}$$

$$G(n, \mu, \{\lambda_l\}) = \sum_L e^{-\mu L} Z(n, L, \{\lambda_l\}). \tag{159}$$

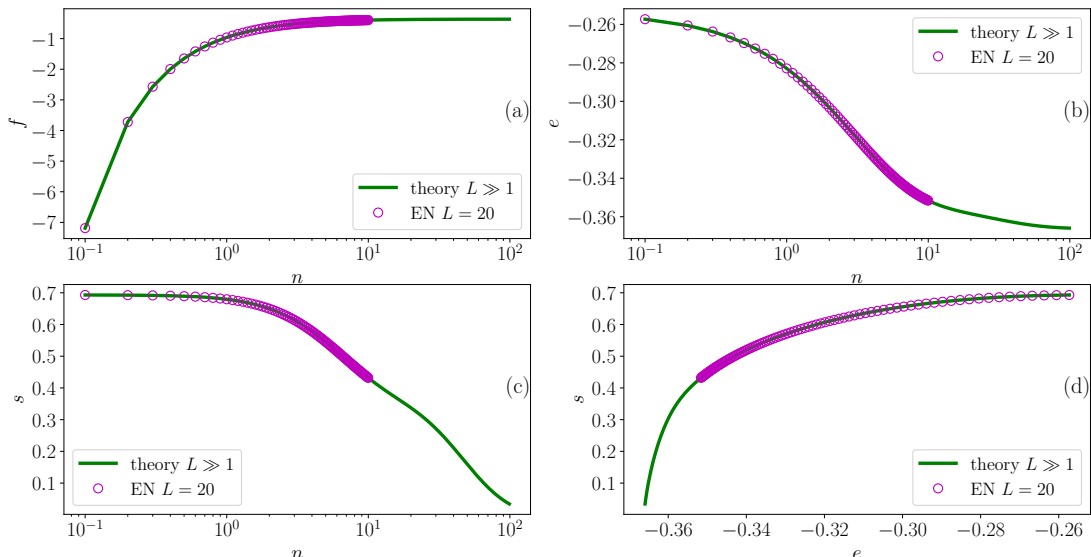

Figure 2: Exact solution for a large closed chain. (a) free energy, (b) energy, and (c) entropy vs the parameter $n$. (d) entropy vs energy. Analytical solution for very large sizes (theory $L \gg 1$) is compared with exact numerical solution for a small problem size (EN $L = 20$).

Then, we have

$$\langle m_l^+ \rangle = \frac{\partial \ln G(n, \mu, \{\lambda_l\})}{\partial \lambda_l}\Big|_{\{\lambda_l = 0\}}, \tag{160}$$

where

$$G(n, \mu, \{\lambda_l\}) = \frac{1}{1 - e^{-\mu}} + \frac{\partial}{\partial \mu} \ln\left(1 - \frac{e^{-\mu}}{1 - e^{-\mu}}\left(\sum_l e^{-\mu l + n \ln(l+1) + \lambda_l}\right)\right). \tag{161}$$

In this way, we obtain

$$\langle m_l^+ \rangle = -\frac{1}{G(n, \mu)} \frac{\partial}{\partial \mu}\left(\frac{e^{-\mu(l+1) + n \ln(l+1)}}{1 - e^{-\mu} - \sum_{l'} e^{-\mu(l'+1) + n \ln(l'+1)}}\right). \tag{162}$$

Figure 3 shows the distribution of present clusters as a function of $l$ and $n$ for very large closed chains. The numerical values are obtained as described above in the last paragraph. We observe that after $n = 8$ the system is dominated by clusters of size $l = 2$ (dimers).

In Appendix F we show that index configurations of maximum determinant (the ground states) of this system are dimer-covering configurations $\ldots \downarrow\uparrow\uparrow\downarrow\uparrow\uparrow\downarrow\uparrow\uparrow \ldots$, which are consistent with the size of the chain $L$. In fact, there are a sub-exponential number of such dimer-coverings which are relevant at zero temperature ($n \to \infty$). These ground states are organized in clusters of index configurations which are separated by extensive Hamming distances (number of different variables) in the configuration space. The complexity or configurational entropy of this system at zero temperature $\Sigma$ is define by the entropy density of such clusters. More precisely, depending on the size of chain we have,

$$\begin{cases} s = \Sigma = \ln(3)/L, & L \equiv 0 \mod 3, \\ s = \Sigma = \ln(L)/L, & L \equiv -1 \mod 3, \\ s = \ln(2L + X)/L, \quad \Sigma = \ln(L + X)/L, & L \equiv +1 \mod 3, \end{cases} \tag{163}$$

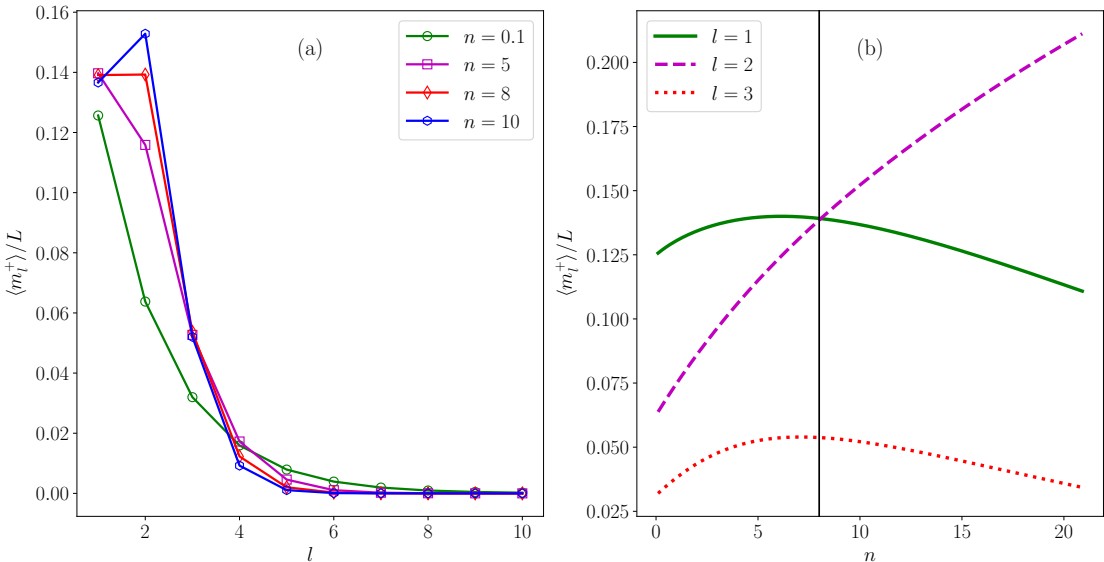

Figure 3: Distribution of the present clusters $m_l^+$ for a large closed chain. The results for different values of $l$ and $n$ are obtained by the exact solution as described in the text.

where $X$ is a number proportional to $L^2$ (see Appendix F for more details). Therefore, both the entropy density and complexity approach zero in limit $n, L \to \infty$. In Fig. 4 we compare the above theoretical results with numerical values that are obtained by exhaustive enumeration for small system sizes. Finally, it should be mentioned that we do not observe a finite-temperature phase transition in this problem despite the presence of extensively distant ground states in the configuration space. This stems from the one-dimensional character of the system; that is the domain walls which separate two different phases (ground states) of the system have a finite energy cost. Therefore, any finite temperature can destroy and destabilize the ordered phases by the extensive contribution of the entropy of such low-energy excitations.

### 7.2 Mean field results

In this subsection, we utilize the MF theory to analyze the SPPM problem for the DL matrix when $n = 2$. Initially, we investigate the problem within the original space, and subsequently, we demonstrate a substantial improvement in results achieved by exploiting the dual space formalism.

For the DL matrix $\Delta_2$ and $\Delta_3$ are identically zero based on numerical findings, whereas $\Delta_1$ exhibits non-zero behavior. To be more specific, $\Delta_1$ conforms to the equation G.2 (Sec. Appendix 7):

$$\Delta_1 = \frac{\det\left(\mathbf{A}_{\mathrm{DL},1} + \Delta_1 \mathbf{I}_1\right)}{\det\left(\mathbf{A}_{\mathrm{DL}} + \Delta_1 \mathbf{I}\right)} . \tag{164}$$

Here $\mathbf{A}_{\mathrm{DL},1}$ and $\mathbf{I}_1$ are $\mathbf{A}_{\mathrm{DL}}$ and $\mathbf{I}$, respectively, with the first row and column removed. While this equation can be solved numerically, an analytical expression is found using the continuum limit Eq. 106 based on the Fourier transformation of the Laplace matrix. This Fourier component is

$$A_{\mathrm{DL}}(q) = 4 \sin^2 \frac{q}{2} . \tag{165}$$

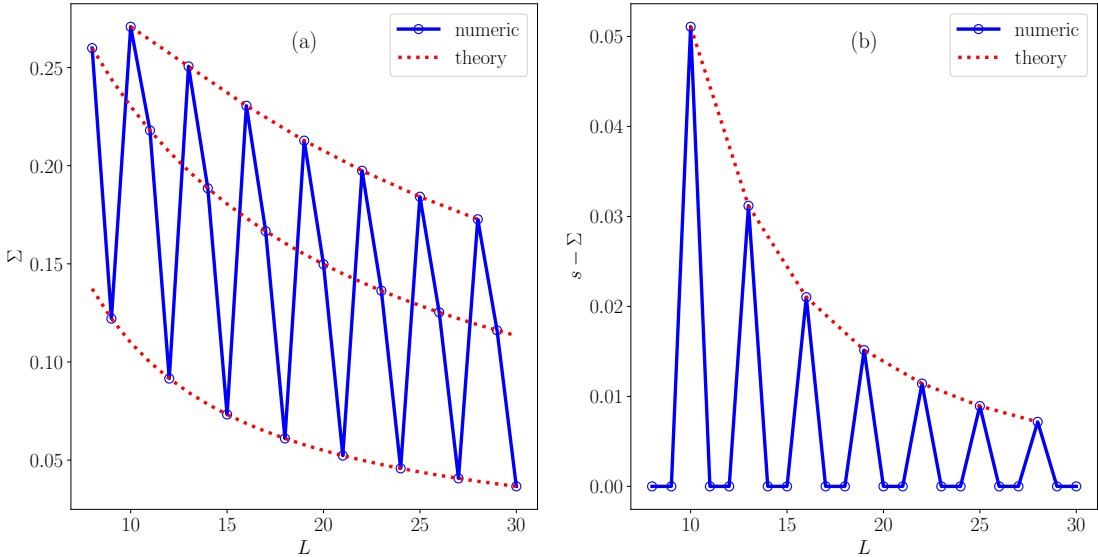

Figure 4: The entropy of ground states for Laplacian of a closed chain. The total entropy is $s = \log(\mathcal{N}_s)/L$ and the complexity is $\Sigma = \log(\mathcal{N}_c)/L$. Here $\mathcal{N}_s$ is the number of ground states and $\mathcal{N}_c$ is the number of clusters of the ground states. Two ground states are connected in the configuration space if their Hamming distance is 1. The exact numerical values are compared with the exact theoretical solutions.

The self-consistent equation is found to be

$$\Delta_1^3(\Delta_1 + 4) = 1, \tag{166}$$

the real solutions of which are $\Delta_1 = 0.601232$ and $\Delta_1 = -4.01545$, the second being less stable. It is worth noting that a method based on matrix calculus, presented in Appendix. 7 leads to the same result. Finally, using Eq. 98 one finds that for even $L$ values ($\Delta_4 = \Delta_1^2$)

$$Z(2, L) = M^{(2)}(\mathbf{A}_{\mathrm{DL}}) \propto e^{-2fL+\beta}, \tag{167a}$$

$$f = \log\left[\frac{1}{2}\left(\Delta_1 + 2 + \sqrt{\Delta_1(4 + \Delta_1)}\right)\right] + \frac{\Delta_1^2}{2}, \tag{167b}$$

$$\beta = \log \Delta_1^2. \tag{167c}$$

Using the above equation we find that $f \approx -0.576$ and $\beta \approx -1.01755$. This should be compared with the analytical result (the previous section) $f_{\mathrm{analytic}} \approx -0.63$.

For the dual space MF with $m_u = -\sqrt{\sqrt{u} - u}$, using Eq. D.42 one finds the following self-consistent equation for $\Delta'_2 = \Delta'_3 = 0$:

$$\Delta'_1 = \frac{u}{\Delta'_1 + m_u}\left(1 - \frac{u^{\frac{1}{4}}}{\sqrt{\Delta_1'^2(1 + 4m_u - \sqrt{u}) + \Delta'_1(4\sqrt{u} - 2m_u u^{\frac{1}{2}} - 8u) - 4m_u u + u^{\frac{3}{2}}}}\right). \tag{168}$$

Then using Eq. D.38, and especially Eq. D.43 we find

$$\ln M^{(2)}_{u,\mathrm{MF}}(\mathbf{A}_{\mathrm{DL}}) = \frac{L}{\pi}\int_0^{2\pi} dq \ln\left|A_{\mathrm{DL}}(q)(m_u + \Delta'_1) + \frac{m_u}{\sqrt{u}}\Delta'_1 - \sqrt{u}\right| - L\left[\frac{1}{2}\ln u + \Delta'_4\right] + O(1). \tag{169}$$

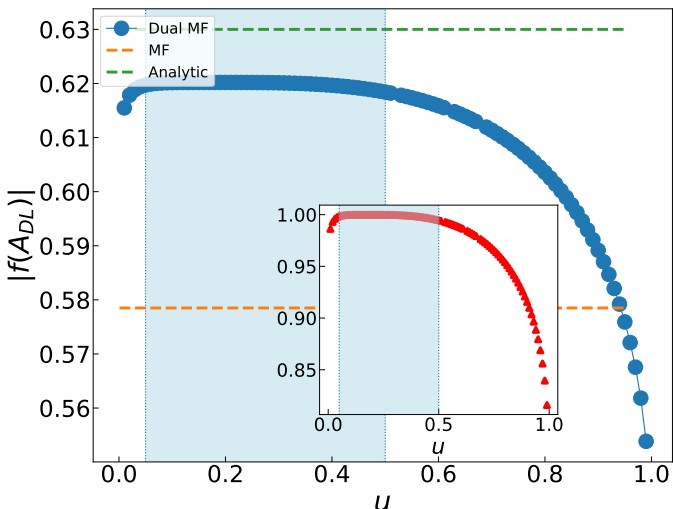

Figure 5: The stability of the MF results for discrete Laplacian. The main panel shows the dual MF result for $f$ with ($L = 100$) in terms of the Sourlas transformation parameter $u$. For comparison, the MF and analytical results are also presented by the dashed lines. The error of the result for dual MF in the blue area is less than 1.5%, while the error of the MF result in the real space is about 8%. The inset shows the same quantity using the saddle-point approximation $f_{SP}^u$ normalized by $f_{MF}^u$, i.e. $f_{SP}^u/f_{MF}^u$. The blue area exhibits the region where the solution is most stable defined by the threshold $f_{SP}^u/f_{MF}^u > 0.99$.

The efficiency of the dual MF solution is another issue to be addressed here, for which we use Eq. 131. Figure 5 shows the results for DL for $L = 100$. $f(\mathbf{A}_{DL})$ is shown in the main figure, and the stability test result is presented in the inset. First observe that $|f(\mathbf{A}_{DL})|$ is greater than that resulted from the MF theory in the direct and the dual space as predicted in Eq. 141. The blue area in the inset shows the region where the solution satisfies $f_{SP}^u/f_{MF}^u > 0.99$, i.e. the MF solution is highly stable. In the main part of the figure, the same area is emphasized. It's noteworthy that within this region, where the mean field (MF) solution is stable, there is a close correspondence between the MF results and the exact numerical outcomes. Specifically, the deviation from the exact results is just about 1.5 percent, indicating a good agreement in this particular region.

# 8 Shannon-Rényi entropy of the transverse field Ising chain

This section is devoted to the Rényi entropy ($n = 2$) of the ground state of TFI chain. We will demonstrate that the MF solution is capable of detecting the phase transition point, and estimating quantities of interest like $R_2$ (Eq. 14). For the relation between the Rényi entropy and SPPM see Eq. 8. The Hamiltonian of the TFI chain is given by:

$$H_{\text{Ising}} = -\frac{1}{2}\sum_{j=1}^{L}\sigma_j^x\sigma_{j+1}^x - \frac{h}{2}\sum_{j=1}^{L}\sigma_j^z, \qquad (170)$$

where $\sigma^{x,y,z}$ are Pauli matrices, $h$ is a magnetic field, and periodic boundary conditions are imposed, i.e. $\sigma_{L+1}^x \equiv \sigma_1^x$. The Ising model is critical at $|h| = 1$. More precisely, the system is in the paramagnetic phase for $|h| > 1$, while for $|h| < 1$ it is in the ferromagnetic phase, and

$|h| = 1$ is the transition point, where the system becomes critical. Using the Jordan-Wigner transformation defined as

$$c_j^\dagger \equiv \prod_{m<j} \sigma_m^z \sigma_j^+, \qquad c_j \equiv \prod_{m<j} \sigma_m^z \sigma_j^-, \tag{171}$$

where $\sigma_j^+ \equiv \frac{1}{2}\left(\sigma_j^x + i\sigma_j^y\right)$, and $\sigma_j^- \equiv \frac{1}{2}\left(\sigma_j^x - i\sigma_j^y\right)$, one can map the TFI chain problem to a discrete fermionic Hamiltonian

$$H_{\text{Ising}}^{\text{F}} = \frac{1}{2}\sum_{j=1}^{L}\left[c_j^\dagger c_{j+1} + c_j^\dagger c_{j+1}^\dagger + h.c.\right] - h\sum_{j=1}^{L} c_j^\dagger c_j, \tag{172}$$

which is an especial case of Eq. 2. In this equation $c_j$ ($c_j^\dagger$) is annihilation (creation) operator defined in Eq. 2. The Jordan-Wigner transformation leaves a degree of freedom (concerning the periodicity of the system), according to which two sectors can be taken: $c_{L+1} = c_1$ is the Ramond (R, periodic) sector, while $c_{L+1} = -c_1$ is the Neveu-Schwartz (NS, antiperiodic) sector. For the Ising model the ground state of the spin chain corresponds always to the ground state of $H_{\text{Ising}}^{\text{F}}$ in the NS sector. Therefore, we consider the NS sector all over this paper. By diagonalizing the Hamiltonian, one can find the energy spectrum of the system. The single particle energy spectrum of this system is $\epsilon_q \equiv \sqrt{1+h^2 - 2h\cos q}$. The correlation matrix (defined in Eq. 3) for the Ising model is [24]

$$G_{jk} = \frac{1}{L}\sum_{m=1}^{L} \sigma_{q_m} e^{iq_m(j-k)}. \tag{173}$$

Here $\sigma_q = \frac{f(e^{iq})}{|f(e^{iq})|}$ with $f(z) = h - z$ and $q_m = \frac{2\pi}{L}(m - \frac{1}{2})$. Using the properties of the circulant matrices, one can find the elements of $\mathbf{F}$ matrix defined in Eq. 5 as

$$F_{jk} = \frac{1}{L}\sum_{m=1}^{L} F_{q_m} e^{iq_m(j-k)}, \tag{174}$$

where $F_q \equiv \frac{1+\sigma_q}{1-\sigma_q}$. Note that $F_q = -i\cot\frac{\theta_q}{2}$, $\theta_q \equiv \tan^{-1}\frac{\sin q}{\cos q - h}$, so that $F_q = -i\tan\frac{q+\pi}{4}$ for $h = 1$. To test the scaling properties of $F_q$, we notice that there are two singular points for $\ln F_q$: $q \to 0$ and $q \to \pi$, in the vicinity of which $F_q$ has the following asymptotic expansions

$$F_q|_{h\neq 1, q\to 0} = -i\left(\frac{q}{2|1-h|}\right)^{\text{sgn}(1-h)} + O\left[q^{2+\text{sgn}(1-h)}\right], \tag{175a}$$

$$F_q|_{h\neq -1, q\to \pi} = i\left(\frac{2|1+h|}{q-\pi}\right)^{\text{sgn}(1+h)} + O\left[(q-\pi)^{2-\text{sgn}(1+h)}\right]. \tag{175b}$$

In addition, $\lim_{q\to 0} F_q(h=1) = -i\,\text{sgn}(q) - \frac{iq}{2}$ and $\lim_{q\to\pi} F_q(h=-1) = i\,\text{sgn}(q-\pi) - \frac{i(q-\pi)}{2}$. Such singularities and the corresponding exponents are important in the integral representations like Eq. 98, in the vicinity of the transition point (critical region) where $\Delta$ functions approach zero. This becomes clearer in the following discussions. It is worth noting that the transformation $h \to -h$ (along with $q \to q + \pi$) corresponds to $\sigma_q \to -\sigma_q$, giving rise to $\mathbf{F} \to \mathbf{F}^{-1}$. It corresponds to a dual transformation Eqs. 37 with $u = 1$ and $m = 0$ (with an extra minus factor). This shows that $R_2$ for the Ising model has the symmetry $h \to -h$.

Although the determination of $R_2$ is a challenge, having analytic expressions for specific values of the transverse field is very helpful in characterizing this function. Analytical and

numerical findings provide evidence that, for all values of $h$, to the leading order of $L$ we have [40]

$$R_2 = \alpha_2(h)L + \beta_2(h).\tag{176}$$

An important limit is $h = 0$, for which the $Z_2$ symmetric ground state is given by $|g\rangle_{h=0} = \frac{1}{\sqrt{2}}[|\uparrow\uparrow ... \uparrow\rangle_x + |\downarrow\downarrow ... \downarrow\rangle_x]$. Then the probability of the configuration $|\sigma_1\sigma_2...\sigma_L\rangle_z$ in the ground state is easily found to be $P_{h=0}(\sigma_1, \sigma_2, ..., \sigma_l) = \frac{1}{2^{L-1}}\left[\frac{1+(-1)^{\sigma_1\sigma_2...\sigma_L}}{2}\right]$, resulting to an exact result $\alpha_2(0) = \beta_2(0) = \ln 2$. The other limit is the critical Ising chain ($h = 1$) that was numerically investigated in Ref. [40], where based on numerical fittings up to size $L = 44$ it was asserted that to the leading orders $\alpha_2(1) = 0.2138$ and $\beta_2(1) \simeq 0$.

Next, we shift our focus to the mean field analysis and employ the self-consistent equations 91 with **A** substituting with **F**. In the critical region one expects that $\Delta$ functions scale with $a_0 \equiv |1-h|$ (around $h = 1$ for which $\tau = 1$) or $a_0 \equiv \frac{1}{|1+h|}$ (around $h = -1$ for which $\tau = -1$), with a $\tau$-dependent exponent according to Eq. 104. For the former case ($h \to 1^-$), this relation predicts that $\Delta_4 \propto (1-h)^2$. Equation 175b can also be used for estimating the last term of Eq. 98 in the $L \to \infty$ limit

$$\lim_{L\to\infty} \ln M_{\mathrm{MF}}^{(2)} = \left(\int_{-\pi}^{\pi} \frac{\mathrm{d}q}{2\pi} \ln\left[-F_q^2 + \Delta_4\right] - \Delta_4\right)L + \lim_{q\to 0} \ln\left[-F_q^2 + \Delta_4\right] + O(L^{-1}).\tag{177}$$

For $h \geq 1$ as we will see $\Delta_4 = 0$, which, when combined with Eq. 8 gives

$$\begin{aligned}R_2^{\mathrm{MF}}(\Delta_4 = 0) &= -2\ln\det\left(\frac{\mathbf{I}-\mathbf{G}}{2}\right) - 2\ln\det\mathbf{F}\\ &= -2\ln\det\left(\frac{\mathbf{I}+\mathbf{G}}{2}\right),\end{aligned}\tag{178}$$

showing that in the paramagnetic phase $R_2$ is related to the probability of a single spin configuration, known as the emptiness formation probability (EFP). Writing the EFP as

$$\log\left[|\langle\uparrow\uparrow ... \uparrow |g\rangle|^2\right] = -\zeta(h)L + O(L^{-1}),\tag{179}$$

and using Eq. 176 one concludes that for $h \geq 1$,

$$\alpha_2(h) \approx -2\zeta(h), \quad \beta_2(h) \approx 0.\tag{180}$$

The exact form of $\zeta(h)$ is given by [44]

$$\zeta(h) \equiv \frac{1}{2\pi}\int_0^{\pi} \mathrm{d}q \ln\left(\frac{1}{2} + \frac{h-\cos q}{2\epsilon_q}\right).\tag{181}$$

For $h = 1$, we have

$$\zeta(h = 1) = \frac{2C - \pi\ln 2}{\pi},\tag{182}$$

where $C$ is the Catalan number, giving rise to

$$\alpha_2^{\mathrm{MF}}(h = 1) = -2\zeta(h = 1) \approx 0.22005,\tag{183a}$$

$$\beta_2^{\mathrm{MF}}(h = 1) = 0.\tag{183b}$$

The single spin configuration plays dominant role also in large $n$ limit. Suppose that there is a single configuration ($C_0$) in the expansion 7 such that $P(C_0)$ is larger than the probability of any other configuration. Then

$$R_n(L) = \frac{n}{1-n}\ln P(C_0) + \frac{1}{1-n}\ln\left(1 + \sum_C{}' x_C^n\right),\tag{184}$$

can be treated perturbatively for large enough $n$ values, where $x_C \equiv P(C)/P(C_0) < 1$, and $\sum'_C$ runs over all configurations except $C_0$. To the leading order we find

$$\lim_{n\to\infty} R_n(L) = -\ln P(C_0). \tag{185}$$

The first correction to this formulae is $-\frac{x_{C_1}^n}{n}$, where $C_1$ is the next most probable configuration.

We see that for large $n$ values when $h \geq 1$ the way to get closer to the exact result is to consider the contribution of the other most probable configurations. For Eq. 178 one may add *kink*s (spin flips). For example a spin flip at site $i$ corresponds to removing $i$th row and column of $\mathbf{F}$. Taking into account the effect of configurations with two spin flips (which serves as a first order correction to MF due to the parity number symmetry), up to $L_{\max} = 74$ results in

$$\alpha_2^{1st}(h = 1, L_{\max} = 74) \approx 0.2156, \tag{186}$$

which is just 0.8% away from the best available numerical results.

Interpreting $R_2$ as a free energy, one can study singularity of the derivative with respect to $h$ (which can be viewed as specific heat). One can show that $\frac{\partial \alpha_2^{MF}}{\partial h}$ is logarithmically divergent at $h = 1$. To see this, we note that for $h \neq 1$, and to $O(L^{-1})$:

$$\frac{1}{L} \lim_{L\to\infty} \frac{\partial \ln M_{MF}^{(2)}}{\partial h} = i \int_{-\pi}^{\pi} \frac{dq}{2\pi} F_q \left( \frac{1 - F_q^2}{\Delta_4 - F_q^2} \right) \frac{\sin q}{\epsilon_q^2}, \tag{187a}$$

$$\lim_{L\to\infty} \frac{\partial \zeta(h)}{\partial h} = \int_0^{\pi} \frac{dq}{2\pi} \frac{-h + \cos q + \epsilon_q}{\epsilon_q^2} = -\frac{\Theta(h-1)}{2h} + \frac{1}{\pi(1+h)} K \left( \frac{4h}{(1+h)^2} \right), \tag{187b}$$

where $K(y)$ is a complete elliptic integral of the first kind, and $\Theta(y)$ is the step function, i.e. $\Theta(y) = 1$ if $y > 0$ and zero otherwise. Using Eq.178 for $\Delta_4 = 0$, one finds for all $h \geq 1$, and also the limit $h \to 1^-$ for which $\Delta_4$ is zero or negligibly small, we have ($h \neq 1$):

$$\frac{1}{L} \lim_{L\to\infty} \frac{\partial \ln M_{MF}^{(2)}}{\partial h} = \frac{4}{\pi(1+h)} K \left( \frac{4h}{(1+h)^2} \right). \tag{188}$$

When $h \to 1^\pm$, one finds ($+O(1-h)$)

$$\frac{1}{L} \lim_{L\to\infty} \left. \frac{\partial \ln M_{MF}^{(2)}}{\partial h} \right|_{h\to 1^\pm} = -\frac{2}{\pi} \ln \left( \frac{|h-1|}{8} \right), \tag{189a}$$

$$\lim_{L\to\infty} \left. \frac{\partial \zeta(h)}{\partial h} \right|_{h\to 1^\pm} = -\frac{\Theta(h-1)}{2} - \frac{1}{2\pi} \ln \left( \frac{|h-1|}{8} \right). \tag{189b}$$

This leads us to conclude that $\frac{\partial R_2}{\partial h}$ is logarithmically divergent at $h = 1$.

The above analysis gives the analytic expressions for $R_2$ in the vicinity of the transition point. For a general behavior of $R_2$ we numerically solve the MF equations and compare them with exact numerical (EN) results for small sizes. Equations 119 and H.2 give exact expressions for $M^{(2)}(\mathbf{F})$ in terms of Gaussian integrals of $\mathcal{M}^{(2)}(\mathbf{F}, y)$ and $\mathcal{M}_u^{(2)}(\mathbf{F}, y)$. In Appendix J the MF solutions are processed as the stationary points/lines of $\mathcal{M}^{(2)}(\mathbf{F}, y)$ and $\mathcal{M}_u^{(2)}(\mathbf{F}, y)$. The phase space is generally very big in terms of $\Delta_j^{(i)}$ ($i = 1, 2, 3$, $j = 1, ..., l$). We can, however, project to the subspace $\Delta_j^{(i)} = \Delta_{j'}^{(i)}$ which is physically a significant subspace. Since $\mathbf{F}$ is antisymmetric, one expects that the MF solution of $\mathcal{M}^{(2)}(\mathbf{F}, y)$ depends only on $\Delta_4$. A detailed analysis of the landscape of $\mathcal{M}^{(2)}(\mathbf{F}, y)$ and $\mathcal{M}_u^{(2)}(\mathbf{F}, y)$ in terms of the $\Delta$ functions is presented in Appendix J.

To directly find the MF results, we first solve numerically Eq. 91 for the direct space, and Eq. 107 in the dual space (for which we used the "FindRoot" solver in Mathematica software), and then use Eq. 93 and Eq. 109 to find the numerical estimations for $M^{(2)}(\mathbf{F})$ in the direct and dual space, respectively. We concentrate on positive $h$ values, and for the dual MF analysis we choose the negative branch $m = -\sqrt{\sqrt{u}-u}$. The Rényi entropy in the $\sigma^z$ basis ($R_2$) is analyzed in Fig. 6 where the EN results are compared with the MF estimations. For the direct MF theory, $\Delta_2$ is found to be identically zero for all $h$ values, while $\Delta_1$ and $\Delta_3$ depend on $h$. Noting that $\mathbf{F}$ is an antisymmetric matrix, we expect that the MF functions are sole functions of $\Delta_4$ (see Sec. 6.1). $\Delta_4$ is depicted in Fig. 6a. When we start with small $h$ values, $\Delta_4$ monotonically decreases with $h$ up to a transition point $h^*$ (in the vicinity of the critical point $h_c = 1$) where an abrupt change of behavior is observed. In fact, for $h > h^*$ all the two point functions are identically zero, while for $h < h^*$ they are non-zero. Our observations indicate that $h^*$ approaches $h_c = 1$ in the $L \to \infty$ limit. More precisely, we found that $1 - h^* = \zeta_1 L^{-\zeta_2}$, where $\zeta_1 \approx 1.0352$, and $\zeta_2 \approx 0.577$, which suggests that MF recognizes the exact critical point. In accordance with the arguments presented in Section 6.2, in the paramagnetic phase ($h > h^*$), the system exhibits complete symmetry, resulting in all correlation functions being zero ($\Delta_1 = \Delta_2 = \Delta_3 = 0$). However, as the system undergoes a transition to the ferromagnetic phase ($h < h^*$), certain symmetries of the original effective action are spontaneously broken. As a result, $\Delta_4$ acquire non-zero values in the latter phase. $\Delta_2$ remains zero in the broken phase due to the presence of the axial $U(1)$ symmetry.

Figure 6b reveals that $R_2$ monotonically decreases with $h$, satisfying Eq. 141 for all $h$ and $L$ values, i.e. $R_2^{\mathrm{EN}} \le R_2^{\mathrm{MF}}$ (Note that it is related to the minus of logarithm of $M^{(2)}(\mathbf{F})$). It shows a linear behavior in terms of the system size in accordance with Eq. 177. The local slope (derivative) of this function with respect to $h$ shows a peaked structure which is depicted in Fig. 6c. As the value of $L$ increases, the peak position shifts towards the right and becomes sharper in accordance with Eq. 187, approaching the critical point at $h = 1$. The numerical values for $\alpha_2$ and $\beta_2$ are presented in Fig. 6d for both the EN and MF methods. For sufficiently large $h$ values ($h \gtrsim 0.5$), the direct MF results show good agreement with the EN results, while for smaller $h$ values ($h \lesssim 0.5$), the agreement is not as satisfactory. As $h$ increases, a smaller number of configurations play a significant role in the summation Eq. 8 for the Rényi entropy, and for $h > 1$ the $|\uparrow\uparrow \dots \uparrow\rangle_z$ configuration significantly dominates. The dominant configurations have a considerably larger formation probability compared to the rest, which leads to better performance of the MF theory due to reduced fluctuations in the configuration space. In appendix J we argue that the inefficiency of MF theory in the direct space for small $h$ values can be attributed to the appearance of extra peaks for $\mathcal{M}^{(2)}(\mathbf{F}, y)$. For $h < 0$ the dual MF with $u = 1$ and $m = 0$ ($\mathbb{N} = -\mathbb{F}^{-1}$) gives reliable results due to the $h \to -h$ symmetry.

For a systematic analysis, we also examined the dual MF approach by employing the dual MF equations 107 and 109, following the same methodology as the direct MF method. Appendix J provides a general picture of $\mathcal{F}_u^{(2)}\big(\mathbf{F}, y \equiv \{\Delta_i\}_{i=1}^3\big)$ in the dual space. There we illustrate that for the $u$ values where $\mathcal{F}_u^{(2)}$ has a single minimum (stationary point), the MF results are reliable, although the most robust criterion for the validity of the MF results is the stability test presented in Sec. 6.4. In Fig. 6d we show $\alpha_2$ and $\beta_2$ in terms of $h$ for the direct and the dual MF theories as well as the EN results. For example $\alpha_2^{(u=0.2)}(h = 0) = 0.693145$, which should be compared with the exact result $\alpha_2(h = 0) = \ln 2 \approx 0.693147$. For the dual MF theory we used the "best" $u$ values. Figure 7 shows the SPPM of the Ising model for $h = 0$ and $L = 30$ in terms of $u$, where the inset shows the stability according to Eq. 131. Within the shaded blue region, we have identified a highly stable solution that satisfies $f_{\mathrm{SP}}^u / f_{\mathrm{MF}}^u > 0.99$. It is noteworthy that in this region, the dual MF results exhibit good agreement with the exact numerical results, while the direct MF predictions are poor. This finding further supports our main hypothesis that the MF results are more reliable for more stable MF solutions. A similar

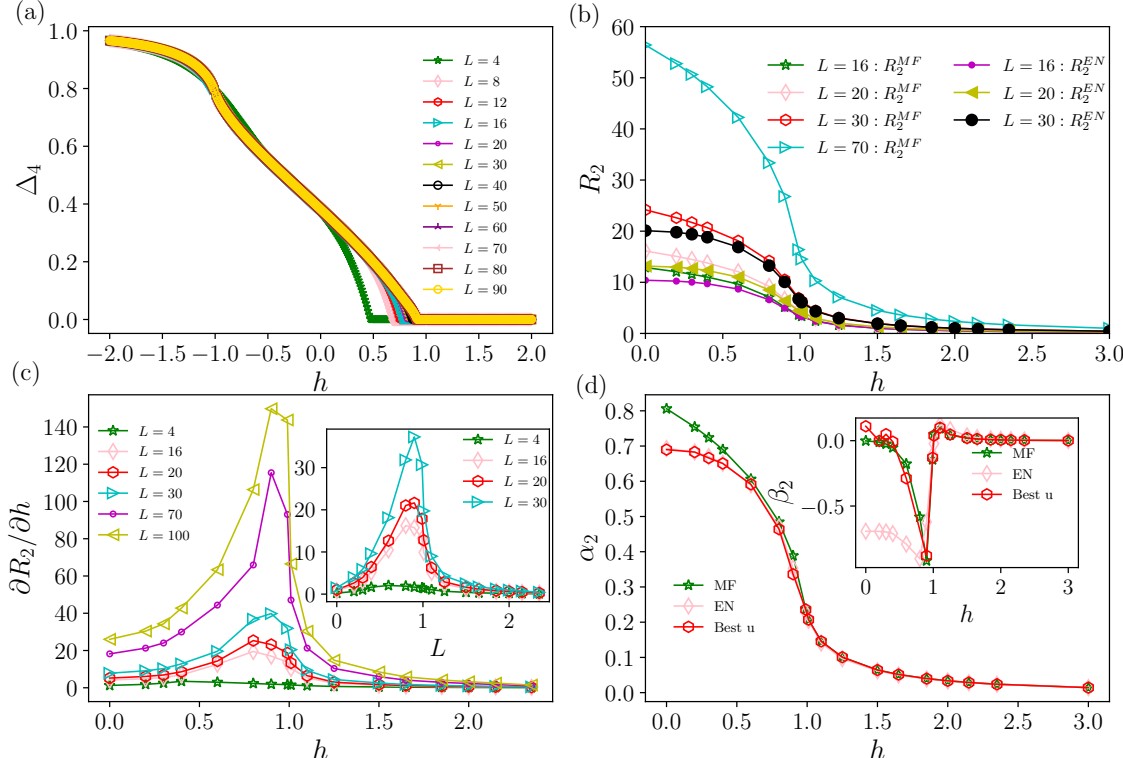

Figure 6: (a) $\Delta_4$ in terms of $h$ in the MF approximation for various values of $L$. $h^*$ is defined as the point below which $\Delta_4$ becomes non-zero for the first time. (b) MF approximation and EN estimation for $R_2$ in terms of $h$ for various values of system size $L$. (c) MF approximation (main) and EN estimation (inset) for $\partial R_2/\partial h$ in terms of $h$ with a peak structure. (d) The fitting parameters $\alpha_2$ and $\beta_2$ according to Eq. 176, in terms of $h$ for MF, EN and dual MF, where $L_{\max}$ (the maximum point up to which the fitting is done) is 18 for EN and $L_{\max} = 80$ for MF. The red hexagons show the results for the dual MF with "best" $u$ values. For more details see Fig. 15, and the "best" $u$ values are reported in Fig 15c.

pattern is also observed for other values of $h$, as detailed in Appendix J. For high magnetic fields ($h > h^*$), where the results of the direct-space MF method are reliable, the dual MF results are not consistent with EN results.

# 9 Conclusions

In this study, we demonstrated a novel approach to approximate the SPPM by writing the sum as a fermionic integral and employing MF theory. We presented three distinct versions of the MF theory, each with its unique advantages. Among these versions, the one based on the SP approximation appears particularly well-suited for extending the approximation to higher levels of perturbation theory. This can be achieved by leveraging the expansion of the exponential function and utilizing Bell polynomials. Although this process can be done systematically, we have deferred the intricate details to future investigations.

Our primary focus in this work was on the case when $n$ equals 2. However, it is worth noting that the generalization to larger values of $n$ is feasible through the judicious application of bosonization techniques in conjunction with the SP approximation. Despite the straightfor-

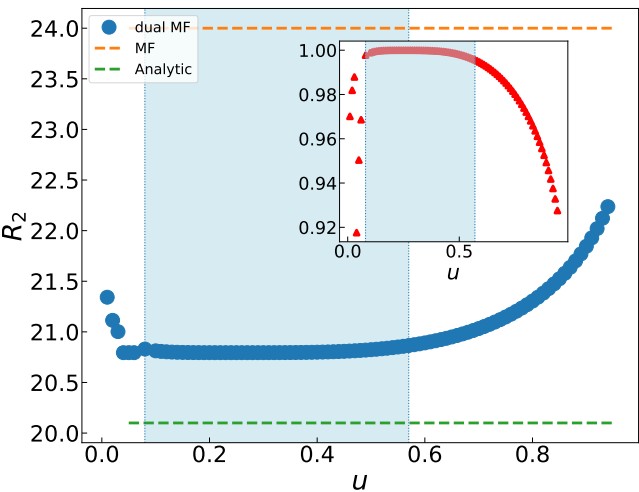

Figure 7: Main panel: The Reǹyi entropy ($R_2$) of the TFI chain in the MF approximation in terms of $u$ (dual space parameter) for $h = 0$ and $L = 30$. The MF in direct space and the analytical results are also presented by the dashed lines. Inset: $R_2^{\mathrm{SP}}(u)/R_2^{\mathrm{MF}}(u)$, showing the stability of the MF results according to Eq. 131. The blue area in the main panel and the inset represents the most stable solution defined by the threshold $R_2^{\mathrm{SP}}(u)/R_2^{\mathrm{MF}}(u) > 0.99$. Note that $(R_2^{\mathrm{MF}} - R_2^{\mathrm{Analytic}})/R_2^{\mathrm{Analytic}} \approx 0.19$ for the MF in real space, while for the best $u$ values in the dual space we have $(R_2^{\mathrm{MF}}(u) - R_2^{\mathrm{Analytic}})/R_2^{\mathrm{Analytic}} \approx 0.04$.

ward conceptual framework, the actual computational aspects can be quite cumbersome. It would be an intriguing endeavor to pursue this generalization in a systematic and methodical manner.

By transitioning to the dual space, we significantly improved the results obtained through the MF theory. Surprisingly, the MF results in the dual space outperformed those in the real space, especially in cases where the real space results were subpar. The concept of dualities, such as the one proposed by Sourlas, appears to have been astonishingly overlooked in this context.

Although the reason behind the success of the duality seems intuitively understandable, as it involves a change in the range of interactions, the extent to which one can approach the exact value by utilizing the dual space remains far from obvious. Addressing this question would mark a noteworthy achievement.

In our study, we tackled two specific examples: the discrete Laplacian of a chain, which can be considered as the partition function of rooted spanning trees, and the Rényi entropy of the ground state of a TFI chain. For the first example, which allowed us to explore the arbitrary SPPM index ($n$) values, we managed to solve the problem exactly. It served as a benchmark, illustrating how closely one can approximate the exact value through MF calculations in both real and dual spaces.

The second example involved investigating the TFI chain and its phases by analyzing the second Rényi entropy of the ground state. This approach offers a remarkably simple, intuitive, and experimentally relevant means of studying this well-known spin chain. Until now, all studies related to this quantity had been numerical, but we were able to provide the first analytical calculation. Surprisingly, MF theory precisely determined the critical point, and gave a good estimation of the entropy even deep within the ferromagnetic phase. The Rényi entropy discussed here is experimentally easier to measure than entanglement, and as demonstrated in this paper, it can also be calculated analytically.

It's important to note that our examples include periodic boundary conditions (PBC), especially when studying the Rényi entropy of the entire system. In such cases, we can formulate and estimate the MF equations analytically. However, when PBC is not present, as in the case of the Rényi entropy of a subsystem of the TFI chain, one possibly needs to rely more on numerical techniques. These analyses are of utmost importance because the Rényi entropy contains the *central charge,* a critical parameter for the underlying conformal field theory at the critical point. Recently, the Rényi entropy has been used to measure the central charge for the first time in quantum spin chains [30]. The techniques outlined in this paper offer a way to calculate it analytically.

## Acknowledgments

We thank F. Alcaraz, M. Dalmonte and Markus Heyl for discussions.

**Funding information**    MAR thanks CNPq and FAPERJ (grant number E-26/210.062/2023) for partial support.

## A    Berezin integrals of Grassmann variables

In this section, we fix the notation and summarize main results regarding the Berezin integration of Grassmann variables, see [45] for further details. Consider an $l \times l$ matrix denoted as $A$. We define sets $I_r$ and $J_r$ as $\{i_1, i_2, ..., i_r\}$ and $\{j_1, j_2, ..., j_r\}$, and the complements of $I_r$ and $J_r$ as $I_r^c$ and $J_r^c$, respectively. $A_{I_r J_r}$ is a submatrix of $A$ corresponding to the rows $I_r$ and the columns $J_r$, all kept in their original order, and $\epsilon(I_r, J_r) \equiv (-1)^{\sum_{i \in I_r} i + \sum_{j \in J_r} j}$. We also represent

$$\int \prod_{\alpha=1}^{r} \big(d\xi_{i_\alpha} d\bar{\xi}_{j_\alpha}\big)(...) \equiv \int_{\xi, \bar{\xi}} (...) \,, \tag{A.1}$$

to facilitate the representation, where $\xi_i$ and $\bar{\xi}_i$ are some Grassmann variables. If $\mathbf{A}$ is invertable, then

$$\int_{\xi, \bar{\xi}} \exp\big(\bar{\xi}^T \mathbf{A} \xi + \bar{\lambda}^T \xi + \bar{\xi} \lambda\big) = \det \mathbf{A} \exp\big[-\bar{\lambda}^T \mathbf{A}^{-1} \lambda\big] \,, \tag{A.2}$$

where "$T$" stands for the "transpose", and $\lambda = \{\lambda_i\}_{i=1}^{l}$ and $\bar{\lambda} = \{\bar{\lambda}_i\}_{i=1}^{l}$ are Grassmann vectors. We also have the following identity for $2r$-point Grassmann functions

$$\int_{\xi, \bar{\xi}} \left(\prod_{\alpha=1}^{r} \bar{\xi}_{i_\alpha} \xi_{j_\alpha}\right) \exp\big(\bar{\xi}^T \mathbf{A} \xi\big) = \epsilon(I_r, J_r)\big(\det \mathbf{A}_{I_r^c J_r^c}\big) \,, \tag{A.3}$$

and if $\mathbf{A}$ is invertable then

$$\int_{\xi, \bar{\xi}} \left(\prod_{\alpha=1}^{r} \bar{\xi}_{i_\alpha} \xi_{j_\alpha}\right) \exp\big(\bar{\xi}^T \mathbf{A} \xi\big) = \det \mathbf{A} \det\big[\big(\mathbf{A}^{-T}\big)_{I_r J_r}\big] \,. \tag{A.4}$$

By employing this expression, it is possible to demonstrate the following identity

$$\det\big[\mathbf{A}^{-T}\big]_{I_r^c J_r^c} = \epsilon(I_r, J_r) \frac{\det[\mathbf{A}]_{I_r J_r}}{\det \mathbf{A}} \,, \tag{A.5}$$

where $\mathbf{A}^{-T} \equiv \big(\mathbf{A}^{-1}\big)^T$.

Additionally, there exist links between Grassmann integrals and Pfaffians in the context of antisymmetric matrices. In the following, we explore a few connections for any arbitrary antisymmetric matrix $\mathbf{A}$ [45]

$$\int_\xi \exp\left[\frac{1}{2}\xi^T\mathbf{A}\xi\right] = \int_\xi \exp\left[-\frac{1}{2}\xi^T\mathbf{A}\xi\right] = \mathrm{Pf}[\mathbf{A}]\,, \tag{A.6a}$$

$$\int_\xi \left(\xi_{i_1}\xi_{i_2}\ldots\xi_{i_r}\right)\exp\left[\frac{1}{2}\xi^T\mathbf{A}\xi\right] = \left(\mathrm{Pf}[\mathbf{A}]\right)\mathrm{Pf}\left[\left(\mathbf{A}^{-T}\right)_{I_rI_r}\right]\,, \tag{A.6b}$$

where $\mathrm{Pf}[\mathbf{A}]$ is the Pfaffian of an antisymmetric matrix $\mathbf{A}$.

# B  Generalized partition function and the Sourlas transformation

This section provides further elaboration on the Sourlas transformation, which is a duality established in [39]. We define $\chi$, $\bar\chi$, $\phi$ and $\bar\phi$ as two-component Grassmann variables with the components $\chi_1$ and $\chi_2$ and $\phi_1$ and $\phi_2$ respectively, just like the ones defined in Sec. 4.2. The *generalized partition function* $\mathcal{Z}_{\bar J,J}^{(n,\lambda)}$ is defined as a generalization of Eq. 31

$$\mathcal{Z}_{\bar J,J}^{(n,\lambda)}(\mathbf{A}) \equiv \int_{\chi\bar\chi} \exp\mathcal{S}_{\mathbf{A}}^{(n,\lambda)}\left\{\bar J,J,\bar\chi,\chi\right\}\,, \tag{B.1}$$

where $J \equiv \left\{\left\{J_j^{(r)}\right\}_{j=1}^l\right\}_{r=1}^n$ and $\bar J \equiv \left\{\left\{\bar J_j^{(r)}\right\}_{j=1}^l\right\}_{r=1}^n$ are external Grassmann sources, and $\mathcal{S}_{\mathbf{A}}^{(n,\lambda)}\left\{\bar\chi,\chi\right\}$ is a generalization of Eq. 28

$$\mathcal{S}_{\mathbf{A}}^{(n,\lambda)}\left\{\bar J,J,\bar\chi,\chi\right\} \equiv \bar\chi\mathbb{A}\chi + \frac{\lambda}{n!}\sum_{j=1}^l\left(\sum_{r=1}^n\bar\chi_j^{(r)}\chi_j^{(r)}\right)^n + \sum_j\left[\bar\chi_jJ_j+\bar J_j\chi_j\right]\,, \tag{B.2}$$

where $\bar\chi_jJ_j \equiv \sum_{r=1}^n\bar\chi_j^{(r)}J_j^{(r)}$, and a similar expression for the other quantities. Note that

$$M^{(n)}(\mathbf{A}) = \mathcal{Z}_{\bar J,J=0}^{(n,\lambda=1)}(\mathbf{A})\,. \tag{B.3}$$

By utilizing a basic Grassmann identity, the Sourlas transformation establishes an identity for Eq. B.1 when $n=2$. To accomplish this, we use the following identity, which holds for any two-component Grassmann variable $\chi = (\chi^{(1)},\chi^{(2)})^T$ and $\bar\chi = (\bar\chi^{(1)},\bar\chi^{(2)})$, and complex numbers $m$ and $u$:

$$\int_{\phi,\bar\phi} \exp\left[\frac{u}{2}\left(\bar\phi\phi\right)^2 + m\bar\phi\phi + \bar\phi\chi + \bar\chi\phi\right] = (u+m^2)\exp\left[-\frac{m}{u+m^2}\bar\chi\chi + \frac{u}{2(u+m^2)^2}\left(\bar\chi\chi\right)^2\right]\,. \tag{B.4}$$

This identity can be proved by expanding both sides and using the identities of Grassmann integrals. Then by inverting Eq. B.4, one proves the following identity

$$\exp\left[\frac{\lambda}{2}\left(\bar\chi\chi\right)^2\right] = (u+m^2)^{-1}\exp\left[\frac{m}{u+m^2}\bar\chi\chi\right]\int_{\phi,\bar\phi}\exp\left[\frac{u}{2}\left(\bar\phi\phi\right)^2 + m\bar\phi\phi + \bar\phi\chi + \bar\chi\phi\right]\,, \tag{B.5}$$

where the relation between $m$ and $u$ and $\lambda$ is

$$\lambda = \frac{u}{(u+m^2)^2}\,. \tag{B.6}$$

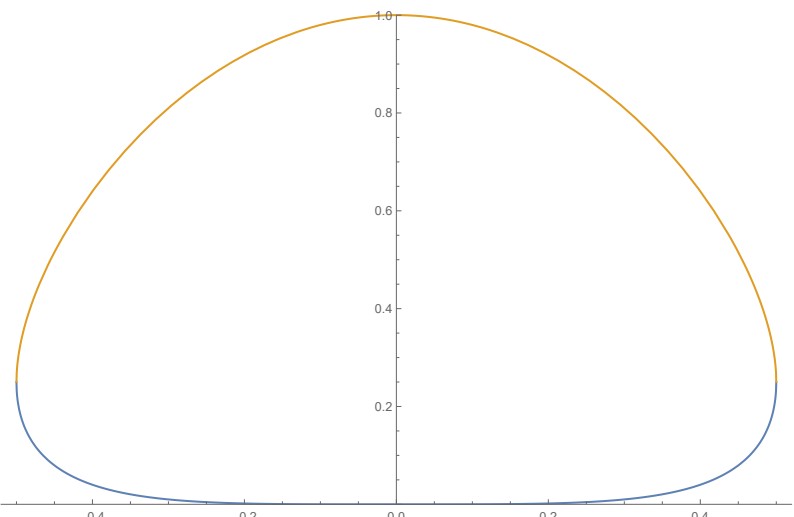

Figure 8: $u$ in terms of $m$ for $\lambda = 1$. The upper (lower) branch is for $u^\pm = \frac{1}{2} - m^2 \pm \sqrt{\frac{1}{4} - m^2}$, showing that $u$ can be chosen arbitrarily small.

This relation is graphically shown in Fig. 8 for $\lambda = 1$ with two branches in different colors ($u^\pm(\lambda)$). Using this identity for all $\phi_i$s, one ends up with the following equation [39]

$$\mathcal{Z}_{\bar{J},J}^{(n=2,\lambda)}(\mathbf{A}) = \prod_j \left(u_j + m_j^2\right)^{-1} \int_{\chi,\bar{\chi},\phi,\bar{\phi}} \exp\Big\{\bar{\chi}\mathbb{A}'\chi + \sum_j \Big[\frac{u_j}{2}\left(\bar{\phi}_j\phi_j\right)^2 + \bar{\chi}_j J_j + \bar{J}_j\chi_j \tag{B.7}$$
$$+ m_j\bar{\phi}_j\phi_j + \bar{\phi}_j\chi_j + \bar{\chi}_j\phi_j\Big]\Big\},$$

where the relationship between each $u_j$ and $m_j$ is now determined individually via the Eq. B.6, i.e. $\lambda \equiv \frac{u_j}{(u_j+m_j^2)^2}$, and $\mathbb{A}'_{ij} \equiv \mathbb{A}_{ij} + \delta_{ij}\frac{m_j}{u_j+m_j^2}$. Now, doing the following Grassmann integral over $\chi$ and $\bar{\chi}$:

$$\int_{\chi,\bar{\chi}} \exp\left[\bar{\chi}\mathbb{A}'\chi + \sum_j \Big[\bar{\chi}_j\left(J_j + \phi_j\right) + \left(\bar{J}_j + \bar{\phi}_j\right)\chi_j\Big]\right] = \det\left[\mathbb{A}'\right]\exp\Big[-\left(\bar{J}_i + \bar{\phi}_i\right) \tag{B.8}$$
$$\times\left[\left(\mathbb{A}'\right)^{-1}\right]_{ij}\left(J_j + \phi_j\right)\Big],$$

results in the following duality of the partition functions

$$\mathcal{Z}_{\bar{J},J}^{(2,\lambda)}(\mathbf{A}) = c_l e^{-\bar{J}\left(\mathbb{A}'\right)^{-1}J} Z_{\bar{J},J}^{(2,u)}(\mathbf{N}), \tag{B.9}$$

where the dual partition function is defined as

$$Z_{\bar{J},J}^{(2,u)}(\mathbf{N}) \equiv \int_{\phi\bar{\phi}} \exp \mathcal{L}_{\mathbf{N}}^{(u)}\left\{\bar{J},J,\bar{\phi},\phi\right\}, \tag{B.10}$$

and the dual effective action is

$$\mathcal{L}_{\mathbf{N}}^{(u)}\left\{\bar{J},J,\bar{\phi},\phi\right\} \equiv \bar{\phi}\mathbb{N}\phi + \sum_j\Big[\frac{u_j}{2}\left(\bar{\phi}_j\phi_j\right)^2\Big] - \sum_{ij}\Big[\bar{\phi}_i\left[\left(\mathbb{A}'\right)^{-1}\right]_{ij}J_j - \bar{J}_i\left[\left(\mathbb{A}'\right)^{-1}\right]_{ij}\phi_j\Big]. \tag{B.11}$$

In these relations we defined

$$\mathbb{N} \equiv \mathbb{M} - \left[\left(\mathbb{A}'\right)^{-1}\right], \quad c_l = \prod_{j=1}^l \left[\left(u_j + m_j^2\right)^{-1}\right] \det \mathbb{A}', \tag{B.12}$$

where $\mathbb{M} \equiv \begin{pmatrix} \mathbf{M} & 0 \\ 0 & \mathbf{M} \end{pmatrix}$, and $\mathbf{M}_{ij} \equiv m_i \delta_{ij}$. This implies that $\mathbb{N} \equiv \begin{pmatrix} \mathbf{N} & 0 \\ 0 & \mathbf{N} \end{pmatrix}$, where $\mathbf{N} \equiv \mathbf{M} + \mathbf{A}'^{-1}$.
For the case $\bar{J} = J = 0$, we simplify the notation by introducing

$$\mathcal{L}_{\mathbf{N}}^{(u)}\{\bar{\phi}, \phi\} \equiv \mathcal{L}_{\mathbf{N}}^{(u)}\{\bar{J} = 0, J = 0, \bar{\phi}, \phi\}, \qquad Z_u^{(n)}(\mathbf{A}) \equiv Z_{\bar{J},J=0}^{(n,u)}(\mathbf{N}), \qquad \mathcal{Z}_{\lambda}^{(n)}(\mathbf{A}) \equiv \mathcal{Z}_{\bar{J},J=0}^{(n,\lambda)}(\mathbf{N}). \tag{B.13}$$

For the case of interest, i.e. $\lambda = 1$ and $J = \bar{J} = 0$, we have (see Eqs. B.3 and B.9)

$$M^{(2)}(\mathbf{A}) = c_l M_u^{(2)}(\mathbf{N}), \quad \text{where} \quad M_u^{(2)}(\mathbf{N}) \equiv \int_{\phi,\bar{\phi}} \exp \mathcal{L}_{\mathbf{N}}^{(u)}\{\bar{\phi}, \phi\}. \tag{B.14}$$

It is instructive to expand this relation in terms of $u$. For the rest of this section we consider the case $u_j$ is independent of $j$, i.e. $u_j \equiv u$ for all $j \in \{1, 2, ..., l\}$. Expanding the interaction term in Eq. B.14 we find

$$\begin{aligned} M^{(2)}(\mathbf{A}) &= c_l \left[ (\det \mathbf{N})^2 + u \sum_j (\det \mathbf{N}_j)^2 + u^2 \sum_{j_1 > j_2} (\det \mathbf{N}_{j_1 j_2})^2 + ... + 1 \right] \\ &= c_l \sum_I u^{n_I} (\det \mathbf{N}_I)^2, \end{aligned} \tag{B.15}$$

where $n_I \equiv |I|$ is the number of elements of the set $I$. The two-point functions (see Eq. 47) are then expressed in terms of the determinants of $\mathbf{N}$ as follows:

$$\left\langle \bar{\phi}_j^{(1)} \phi_j^{(1)} \right\rangle_u = \frac{\sum_I u^{n_I} (\det \mathbf{N}_{I,j})(\det \mathbf{N}_I)}{\sum_I u^{n_I} (\det \mathbf{N}_I)^2}, \tag{B.16}$$

where $\mathbf{N}_I$ and $\mathbf{N}_{I,j}$ are defined in accordance with Eq. 47.

## C  The partition function of the Hubbard model

In this section we map the partition function of the Hubbard model to the SPPM problem. To this end, we divide the trace in Eq. 18 into $N$ imaginary time slices and then incorporate Berezin-Grassmann integration in each time slice, resulting in the following expression (see Ref. [38] for the details)

$$Z^{\text{Hub}}(L) = \int_{\psi\bar{\psi}} \exp\left[ S^{\text{Hub}}(\bar{\psi}, \psi) \right], \tag{C.1}$$

where the Hubbard discrete action is defined as ($\epsilon \equiv \frac{\beta}{N}$, and $N$ is the number of imaginary time slices)

$$\begin{aligned} S^{\text{Hub}}(\bar{\psi}, \psi) &= \sum_{k=2}^{N} \left[ \sum_{i=1}^{L} \sum_{\sigma} \bar{\psi}_{ik,\sigma} \left( -\psi_{ik,\sigma} + \psi_{ik-1,\sigma} + \mu\epsilon\psi_{ik-1,\sigma} \right) + \epsilon H\left( \bar{\psi}_{ik,\sigma}, \psi_{ik-1,\sigma} \right) \right] \\ &\quad + \sum_{i=1}^{L} \sum_{\sigma} \left[ -\bar{\psi}_{i1,\sigma} \left( \psi_{i1,\sigma} + \psi_{iN,\sigma} + \mu\epsilon\psi_{iN,\sigma} \right) + \epsilon H\left( \bar{\psi}_{i1,\sigma}, -\psi_{iN,\sigma} \right) \right], \end{aligned} \tag{C.2}$$

and the Hubbard discrete Hamiltonian for each time slice is defined as

$$H\left( \bar{\psi}_{ik,\sigma}, \psi_{ik-1,\sigma} \right) \equiv t \sum_{\langle i,j \rangle, \sigma} \left( \bar{\psi}_{ik,\sigma} \psi_{jk-1,\sigma} + \bar{\psi}_{jk,\sigma} \psi_{ik-1,\sigma} \right) - \frac{U}{2} \sum_{i=1}^{L} \left( \sum_{\sigma} \bar{\psi}_{ik,\sigma} \psi_{ik-1,\sigma} \right)^2. \tag{C.3}$$

In these equations $i \in [1, 2, ..., L]$ runs over lattice sites, $k$ runs over the discrete imaginary time, and $\sigma = \pm 1$ shows the spin. One can simplify the notation by defining $\vec{\psi}_\sigma \equiv \left\{ \{ \psi_{ik,\sigma} \}_{i=1}^L \right\}_{k=1}^N$ the size of which is $l \equiv LN$, and also $\Psi \equiv \begin{pmatrix} \vec{\psi}_\uparrow \\ \vec{\psi}_\downarrow \end{pmatrix}$ (and the same for $\bar{\Psi}$), so that $Z^{\mathrm{Hub}}(L) = \lim_{N \to \infty} Z_N^{\mathrm{Hub}}(L)$, and

$$Z_N^{\mathrm{Hub}}(L) = \int_{\Psi\bar{\Psi}} \exp\left( \bar{\Psi} \mathbb{S}^{\mathrm{TB}} \Psi - \frac{U\epsilon}{2} \sum_{k=2}^N \sum_{i=1}^L \left( \sum_\sigma \bar{\psi}_{ik,\sigma} \psi_{ik-1,\sigma} \right)^2 - \frac{U\epsilon}{2} \sum_{i=1}^L \left( \sum_\sigma \bar{\psi}_{i1,\sigma} \psi_{iN,\sigma} \right)^2 \right),$$
(C.4)

where $\mathbb{S}^{\mathrm{TB}} \equiv \begin{pmatrix} \mathbf{S}^{\mathrm{TB}} & 0 \\ 0 & \mathbf{S}^{\mathrm{TB}} \end{pmatrix}$. In this relation "TB" stands for "tight binding" limit and

$$\left( \mathbf{S}^{\mathrm{TB}} \right)_{jk,j'k'} \equiv \begin{cases} -\delta_{jj'}\delta_{kk'} + \delta_{kk'+1}\left[ (1+\mu\epsilon)\delta_{jj'} + t\epsilon(\delta_{jj'+1} + \delta_{jj'-1}) \right], & k \geq 2, k' < N, \\ -\delta_{jj'}\delta_{k1}\delta_{k'1} + \delta_{k1}\delta_{k'N}\left[ -(1+\mu\epsilon)\delta_{jj'} - t\epsilon(\delta_{jj'+1} + \delta_{jj'-1}) \right], & \text{otherwise}. \end{cases}$$
(C.5)

Defining $(\mathbf{B})_{jj'} \equiv (1+\mu\epsilon)\delta_{jj'} + t\epsilon \left( \delta_{j,j'+1} + \delta_{j+1,j'} \right)$, which is a circulant matrix, one finds

$$\mathbf{S}^{\mathrm{TB}} = \begin{pmatrix} -\mathbf{I} & 0 & 0 & 0 & \dots & -\mathbf{B} \\ \mathbf{B} & -\mathbf{I} & 0 & 0 & \dots & 0 \\ 0 & \mathbf{B} & -\mathbf{I} & 0 & \dots & 0 \\ . & & & & & . \\ . & & & & & . \\ . & & & & & 0 \\ 0 & 0 & 0 & \dots & \mathbf{B} & -\mathbf{I} \end{pmatrix}_{N \times N},$$

$$\mathbf{B} = \begin{pmatrix} 1+\mu\epsilon & t\epsilon & 0 & 0 & \dots & t\epsilon \\ t\epsilon & 1+\mu\epsilon & t\epsilon & 0 & \dots & 0 \\ 0 & t\epsilon & 1+\mu\epsilon & t\epsilon & \dots & 0 \\ . & & & & & . \\ . & & & & & . \\ . & & & & & t\epsilon \\ t\epsilon & 0 & 0 & \dots & t\epsilon & 1+\mu\epsilon \end{pmatrix}_{L \times L}.$$
(C.6)

To make a connection to the SPPM problem, we define a transformation $\bar{\phi}_{ik,\sigma} = \bar{\psi}_{ik,\sigma}$, and $\phi_{ik,\sigma} = \psi_{ik-1,\sigma}$, where $m = 1, 2$ is spin degrees of freedom, and $k = 2, ..., N$ accounts for time slices, and $i = 1, ..., L$ is the space index. For a consistent notation we define $\bar{\phi}_{i1,\sigma} = \bar{\psi}_{i1,\sigma}$ and $\phi_{i1,\sigma} = \psi_{iN,\sigma}$ which results in

$$\sum_{kk'ii'} \bar{\psi}_{ik,\sigma} \left( \mathbf{S}^{\mathrm{TB}} \right)_{ik,i'k'} \psi_{i'k',\sigma} = \sum_{kk'ii'} \bar{\phi}_{ik,\sigma} \left( \mathbf{A}_{\mathrm{Hub}} \right)_{ik,i'k'} \phi_{i'k',\sigma},$$
(C.7)

where

$$\mathbf{A}_{\mathrm{Hub}} = \begin{pmatrix} -\mathbf{B} & -\mathbf{I} & 0 & 0 & \dots & 0 \\ 0 & \mathbf{B} & -\mathbf{I} & 0 & \dots & 0 \\ 0 & 0 & \mathbf{B} & -\mathbf{I} & \dots & 0 \\ . & & & & & . \\ . & & & & & . \\ . & & & & & -\mathbf{I} \\ -\mathbf{I} & 0 & 0 & \dots & 0 & \mathbf{B} \end{pmatrix}_{N \times N}.$$
(C.8)

Then it is easily verified that

$$Z_N^{\mathrm{Hub}}(L) = \int_{\Phi\bar{\Phi}} \exp\left( \bar{\Phi} \mathbb{A}_{\mathrm{Hub}} \Phi - \frac{U\epsilon}{2} \sum_{k=1}^N \sum_{i=1}^L \left( \sum_\sigma \bar{\phi}_{ik,\sigma} \phi_{ik,\sigma} \right)^2 \right),$$
(C.9)

where $\vec{\phi}_\sigma \equiv \left\{\{\phi_{ik,\sigma}\}_{i=1}^L\right\}_{k=1}^N$ the size of which is $l \equiv LN$, and also $\Phi \equiv \begin{pmatrix} \vec{\phi}_\uparrow \\ \vec{\phi}_\downarrow \end{pmatrix}$ (and the same

for $\bar{\Phi}$), and $\mathbb{A}_{\text{Hub}} \equiv \begin{pmatrix} \mathbf{A}_{\text{Hub}} & 0 \\ 0 & \mathbf{A}_{\text{Hub}} \end{pmatrix}$. Before mapping this problem to SPPM, we test it for the

case $L = 1$, for which $\mathbf{B}$ is $1 \times 1$ with the element $b = 1 + \mu\epsilon$. In this case one may use the standard identities for the minors of $\mathbf{A}$ matrix, namely

$$
\begin{aligned}
\lim_{N\to\infty} \det \mathbf{A}_{\text{Hub}} &= -\lim_{N\to\infty}(1 + b^N) = 1 + e^{\beta\mu}, \\
\det(\mathbf{A}_{\text{Hub}})_{k_1^c} &= (2\delta_{k_1;1} - 1)b^{N-1}, \\
\det(\mathbf{A}_{\text{Hub}})_{k_1^c,k_2^c} &= (2\delta_{k_1,k_2;1} - 1)b^{N-2}, & k_2 \neq k_1, \\
\det(\mathbf{A}_{\text{Hub}})_{k_1^c,k_2^c,k_3^c} &= (2\delta_{k_1,k_2,k_3;1} - 1)b^{N-3}, & k_3 \neq k_2 \neq k_1, \dots,
\end{aligned}
\tag{C.10}
$$

where the subscripts $k_i^c$ shows that the rows and columns $k_i$ are removed. These determinants are involved in the series expansion of Eq. C.9 in terms of the $U$. Each expansion term generates a minor of the $\mathbb{A}_{\text{hub}}$ with some rows and columns removed. Noting that there are $\binom{N}{n}$ (binomial coefficient) possible ways for removing $n$ rows and columns from a $N \times N$ matrix (to calculate the principal minors), one can do the summation, which eventually gives

$$
\begin{aligned}
Z^{\text{Hub}}(L = 1) &= \lim_{N\to\infty} Z_N^{\text{Hub}}(L = 1) = \lim_{N\to\infty} \sum_{I_n}(-\epsilon U)^n(\det(\mathbf{A}_{\text{Hub}})_{I_n})^2 \\
&= (1 + e^{\beta\mu})^2 + \lim_{N\to\infty} \sum_{n=1}^N \binom{N}{n} b^{2(N-n)}(-\epsilon U)^n = 1 + 2e^{\beta\mu} + e^{2\beta\mu}e^{-\beta U},
\end{aligned}
\tag{C.11}
$$

as expected from a direct calculation. For a generic value of $L$, the Eq. C.9 can be easily mapped to a SPPM problem using the duality transformation reviewed in Sec. 4.2 and B. The result is

$$
Z_N^{\text{Hub}}(L) = \left(1 + m^2\right)^{-l} \det\left[\mathbb{A}_{\text{Hub}} + \frac{m}{m^2+1}\mathbb{I}\right] \int_{\Psi\bar{\Psi}} \exp\left(\bar{\Psi}\mathbb{N}_{\text{Hub}}\Psi + \frac{1}{2}\sum_{ik}\left(\sum_\sigma \bar{\psi}_{ik}\psi_{ik}\right)^2\right), \tag{C.12}
$$

where $m^2 = -1 \pm \frac{1}{\sqrt{-U\epsilon}}$, and

$$
\mathbb{N}_{\text{Hub}} \equiv m\mathbb{I} - \left(\mathbb{A}_{\text{Hub}} + \frac{m}{1+m^2}\mathbb{I}\right)^{-1} = \begin{pmatrix} m\mathbf{I} - \left(\mathbf{A}_{\text{Hub}} + \frac{m}{m^2+1}\mathbf{I}\right)^{-1} & 0 \\ 0 & m\mathbf{I} - \left(\mathbf{A}_{\text{Hub}} + \frac{m}{m^2+1}\mathbf{I}\right)^{-1} \end{pmatrix}. \tag{C.13}
$$

Using the Berezin-Grassmann integral representation described in Sec. 4 (see also Appendix A) one can easily verify that

$$
Z_N^{\text{Hub}}(L) = \left(1 + m^2\right)^{-l} \det\left(\mathbf{A}_{\text{Hub}} + \frac{m}{m^2+1}\mathbf{I}\right)^2 M^{(2)}\left(m\mathbf{I} - \left(\mathbf{A}_{\text{Hub}} + \frac{m}{m^2+1}\mathbf{I}\right)^{-1}\right). \tag{C.14}
$$

An equivalent expression is obtained using the Hubbard-Stratonovic transformation. Using Eq. 32, one can write the partition function as

$$
Z_N^{\text{Hub}} = \frac{1}{2^l(m^2+1)^l}\sum_S \left(\det\left[\mathbf{A}_{\text{Hub}}^{(S)}\right]\right)^2, \tag{C.15}
$$

where $\sum_S \equiv \sum_{\left\{\{s_i^{(k)}\}_{i=1}^K\right\}_{k=1}^L}$ is the sum over auxiliary spins, and

$$
\mathbf{A}_{\text{Hub}}^{(S)} \equiv (m\mathbf{I}_l + \mathbf{S})\left(\mathbf{A}_{\text{Hub}} + \frac{m}{m^2+1}\mathbf{I}_l\right) - \mathbf{I}_l. \tag{C.16}
$$

In the above equation $\mathbf{S} \equiv \mathrm{diag}\{\mathbf{s}_1, \mathbf{s}_2, ..., \mathbf{s}_N\}$ is a block diagonal matrix, and $\mathbf{s}_i \equiv \mathrm{diag}\left\{s_i^{(1)}, ..., s_i^{(L)}\right\}$, and $\mathbf{I}_l$ is an identity matrix of dimension $l$. One can easily show that

$$
\mathbf{A}_{\mathrm{Hub}}^{(S)} = \begin{bmatrix}
\mathbf{f}_-(\mathbf{B}, \mathbf{s}_1) & -(m\mathbf{I}_L + \mathbf{s}_1) & 0 & 0 & \cdots & 0 & 0 \\
0 & \mathbf{f}_+(\mathbf{B}, \mathbf{s}_2) & -(m\mathbf{I}_L + \mathbf{s}_2) & 0 & \cdots & 0 & 0 \\
. & & & & & & \\
. & & & & & & \\
. & & & & & & \\
0 & 0 & 0 & 0 & \cdots & \mathbf{f}_+(\mathbf{B}, \mathbf{s}_{N-1}) & -(m\mathbf{I}_L + \mathbf{s}_{N-1}) \\
-(m\mathbf{I}_L + \mathbf{s}_N) & 0 & 0 & 0 & \cdots & 0 & \mathbf{f}_+(\mathbf{B}, \mathbf{s}_N)
\end{bmatrix}, \quad \text{(C.17)}
$$

where $\mathbf{f}_\pm(\mathbf{B}, \mathbf{s}_i) \equiv (m\mathbf{I}_L + \mathbf{s}_i)\left(\pm\mathbf{B} + \frac{m}{m^2+1}\mathbf{I}_L\right) - \mathbf{I}_L$. It is useful to write $\det \mathbf{A}_{\mathrm{Hub}}^{(S)}$ as follows

$$
\det \mathbf{A}_{\mathrm{Hub}}^{(S)} = \prod_{i=1}^{N}\prod_{j=1}^{L}\left(m + s_i^{(j)}\right) \times \det\left\{\left[\left(\mathbf{B} + \frac{m}{m^2+1}\mathbf{I}_L\right) - (m\mathbf{I}_L + \mathbf{s}_N)^{-1}\right]\right.
$$
$$
\times \left[\left(\mathbf{B} + \frac{m}{m^2+1}\mathbf{I}_L\right) - (m\mathbf{I}_L + \mathbf{s}_{N-1})^{-1}\right] \quad \text{(C.18)}
$$
$$
\left. \cdots \left[\left(-\mathbf{B} + \frac{m}{m^2+1}\mathbf{I}_L\right) - (m\mathbf{I}_L + \mathbf{s}_1)^{-1}\right] - \mathbf{I}_L\right\}.
$$

It is instructive to obtain the partition function in two limits: the single site Hubbard model $L = 1$, and the atomic limit $t = 0$ and the atomic limit $t = 0$.

For the single site Hubbard model, $\mathbf{B}$ is a one by one matrix with the element $B \equiv 1 + \mu\epsilon = 1 + \frac{\mu\beta}{N}$. One can readily find the determinant of $\mathbf{A}_{\mathrm{Hub}^{(S)}}$ as follows

$$
\det\left(\mathbf{A}_{\mathrm{Hub}^{(S)}}\right) = -\prod_{i=1}^{N}(m + s_i) + \prod_{i=1}^{N}\left[(m + s_i)\left(f_i B + \frac{m}{m^2+1}\right) - 1\right], \quad \text{(C.19)}
$$

where $f_i = 1 - 2\delta_{i,1}$, which gives

$$
\left[\det\left(\mathbf{A}_{\mathrm{Hub}^{(S)}}\right)\right]^2 = \prod_{i=1}^{N}[(m + s_i)]^2 + \prod_{i=1}^{N}\left(\left[(m + s_i)\left(f_i B + \frac{m}{m^2+1}\right) - 1\right]\right)^2
$$
$$
- 2\prod_{i=1}^{N}\left[(m + s_i)^2\left(f_i B + \frac{m}{m^2+1}\right) - (m + s_i)\right]. \quad \text{(C.20)}
$$

Summing over auxiliary spins, and keeping the terms proportional to the even powers of spins, we find

$$
\sum_{S}\left[\det\left(\mathbf{A}_{\mathrm{Hub}^{(S)}}\right)\right]^2 = 2^N\left[(m^2 + 1)\right]^N + 2^N\left[(mx_1 - 1)^2 + x_1^2\right]\left[(mx - 1)^2 + x^2\right]^{N-1}
$$
$$
- 2^{N+1}\left[(m^2 + 1)x_1 - m\right]\left[(m^2 + 1)x - m\right]^{N-1}, \quad \text{(C.21)}
$$

where $x_1 \equiv -B + \frac{m}{m^2+1}$, and $x \equiv B + \frac{m}{m^2+1}$. Noting that

$$
\frac{(mx - 1)^2 + x^2}{m^2 + 1} = 1 + \frac{1}{(m^2 + 1)^2} + \frac{2\beta\mu}{N} + O\left(1/N^2\right), \qquad x - \frac{m}{m^2+1} = 1 + \frac{\beta\mu}{N}, \quad \text{(C.22)}
$$

and $(m^2 + 1)^2 = -(\beta U/N)^{-1}$, and using Eq. C.15 one finally finds (SS stands for single site)

$$
Z_{\mathrm{SS}}^{\mathrm{Hub}} = 1 + 2e^{\beta\mu} + e^{2\beta\mu}e^{-\beta U}, \quad \text{(C.23)}
$$

which is the known result.

For $t = 0$, the matrix $\mathbf{B}$ is diagonal $\mathbf{B} = (1 + \mu\beta/N)\mathbf{I}_L$ so that Eq. C.18 gives us

$$\det \mathbf{A}_{\text{Hub}}^{(S)}(t=0) = \prod_{j=1}^{L}\left\{\prod_{i=1}^{N}\left[\left(m+s_i^{(j)}\right)\left(f_iB + \frac{m}{m^2+1}\right)-1\right] - \prod_{i=1}^{N}\left(m+s_i^{(j)}\right)\right\}. \tag{C.24}$$

The expression inside the parenthesis is the same as Eq. C.19, giving rise to

$$Z^{\text{Hub}}(t=0) = \left(Z_{\text{SS}}^{\text{Hub}}\right)^L. \tag{C.25}$$

## D  Details of the mean field theory

The objective of this section is to provide a detailed explanation of our mean field theory starting from the effective action Eq. 89.

### D.1  MF theory; the main scheme

The effective action Eq. 89 is not block-diagonal and also contains the terms like $\chi^{(1)}\chi^{(2)}$ and $\bar{\chi}^{(1)}\bar{\chi}^{(2)}$. Using the following Bogoliobov transformation

$$\begin{cases}\chi_j^{(1)} = \frac{1}{\sqrt{2}}\left[\psi_j^{(1)}+\bar{\psi}_j^{(2)}\right], & \bar{\chi}_j^{(1)} = \frac{1}{\sqrt{2}}\left[\bar{\psi}_j^{(1)}+\psi_j^{(2)}\right], \\ \chi_j^{(2)} = \frac{1}{\sqrt{2}}\left[\psi_j^{(1)}-\bar{\psi}_j^{(2)}\right], & \bar{\chi}_j^{(2)} = \frac{1}{\sqrt{2}}\left[\bar{\psi}_j^{(1)}-\psi_j^{(2)}\right],\end{cases} \tag{D.1}$$

one easily finds

$$S_{\text{MF}}\left\{\bar{\psi},\psi\right\} = \bar{\psi}\,\mathbb{S}_{\text{MF}}(\mathbf{A})\psi\,, \qquad \mathbb{S}_{\text{MF}}(\mathbf{A}) = \begin{pmatrix} \mathbf{A}+(\Delta_1-\Delta_2)\mathbf{I} & \Delta_3\mathbf{I} \\ -\Delta_3\mathbf{I} & -\mathbf{A}^T-(\Delta_1+\Delta_2)\mathbf{I} \end{pmatrix}, \tag{D.2}$$

where $\psi^T \equiv \left(\psi^{(1)},\psi^{(2)}\right)$ and $\bar{\psi} \equiv \left(\bar{\psi}^{(1)},\bar{\psi}^{(2)}\right)$. Substituting this equation into Eq. 85, one finds

$$M^{(2)}(\mathbf{A}) = e^{-l\Delta_4}\det\mathbb{S}_{\text{MF}}(\mathbf{A})\,. \tag{D.3}$$

The self-consistent two-point functions are then expressed in terms of two-point functions of $\psi$ and $\bar{\psi}$ as follows

$$\Delta_1 = \frac{1}{2}\left\langle\left[\bar{\psi}^{(1)}+\psi^{(2)}\right]\left[\psi^{(1)}+\bar{\psi}^{(2)}\right]\right\rangle, \tag{D.4a}$$

$$\Delta_2 = \frac{1}{2}\left\langle\left[\bar{\psi}^{(1)}+\psi^{(2)}\right]\left[\psi^{(1)}-\bar{\psi}^{(2)}\right]\right\rangle, \tag{D.4b}$$

$$\Delta_3 = \frac{1}{2}\left\langle\left[\bar{\psi}^{(1)}+\psi^{(2)}\right]\left[\bar{\psi}^{(1)}-\psi^{(2)}\right]\right\rangle, \tag{D.4c}$$

in which the following equations hold due to the explicit symmetries of $\mathbb{S}_{\text{MF}}(\mathbf{A})$:

$$\left\langle\bar{\psi}_j^{(r)}\bar{\psi}_j^{(r')}\right\rangle = 0\,, \qquad \left\langle\psi_j^{(r)}\psi_j^{(r')}\right\rangle = 0\,, \quad r,r' = 1,2\,. \tag{D.5}$$

The other two-point functions are calculated using $\mathbb{S}_{\text{MF}}(\mathbf{A})$ as follows:

$$\left\langle\bar{\psi}_j^{(1)}\psi_j^{(1)}\right\rangle = e^{l\Delta_4^{\text{MF}}}\left(M^{(2)}(\mathbf{A})\right)^{-1}\int_{\bar{\psi},\psi}\bar{\psi}_j^{(1)}\psi_j^{(1)}e^{\bar{\psi}\mathbb{S}_{\text{MF}}(\mathbf{A})\psi}$$

$$= \frac{\det\mathbb{S}_{\text{MF}(1j)^c,(1j)^c}}{\det\mathbb{S}_{\text{MF}}} = \left[\mathbb{S}_{\text{MF}}^{-T}\right]_{(1j,1j)}, \tag{D.6}$$

where we defined $\mathbb{S}_{\mathrm{MF}(rj)(r'j)}$ as the matrix $\mathbb{S}_{\mathrm{MF}}(\mathbf{A})$ in which the rows and columns $j$ are removed from the block $r, r'$. The other functions are similarly found to be

$$\left\langle \bar{\psi}_j^{(2)} \psi_j^{(2)} \right\rangle = \frac{\det \mathbb{S}_{\mathrm{MF}(2j)^c(2j)^c}}{\det \mathbb{S}_{\mathrm{MF}}} = \left[ \mathbb{S}_{\mathrm{MF}}^{-T} \right]_{(2j,2j)}, \tag{D.7a}$$

$$\left\langle \bar{\psi}_j^{(1)} \psi_j^{(2)} \right\rangle = (-1)^l \frac{\det \mathbb{S}_{\mathrm{MF}(1j)^c(2j)^c}}{\det \mathbb{S}_{\mathrm{MF}}} = (-1)^l \left[ \mathbb{S}_{\mathrm{MF}}^{-T} \right]_{(1j,2j)}, \tag{D.7b}$$

$$\left\langle \bar{\psi}_j^{(2)} \psi_j^{(1)} \right\rangle = (-1)^l \frac{\det \mathbb{S}_{\mathrm{MF}(2j)^c(1j)^c}}{\det \mathbb{S}_{\mathrm{MF}}} = (-1)^l \left[ \mathbb{S}_{\mathrm{MF}}^{-T} \right]_{(2j,1j)}, \tag{D.7c}$$

where $\mathbb{O}^{-T}$ is defined as $\left( \mathbb{O}^{-1} \right)^T$. The self-consistent two-point functions of interest ($\Delta$ functions) are expressed as the liner combination of these functions:

$$\begin{aligned}
\Delta_1 &= \frac{1}{2} \left[ \left\langle \bar{\psi}_j^{(1)} \psi_j^{(1)} \right\rangle - \left\langle \bar{\psi}_j^{(2)} \psi_j^{(2)} \right\rangle \right], \\
\Delta_2 &= \frac{1}{2} \left[ \left\langle \bar{\psi}_j^{(1)} \psi_j^{(1)} \right\rangle + \left\langle \bar{\psi}_j^{(2)} \psi_j^{(2)} \right\rangle \right], \\
\Delta_3 &= - \left\langle \bar{\psi}_j^{(1)} \psi_j^{(2)} \right\rangle.
\end{aligned} \tag{D.8}$$

The inverse of $\mathbb{S}_{\mathrm{MF}}$ can be readily obtained using their circulant nature (note that $\mathbf{A}$ and $\mathbf{A}^T$ commute). Using Schur complement method and LDU decomposition one obtains

$$\begin{aligned}
\mathbb{S}_{\mathrm{MF}}(\mathbf{A}) = &\begin{pmatrix} \mathbf{I} & -\frac{\Delta_3}{\mathbf{A}^T + (\Delta_1 + \Delta_2)\mathbf{I}} \\ \mathbf{0} & \mathbf{I} \end{pmatrix} \begin{pmatrix} \mathbf{A} + (\Delta_1 - \Delta_2)\mathbf{I} - \frac{\Delta_3^2}{\mathbf{A}^T + (\Delta_1 + \Delta_2)\mathbf{I}} & \mathbf{0} \\ \mathbf{0} & -\mathbf{A}^T - (\Delta_1 + \Delta_2)\mathbf{I} \end{pmatrix} \\
&\times \begin{pmatrix} \mathbf{I} & \mathbf{0} \\ \frac{\Delta_3}{\mathbf{A}^T + (\Delta_1 + \Delta_2)\mathbf{I}} & \mathbf{I} \end{pmatrix}.
\end{aligned} \tag{D.9}$$

Using this equation, and defining $\mathbf{A}_{s,a} \equiv \frac{1}{2} \left( \mathbf{A} \pm \mathbf{A}^T \right)$, one finds by inspection that

$$\det \mathbb{S}_{\mathrm{MF}}(\mathbf{A}) = \det \left( -\mathbf{x} - 2\Delta_2 \mathbf{A}_a \right), \tag{D.10}$$

where

$$\begin{aligned}
\mathbf{x} &\equiv (\mathbf{A} + \Delta_1 \mathbf{I}) \left( \mathbf{A}^T + \Delta_1 \mathbf{I} \right) - \left( \Delta_2^2 + \Delta_3^2 \right) \mathbf{I} \\
&= \mathbf{A}\mathbf{A}^T + 2\Delta_1 \mathbf{A}_s + \Delta_4 \mathbf{I}.
\end{aligned} \tag{D.11}$$

Furthermore, one is able to calculate the inverse as

$$\mathbb{S}_{\mathrm{MF}}^{-1}(\mathbf{A}) = \begin{bmatrix} \frac{\mathbf{A}^T + (\Delta_1 + \Delta_2)\mathbf{I}}{\mathbf{x} + 2\Delta_2 \mathbf{A}_a} & \frac{\Delta_3 \mathbf{I}}{\mathbf{x} + 2\Delta_2 \mathbf{A}_a} \\ -\frac{\Delta_3 \mathbf{I}}{\mathbf{x} + 2\Delta_2 \mathbf{A}_a} & -\frac{\mathbf{A} + (\Delta_1 - \Delta_2)\mathbf{I}}{\mathbf{x} + 2\Delta_2 \mathbf{A}_a} \end{bmatrix}. \tag{D.12}$$

We adopted this notation (utilizing matrices in the denominators) because all the matrices within this equation mutually commute, allowing us to express $\frac{\mathbf{M}}{\mathbf{N}}$ as $\mathbf{N}^{-1}\mathbf{M} = \mathbf{M}\mathbf{N}^{-1}$ for any matrices $\mathbf{M}$ and $\mathbf{N}$ that commute. We derive the following matrix expressions for two-point functions:

$$\begin{aligned}
\left\langle \bar{\psi}_j^{(1)} \psi_j^{(1)} \right\rangle &= \left[ \frac{\mathbf{A}^T + (\Delta_1 + \Delta_2)\mathbf{I}}{\mathbf{x} + 2\Delta_2 \mathbf{A}_a} \right]_{j,j}^T = \left[ \frac{\mathbf{A}^T + (\Delta_1 + \Delta_2)\mathbf{I}}{\mathbf{x} + 2\Delta_2 \mathbf{A}_a} \right]_{j,j}, \\
\left\langle \bar{\psi}_j^{(2)} \psi_j^{(2)} \right\rangle &= -\left[ \frac{\mathbf{A} + (\Delta_1 - \Delta_2)\mathbf{I}}{\mathbf{x} + 2\Delta_2 \mathbf{A}_a} \right]_{j,j}^T = -\left[ \frac{\mathbf{A} + (\Delta_1 - \Delta_2)\mathbf{I}}{\mathbf{x} + 2\Delta_2 \mathbf{A}_a} \right]_{j,j}, \\
\left\langle \bar{\psi}_j^{(2)} \psi_j^{(1)} \right\rangle &= -\left\langle \bar{\psi}_j^{(1)} \psi_j^{(2)} \right\rangle = (-1)^l \left[ \frac{\Delta_3 \mathbf{I}}{\mathbf{x} + 2\Delta_2 \mathbf{A}_a} \right]_{j,j}^T = (-1)^l \left[ \frac{\Delta_3 \mathbf{I}}{\mathbf{x} + 2\Delta_2 \mathbf{A}_a} \right]_{j,j}.
\end{aligned} \tag{D.13}$$

The final result is the following set of self-consistent MF equations:

$$\Delta_1 = \left[ \frac{\mathbf{A}_s + \Delta_1 \mathbf{I}}{\mathbf{x} + 2\Delta_2 \mathbf{A}_a} \right]_{j,j}, \qquad \Delta_2 = \left[ \frac{-\mathbf{A}_a + \Delta_2 \mathbf{I}}{\mathbf{x} + 2\Delta_2 \mathbf{A}_a} \right]_{j,j}, \qquad \Delta_3 = \left[ \frac{\Delta_3 \mathbf{I}}{\mathbf{x} + 2\Delta_2 \mathbf{A}_a} \right]_{j,j}. \tag{D.14}$$

## D.2  Pfaffian formalism

In this part we re-formulate the MF equations in terms of Pfaffian. To this end, we define a 4-Grassmann variable

$$\zeta \equiv \begin{bmatrix} \bar{\chi} \\ \chi \end{bmatrix} \equiv \begin{bmatrix} \bar{\chi}^{(1)} \\ \bar{\chi}^{(2)} \\ \chi^{(1)} \\ \chi^{(2)} \end{bmatrix}. \tag{D.15}$$

Having defined this 4-Grassmann, we use the following property for any given $\mathbb{O}$ matrix

$$\begin{aligned}
\frac{1}{2} \begin{bmatrix} \bar{\chi}^T, \chi^T \end{bmatrix} \begin{bmatrix} 0 & \mathbb{O} \\ -\mathbb{O}^T & 0 \end{bmatrix} \begin{bmatrix} \bar{\chi} \\ \chi \end{bmatrix} &= -\frac{1}{2} \chi^T \mathbb{O}^T \bar{\chi} + \frac{1}{2} \bar{\chi}^T \mathbb{O} \chi \\
&= \frac{1}{2} \sum_{ij} \bar{\chi}_i \mathbb{O}_{ij} \chi_j + \frac{1}{2} \sum_{ij} \bar{\chi}_i \mathbb{O}_{ij} \chi_j = \bar{\chi} \mathbb{O} \chi.
\end{aligned} \tag{D.16}$$

Using this, one casts the Eq. 81 to

$$S_{\mathrm{MF}}\{\bar{\chi}, \chi\} \equiv \bar{\chi} \mathbb{O} \chi - \frac{1}{2} \chi \mathbb{I}_3 \chi - \frac{1}{2} \bar{\chi} \bar{\mathbb{I}}_3 \bar{\chi} \equiv -\frac{1}{2} \zeta^T \mathbb{S}_{\mathrm{MF}}^{\mathrm{Pf}} \zeta, \tag{D.17}$$

where

$$\mathbb{S}_{\mathrm{MF}}^{\mathrm{Pf}} = \begin{bmatrix} \bar{\mathbb{I}}_3 & -\mathbb{O} \\ \mathbb{O}^T & \mathbb{I}_3 \end{bmatrix}, \tag{D.18}$$

which is antisymmetric and $\mathbb{I}_3$ and $\mathbb{O}$ have been defined in Eq. 82. Using the Pfaffian identities for $\zeta$'s, we obtain

$$M_{\mathrm{MF}}^{(2)} = e^{-\sum_{j=1}^l \Delta_4(j)} \mathrm{Pf}\left[\mathbb{S}_{\mathrm{MF}}^{\mathrm{Pf}}\right]. \tag{D.19}$$

One additionally obtains the MF self-consistency equations as follows:

$$\begin{aligned}
\Delta_j^{(\bar{1}1)} \equiv \left\langle \bar{\chi}_j^{(1)} \chi_j^{(1)} \right\rangle = \left\langle \zeta_j^{(1)} \zeta_j^{(3)} \right\rangle &= \frac{\int_\zeta \zeta_j^{(1)} \zeta_j^{(3)} \exp\left[\frac{1}{2} \zeta^T \mathbb{S}_{\mathrm{MF}}^{\mathrm{Pf}} \zeta\right]}{\int_\zeta \exp\left[\frac{1}{2} \zeta^T \mathbb{S}_{\mathrm{MF}}^{\mathrm{Pf}} \zeta\right]} \\
&= \mathrm{Pf} \begin{bmatrix} \left(\mathbb{S}_{\mathrm{MF}}^{\mathrm{Pf}}\right)_{1j,1j}^{-T} & \left(\mathbb{S}_{\mathrm{MF}}^{\mathrm{Pf}}\right)_{1j,3j}^{-T} \\ \left(\mathbb{S}_{\mathrm{MF}}^{\mathrm{Pf}}\right)_{3j,1j}^{-T} & \left(\mathbb{S}_{\mathrm{MF}}^{\mathrm{Pf}}\right)_{3j,3j}^{-T} \end{bmatrix},
\end{aligned} \tag{D.20a}$$

$$\begin{aligned}
\Delta_j^{(\bar{1}2)} \equiv \left\langle \bar{\chi}_j^{(1)} \chi_j^{(2)} \right\rangle = \left\langle \zeta_j^{(1)} \zeta_j^{(4)} \right\rangle &= \frac{\int_\zeta \zeta_j^{(1)} \zeta_j^{(4)} \exp\left[\frac{1}{2} \zeta^T \mathbb{S}_{\mathrm{MF}}^{\mathrm{Pf}} \zeta\right]}{\int_\zeta \exp\left[\frac{1}{2} \zeta^T \mathbb{S}_{\mathrm{MF}}^{\mathrm{Pf}} \zeta\right]} \\
&= \mathrm{Pf} \begin{bmatrix} \left(\mathbb{S}_{\mathrm{MF}}^{\mathrm{Pf}}\right)_{1j,1j}^{-T} & \left(\mathbb{S}_{\mathrm{MF}}^{\mathrm{Pf}}\right)_{1j,4j}^{-T} \\ \left(\mathbb{S}_{\mathrm{MF}}^{\mathrm{Pf}}\right)_{4j,1j}^{-T} & \left(\mathbb{S}_{\mathrm{MF}}^{\mathrm{Pf}}\right)_{4j,4j}^{-T} \end{bmatrix},
\end{aligned} \tag{D.20b}$$

$$\begin{aligned}
\Delta_j^{(\bar{1}\bar{2})} \equiv \left\langle \bar{\chi}_j^{(1)} \bar{\chi}_j^{(2)} \right\rangle = \left\langle \zeta_j^{(1)} \zeta_j^{(2)} \right\rangle &= \frac{\int_\zeta \zeta_j^{(1)} \zeta_j^{(2)} \exp\left[\frac{1}{2} \zeta^T \mathbb{S}_{\mathrm{MF}}^{\mathrm{Pf}} \zeta\right]}{\int_\zeta \exp\left[\frac{1}{2} \zeta^T \mathbb{S}_{\mathrm{MF}}^{\mathrm{Pf}} \zeta\right]} \\
&= \mathrm{Pf} \begin{bmatrix} \left(\mathbb{S}_{\mathrm{MF}}^{\mathrm{Pf}}\right)_{1j,1j}^{-T} & \left(\mathbb{S}_{\mathrm{MF}}^{\mathrm{Pf}}\right)_{1j,2j}^{-T} \\ \left(\mathbb{S}_{\mathrm{MF}}^{\mathrm{Pf}}\right)_{2j,1j}^{-T} & \left(\mathbb{S}_{\mathrm{MF}}^{\mathrm{Pf}}\right)_{2j,2j}^{-T} \end{bmatrix}.
\end{aligned} \tag{D.20c}$$

In this formula, we used $\mathbb{M}_{ur,vr}$ (for a block matrix $\mathbb{M}$, and $u, v \in \{1, 2, 3, 4\}$) to show $(r, r)$ component of the $(u, v)$th sub-matrix, i.e. $\left(\mathbb{M}_{u,v}\right)_{l \times l}$. For the Pfaffian, according to the Eq. D.20, we should calculate the inverse of $\mathbb{S}_{\text{MF}}^{\text{Pf}}$. In this section we consider the case $\Delta$ functions are independent of $r$, and we restrict ourselves to the case where there are only three independent functions: $\Delta_1$, $\Delta_2$ and $\Delta_3$. When $\mathbf{A}$ is circulant, all of its functions are also circulant and commutative, allowing us to determine its inverse as follows

$$
\left(\mathbb{S}_{\text{MF}}^{\text{Pf}}\right)^{-1} = \begin{bmatrix} -\Delta_2\Delta_3\frac{\mathbf{A}_a}{\mathbf{y}} & \Delta_3\frac{\mathbf{x}}{\mathbf{y}} & \frac{\mathbf{xO}_1+\Delta_2^2\mathbf{A}_a}{\mathbf{y}} & \Delta_2\frac{\mathbf{x}+\mathbf{O}_1\mathbf{A}_a}{\mathbf{y}} \\ -\Delta_3\frac{\mathbf{x}}{\mathbf{y}} & \Delta_2\Delta_3\frac{\mathbf{A}_a}{\mathbf{y}} & \Delta_2\frac{\mathbf{x}+\mathbf{O}_1\mathbf{A}_a}{\mathbf{y}} & \frac{\mathbf{xO}_1+\Delta_2^2\mathbf{A}_a}{\mathbf{y}} \\ -\frac{\mathbf{xO}_1^T-\Delta_2^2\mathbf{A}_a}{\mathbf{y}} & -\Delta_2\frac{\mathbf{x}-\mathbf{O}_1^T\mathbf{A}_a}{\mathbf{y}} & \Delta_2\Delta_3\frac{\mathbf{A}_a}{\mathbf{y}} & \Delta_3\frac{\mathbf{x}}{\mathbf{y}} \\ -\Delta_2\frac{\mathbf{x}-\mathbf{O}_1^T\mathbf{A}_a}{\mathbf{y}} & -\frac{\mathbf{xO}_1^T-\Delta_2^2\mathbf{A}_a}{\mathbf{y}} & -\Delta_3\frac{\mathbf{x}}{\mathbf{y}} & -\Delta_2\Delta_3\frac{\mathbf{A}_a}{\mathbf{y}} \end{bmatrix} \equiv \begin{bmatrix} \mathbf{C} & \mathbf{D} \\ -\mathbf{D}^T & \mathbf{C}' \end{bmatrix}, \qquad \text{(D.21)}
$$

where $\mathbf{A}_a \equiv \frac{1}{2}\left[\mathbf{F} - \mathbf{F}^T\right]$, $\mathbf{O}_1 \equiv \mathbf{A} + \Delta_1\mathbf{I}$, $\mathbf{x} \equiv \mathbf{O}_1^T\mathbf{O}_1 - \left(\Delta_2^2 + \Delta_3^2\right)\mathbf{I}$, and $\mathbf{y} \equiv \mathbf{x}^2 - 4\Delta_2^2\mathbf{A}_a^2$. We also defined

$$
\mathbf{C} = \begin{bmatrix} \mathbf{C}_{11} & \mathbf{C}_{12} \\ -\mathbf{C}_{12} & -\mathbf{C}_{11} \end{bmatrix}, \qquad \mathbf{C}' = \begin{bmatrix} -\mathbf{C}_{11} & \mathbf{C}_{12} \\ -\mathbf{C}_{12} & \mathbf{C}_{11} \end{bmatrix}, \qquad \mathbf{D} = \begin{bmatrix} \mathbf{D}_{11} & \mathbf{D}_{12} \\ \mathbf{D}_{12} & \mathbf{D}_{11} \end{bmatrix}, \qquad \text{(D.22)}
$$

in which

$$
\mathbf{C}_{11} = -\Delta_2\Delta_3\frac{2\mathbf{A}_a}{\mathbf{y}}, \quad \mathbf{C}_{12} = \Delta_3\frac{\mathbf{x}}{\mathbf{y}}, \quad \mathbf{D}_{11} = \frac{\mathbf{xO}_1 + 2\Delta_2^2\mathbf{A}_a}{\mathbf{y}}, \quad \mathbf{D}_{12} = \Delta_2\frac{\mathbf{x} + 2\mathbf{O}_1\mathbf{A}_a}{\mathbf{y}}. \quad \text{(D.23)}
$$

Now we consider the consistency equations. According to Eq. D.20, for determining $\Delta_1$ we should consider the block corresponding to $I = \{1j, 3j\}$, so that

$$
\begin{aligned}
\Delta_1 &= \text{Pf}[\left(\mathbb{S}_{\text{MF}}^{\text{Pf}}\right)_{II}^{-T}] = \text{Pf}[-\left(\mathbb{S}_{\text{MF}}^{\text{Pf}}\right)_{II}^{-1}] \\
&= \text{Pf}\begin{bmatrix} -[\mathbf{C}_{11}]_{j,j} & -[\mathbf{D}_{11}]_{j,j} \\ [\mathbf{D}_{11}^T]_{j,j} & [\mathbf{C}_{11}]_{j,j} \end{bmatrix} = \text{Pf}\begin{bmatrix} 0 & -[\mathbf{D}_{11}]_{j,j} \\ [\mathbf{D}_{11}^T]_{j,j} & 0 \end{bmatrix} = [\mathbf{D}_{11}]_{j,j}.
\end{aligned} \qquad \text{(D.24)}
$$

In the other hand we have

$$
[\mathbf{D}_{11}]_{j,j} = \left[\frac{\mathbf{xO}_1 + 2\Delta_2^2\mathbf{A}_a}{\mathbf{y}}\right]_{j,j} = \left[\frac{\mathbf{xO}_1}{\mathbf{y}}\right]_{j,j} = \frac{1}{2}\left[\frac{\mathbf{xO}_1}{\mathbf{y}} + \left(\frac{\mathbf{xO}_1}{\mathbf{y}}\right)^T\right]_{j,j}. \qquad \text{(D.25)}
$$

One may be interested in writing this equation in a symmetric form, as follows

$$
\begin{aligned}
\frac{\mathbf{xO}_1}{\mathbf{y}} &= \frac{1}{2}\left[\frac{\mathbf{O}_1}{\mathbf{x} - 2\mathbf{A}_a\Delta_2} + \frac{\mathbf{O}_1}{\mathbf{x} + 2\mathbf{A}_a\Delta_2}\right], \\
\left(\frac{\mathbf{xO}_1}{\mathbf{y}}\right)^T &= \frac{1}{2}\left[\frac{\mathbf{O}_1^T}{\mathbf{x} + 2\mathbf{A}_a\Delta_2} + \frac{\mathbf{O}_1^T}{\mathbf{x} - 2\mathbf{A}_a\Delta_2}\right],
\end{aligned} \qquad \text{(D.26)}
$$

so that

$$
\Delta_1 = \frac{1}{4}\left[\frac{\mathbf{O}_1 + \mathbf{O}_1^T}{\mathbf{x} - 2\mathbf{A}_a\Delta_2}\right]_{j,j} + \frac{1}{4}\left[\frac{\mathbf{O}_1 + \mathbf{O}_1^T}{\mathbf{x} + 2\mathbf{A}_a\Delta_2}\right]_{j,j} = \frac{1}{2}\left[\frac{\mathbf{O}_1 + \mathbf{O}_1^T}{\mathbf{x} + 2\mathbf{A}_a\Delta_2}\right]_{j,j}, \qquad \text{(D.27)}
$$

which coincides with Eq. D.14. Now let us consider $\Delta_2$, for which we have to consider the block corresponding to $I = \{1r, 4r\}$, so that

$$
\begin{aligned}
\Delta_2 &= \text{Pf}[\left(\mathbb{S}_{\text{MF}}^{\text{Pf}}\right)_{II}^{-T}] = \text{Pf}[-\left(\mathbb{S}_{\text{MF}}^{\text{Pf}}\right)_{II}^{-1}] = \text{Pf}\begin{bmatrix} -[\mathbf{C}_{11}]_{j,j} & -[\mathbf{D}_{12}]_{j,j} \\ [\mathbf{D}_{12}^T]_{j,j} & [\mathbf{C}_{11}]_{j,j} \end{bmatrix} \\
&= \text{Pf}\begin{bmatrix} 0 & -[\mathbf{D}_{12}]_{j,j} \\ [\mathbf{D}_{12}^T]_{j,j} & 0 \end{bmatrix} = [\mathbf{D}_{12}]_{j,j}.
\end{aligned} \qquad \text{(D.28)}
$$

Therefore we have

$$\Delta_2 = [\mathbf{D}_{12}]_{j,j} = \Delta_2 \left[ \frac{\mathbf{x} + 2\mathbf{O}_1 \mathbf{A}_a}{\mathbf{y}} \right]_{j,j}. \tag{D.29}$$

For $\Delta_3$ we consider $I = \{1r, 2r\}$, so that

$$\Delta_3 = \text{Pf}[(\mathbb{S}_{\text{MF}}^{\text{Pf}})_{II}^{-T}] = \text{Pf}[-(\mathbb{S}_{\text{MF}}^{\text{Pf}})_{II}^{-1}] = \text{Pf}\begin{bmatrix} -[\mathbf{C}_{11}]_{j,j} & -[\mathbf{C}_{12}]_{j,j} \\ [\mathbf{C}_{12}^T]_{j,j} & [\mathbf{C}_{11}]_{j,j} \end{bmatrix}$$

$$= \text{Pf}\begin{bmatrix} 0 & -[\mathbf{C}_{12}]_{j,j} \\ [\mathbf{C}_{12}^T]_{j,j} & 0 \end{bmatrix} = [\mathbf{C}_{12}]_{j,j}. \tag{D.30}$$

One then obtains

$$\Delta_3 = [\mathbf{C}_{12}]_{r,r} = \Delta_3 \left[ \frac{\mathbf{x}}{\mathbf{y}} \right]_{j,j}. \tag{D.31}$$

One can show by inspection that this equation coincides with Eq. D.14.

## D.3 Mean field in dual space

The MF theory of the Sourlas action $\mathcal{L}_{\mathbf{N}}^{(u)}\{\bar{\phi}, \phi\}$ is just like the previous section written in terms of $\mathbf{N}$ given in Eq. 40. Then with a simple re-definition $\Delta_i' \equiv u\Delta_i^\phi$, $i = 1, 2, 3$ ($\Delta_i^\phi$ are defined like the ones in Eq. 77, but for the $\phi$ variables in the dual space), and $\Delta_4' \equiv \frac{1}{u}\left[\Delta_1'^2 - \Delta_2'^2 - \Delta_3'^2\right]$, one can obtain the same MF equations as the Eq. 85 and the self-consistent two-point functions Eq. 91. Based on these definitions, and the MF scenario we obtain

$$M_u^{(2)}(\mathbf{A}) \approx c_l A' \int_{\bar{\phi}\phi} \exp\left\{ \bar{\phi}\mathbb{N}\phi + \sum_j \left[ \Delta_1' \left( \bar{\phi}_j^{(1)}\phi_j^{(1)} + \bar{\phi}_j^{(2)}\phi_j^{(2)} \right) - \Delta_2' \left( \bar{\phi}_j^{(1)}\phi_j^{(2)} - \bar{\phi}_j^{(2)}\phi_j^{(1)} \right) \right. \right.$$

$$\left. \left. - \Delta_3' \left( \phi_j^{(1)}\phi_j^{(2)} - \bar{\phi}_j^{(1)}\bar{\phi}_j^{(2)} \right) \right] \right\}, \tag{D.32}$$

where $A' \equiv \exp\left[-l\Delta_4'\right]$. If we consider a uniform transformation, i.e. $m_i = m$ and $u_i = u$ for all $i$ values, then we adopt a same method as the MF in the direct space for the functions with "prime". Then, defining $\mathbf{N}_{a,s} \equiv \frac{1}{2}\left[\mathbf{N} \pm \mathbf{N}^T\right]$, the self-consistent equations are proven to be

$$\Delta_1' = u \left[ \frac{\mathbf{N}_s + \Delta_1' \mathbf{I}}{\mathbf{x}^{(N)} + 2\Delta_2' \mathbf{N}_a} \right]_{j,j}, \qquad \Delta_2' = u \left[ \frac{-\mathbf{N}_a + \Delta_2' \mathbf{I}}{\mathbf{x}^{(N)} + 2\Delta_2' \mathbf{N}_a} \right]_{j,j}, \qquad \Delta_3' = u \left[ \frac{\Delta_3' \mathbf{I}}{\mathbf{x}^{(N)} + 2\Delta_2' \mathbf{N}_a} \right]_{j,j}, \tag{D.33}$$

where

$$\mathbf{x}^{(N)} \equiv \left(\mathbf{N} + \Delta_1' \mathbf{I}\right)\left(\mathbf{N}^T + \Delta_1' \mathbf{I}\right) - \left(\Delta_2'^2 + \Delta_3'^2\right)\mathbf{I}$$

$$= \mathbf{N}\mathbf{N}^T + 2\Delta_1' \mathbf{N}_s + u\Delta_4'. \tag{D.34}$$

These equations bear resemblance to the Eqs. 91, featuring an additional multiplication factor $u$, and substituting the matrix $\mathbf{A}$ with $\mathbf{N}$. The general expression for the SPPM in the MF approximation is then given by

$$M_{u,\text{MF}}^{(2)}(\mathbf{A}) = c_l e^{-l\Delta_4'} \det \mathbb{S}_{\text{MF}}^u(\mathbf{N}), \tag{D.35}$$

where

$$\mathbb{S}_{\text{MF}}^u(\mathbf{N}) \equiv \begin{pmatrix} \mathbf{N} + \left(\Delta_1' - \Delta_2'\right)\mathbf{I} & \Delta_3' \mathbf{I} \\ -\Delta_3' \mathbf{I} & -\mathbf{N}^T - \left(\Delta_1' + \Delta_2'\right)\mathbf{I} \end{pmatrix}. \tag{D.36}$$

One can simplify the formulae using the Schur's complement method (Sec. D.9), according to which

$$\det \mathbb{S}_{\text{MF}}^u(\mathbf{N}) = \det\left[-\mathbf{x}^{(N)} - 2\Delta_2' \mathbf{N}_a\right], \tag{D.37}$$

which is similar to Eqs. D.10 and 95. It's worth mentioning that when matrix $\mathbf{A}$ is circulant, the same property holds for $\mathbf{N}$. This implies that for even $l$:

$$
\begin{aligned}
\ln M_{u,\text{MF}}^{(2)}(\mathbf{A}) &= \ln \det \mathbb{S}_{\text{MF}}^u + \ln c_l - l\Delta_4' \\
&= \sum_{k=1}^{l} \ln\left[x^{(N)}(q_k) + 2\Delta_2' N_a(q_k)\right] - l\ln\left(u+m^2\right) + \sum_{k=1}^{l} \ln(A_{q_k}' A_{-q_k}') - l\Delta_4' \\
&= \sum_{k=1}^{l} \ln\left[A_{q_k}' A_{-q_k}' x^{(N)}(q_k) + 2\Delta_2' A_{q_k}' A_{-q_k}' N_a(q_k)\right] - l\left[\ln\left(u+m^2\right) + \Delta_4'\right] \\
&= \sum_{k=1}^{l} \ln\left[\left(A_{q_k}'(m+\Delta_1') - 1\right)\left(A_{-q_k}'(m+\Delta_1') - 1\right) - A_{q_k}' A_{-q_k}'(\Delta_2'^2 + \Delta_3'^2)\right. \\
&\qquad\qquad \left. + 2\Delta_2' A_{q_k}' A_{-q_k}' N_a(q_k)\right] - l\left[\ln\left(u+m^2\right) + \Delta_4'\right] \\
&\to \frac{l}{2\pi}\int_0^{2\pi} dq\,\ln\left[\left(A_q'(m+\Delta_1')-1\right)\left(A_{-q}'(m+\Delta_1')-1\right) - A_q' A_{-q}'(\Delta_2'^2 + \Delta_3'^2)\right. \\
&\qquad\qquad \left. + 2\Delta_2' A_q' A_{-q}' N_a(q)\right] - l\left[\ln\left(u+m^2\right) + \Delta_4'\right] + O(1), \quad (\text{D.38})
\end{aligned}
$$

where $A_q' \equiv A_q + \frac{m}{u+m^2}$, $N(q) \equiv m - \frac{1}{A_q'}$ is the Fourier component of $\mathbf{N}$, $N_{s,a} \equiv \frac{1}{2}\left(N(q)\pm N(-q)\right)$, and $x^{(N)}$ is given in Eq. 111. As an example consider the case $\Delta_2' \equiv 0$ and $\Delta_3' \neq 0$, then as a result of the first and the third relations in Eq. D.33 we have

$$
\left[\frac{\mathbf{N}_s}{\mathbf{x}^{(N)}}\right]_{j,j} = 0, \qquad \left[\left(\mathbf{x}^{(N)}\right)^{-1}\right]_{j,j} = \frac{1}{u}, \qquad (\text{D.39})
$$

or equivalently in the Fourier space:

$$
\int_{-\pi}^{\pi} \frac{dq}{2\pi} \frac{N_s(q)}{N_q N_{-q} + 2\Delta_1' N_s(q) + \Delta_1'^2 - \Delta_3'^2} = 0, \quad \int_{-\pi}^{\pi} \frac{dq}{2\pi} \frac{1}{N_q N_{-q} + 2\Delta_1' N_s(q) + \Delta_1'^2 - \Delta_3'^2} = \frac{1}{u}. \tag{D.40}
$$

For antisymmetric $\mathbf{N}$ only the second identity of the above equation holds

$$
\int_{-\pi}^{\pi} \frac{dq}{2\pi} \frac{1}{-N_q^2 + u\Delta'^4} = \frac{1}{u}. \tag{D.41}
$$

For the symmetric $\mathbf{N}$, according the discussions presented in Sec. 6 (and TABLE 2) one expects that if $\Delta_3$ is zero, then the only equation to be satisfied is

$$
\Delta_1' = \int_{-\pi}^{\pi} \frac{dq}{2\pi} \frac{1}{N_q + \Delta_1'}. \tag{D.42}
$$

Using Eq. D.38 we also finds for the symmetric $\mathbf{N}$ $(l \to \infty)$

$$
\ln M_{u,\text{MF}}^{(2)}(\mathbf{A}) = \frac{l}{\pi}\int_0^{2\pi} dq\,\ln\left|\left(A_q'(m+\Delta_1')-1\right)\right| - l\left[\ln\left(u+m^2\right) + \Delta_4'\right] + O(1). \tag{D.43}
$$

# E Free energy of minors of a closed chain: Finite temperature

In the following, we present the details of computing the free energy of principal minors for the Laplacian of a closed chain of size $L$. Recall that the partition function is:

$$
Z(n,L) = \sum_I e^{-nE_I}, \tag{E.1}
$$

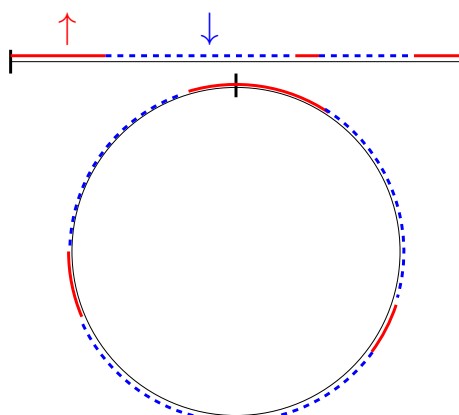

Figure 9: A configuration of clusters formed by the subset of present indices in $I$ for an open and closed chain. The present and absent indices are shown by ↑ and ↓, respectively.

with the energy function $E_I = -\ln \det \mathbf{A}_{DL}(I)$.

More precisely, $E_I = 0$ when index configuration $I$ includes all indices, otherwise,

$$E_I = -\sum_{l=1}^{L-1} m_l^+ \ln(l+1) = E(\{m_l^+\}), \tag{E.2}$$

where $m_l^+$ denotes the number of clusters of size $l$. Similarly, we define $m_l^-$ for the number of clusters of size $l$ formed by the absent (down) indices (see Fig. 9). Thus, we have,

$$L = \sum_l l m_l^+ + \sum_l l m_l^-. \tag{E.3}$$

Let us also define the total number of up and down clusters,

$$m^+ = \sum_l m_l^+, \qquad m^- = \sum_l m_l^-. \tag{E.4}$$

In a closed chain $m^+ = m^- > 0$, except for the single configuration with all indices in $I$. Thus the partition function can be rewritten as

$$Z(n,L) = 1 + \sum_{m=1}^{\infty} \sum_{\{m_l^+, m_l^-\}} \delta_{m^\pm, m} \delta_{\sum_l l(m_l^+ + m_l^-), L} \mathcal{N}_{closed}(\{m_l^+\}, \{m_l^-\}) e^{-nE(\{m_l^+\})}, \tag{E.5}$$

where $\mathcal{N}_{closed}(\{m_l^+\}, \{m_l^-\})$ gives the number of index configurations $I$ with the specified number of up and down clusters.

We first start with an open chain to compute the above entropy function, see Fig. 9. For an open chain either $m^+ = m^- \pm 1$ or $m^+ = m^-$. In the former case, the chain must start with an up cluster if $m^+ = m^- + 1$ or with a down cluster if $m^+ = m^- - 1$. In the latter case, we have the option to start with an up cluster or a down cluster. In each case the number of index configurations is given by the number of cluster permutations $m^+! m^-!$ divided by the symmetry factor $\prod_l (m_l^+!) \prod_l (m_l^-!)$. Therefore, we have,

$$\mathcal{N}_{open}(\{m_l^+\}, \{m_l^-\}) = \frac{m^+! m^-!}{\prod_l (m_l^+!) \prod_l (m_l^-!)} (\delta_{m^+ = m^- \pm 1} + 2\delta_{m^+ = m^-}). \tag{E.6}$$

When the chain is closed $m^+ = m^- \geq 1$, excluding the all up configuration. Consider an imaginary boundary separating indices 1 and $L$. If we fix the state (type, size, and position)

of the cluster at this boundary then the problem reduces to the open chain problem with $m^+ = m^- \pm 1$. Therefore,

$$\mathcal{N}_{closed}(\{m_l^+\}, \{m_l^-\}) = \sum_{l'} l' \left( \frac{m_{l'}^+}{m^+} + \frac{m_{l'}^-}{m^-} \right) \frac{m^+! m^-!}{\prod_l (m_l^+!) \prod_l (m_l^-!)} \,. \tag{E.7}$$

But $m^+ = m^-$ and $\sum_l l m_l^+ + \sum_l l m_l^- = L$, thus we get

$$\mathcal{N}_{closed}(\{m_l^+\}, \{m_l^-\}) = \frac{2L}{m^+ + m^-} \frac{m^+! m^-!}{\prod_l (m_l^+!) \prod_l (m_l^-!)} \,. \tag{E.8}$$

To compute the partition function $Z(n, L)$, we define the following generating function

$$G(n, \mu) = \sum_L e^{-\mu L} Z(n, L), \tag{E.9}$$

with $\mu$ as a chemical potential. Using Eq. E.5 for $Z(n, L)$, the generating function reads as follows:

$$G(n, \mu) = \frac{1}{1 - e^{-\mu}} - \frac{\partial}{\partial \mu} \sum_{m=1}^{\infty} \sum_{\{m_l^+, m_l^-\}} \delta_{m, \sum_l m_l^{\pm}} \frac{e^{-\mu \sum_l (l m_l^+ + l m_l^-)}}{m} \frac{m! m!}{\prod_l (m_l^+!) \prod_l (m_l^-!)} e^{n \sum_l m_l^+ \ln(l+1)} \,. \tag{E.10}$$

Note that here we replace $L e^{-\mu L}$ with $-\frac{\partial}{\partial \mu} e^{-\mu L}$ and then we set $L = \sum_l (l m_l^+ + l m_l^-)$. In the thermodynamic limit we can utilize the following relation

$$\sum_{m^+, m^-} \sum_{\{m_l^+, m_l^-\}} \delta_{m^+, \sum_l m_l^+} \delta_{m^-, \sum_l m_l^-} O(\{m_l^{\pm}\}) = \sum_{\{m_l^+, m_l^-\}} O(\{m_l^{\pm}\}), \tag{E.11}$$

where in the right hand side there is no constraint on the $m_l^{\pm}$. Now, we can do the sum over the $m_l^{\pm}$ to get

$$G(n, \mu) = \frac{1}{1 - e^{-\mu}} - \frac{\partial}{\partial \mu} \sum_{m=1}^{\infty} \frac{1}{m} \left( \sum_l e^{-\mu l} \right)^m \left( \sum_l e^{-\mu l + n \ln(l+1)} \right)^m \,. \tag{E.12}$$

Finally, the sum over $m$ is connected to Taylor expansion of logarithm function, that is,

$$G(n, \mu) = \frac{1}{1 - e^{-\mu}} + \frac{\partial}{\partial \mu} \ln \left( 1 - \frac{e^{-\mu}}{1 - e^{-\mu}} \left( \sum_l e^{-\mu l + n \ln(l+1)} \right) \right) \,. \tag{E.13}$$

## F  Entropy of minors of a closed chain: Zero temperature

In this section, we study the number and structure of the relevant minors of the Laplacian of a closed chain at zero temperature. These are the maximal principal minors, that is index configurations of maximum determinants. Recall that an index configuration is represented by a set of clusters of present indices in that configuration (Fig. 9). The determinant of a principal minor is indeed the product of determinants of these clusters of indices. Thus the energy associated to an index configuration is sum of the energy contributions of such clusters. A cluster of size $l$ has energy

$$E_l = -\log(l+1). \tag{F.1}$$

If there are $m_l^+$ clusters of size $l$ then the total energy is

$$E = -\sum_l m_l^+ \log(l+1). \tag{F.2}$$

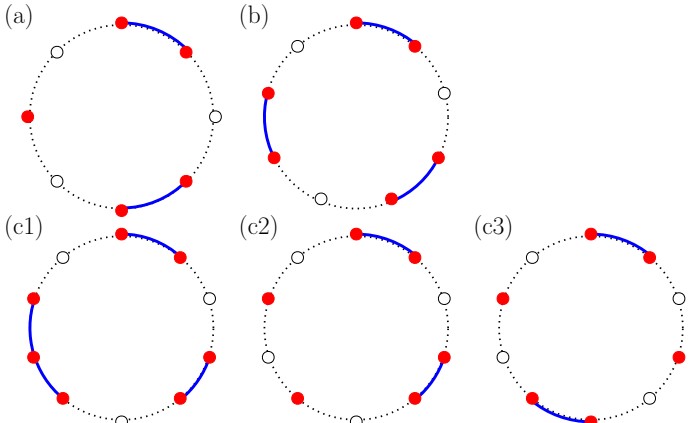

Figure 10: The possible ground states (maximal minors) of Laplacian for a closed chain. The number of ground states depends on the length of the chain. The filled circles are the present elements.

At zero temperature the above energy is minimized by the ground states of the system. Consider index configurations that are a sequence of $L/(l+1)$ clusters of size $l$ separated by single absent indices, e.g., ... $\uparrow\uparrow\uparrow\downarrow\uparrow\uparrow\uparrow\downarrow\uparrow\uparrow\uparrow\downarrow$ .. when $l = 3$. Therefore, the energy density for such configurations is

$$e(l) = -\log(l+1)/(l+1).  \tag{F.3}$$

We see that the above function is minimized for $l = 2$, that is for dimer configurations. Depending on the size of chain we can have different numbers of dimer coverings which minimize the energy (Fig. 10). The three cases that happen are listed below:

- Case $L \equiv 0 \mod 3$:

  Here the chain is covered by a number of dimers; starting from a ground state the others can be obtained from a translation by one lattice constant. So, there are 3 dimer coverings which differ in $2L/3$ sites. Each dimer covering is separated by an extensive Hamming distance from the other two coverings. Thus the complexity $\Sigma = \ln(3)/L$ and the entropy density $s = \Sigma$ approach zero in the thermodynamic limit.

- Case $L \equiv -1 \mod 3$:

  Here the chain is covered by a number of dimers and a cluster of size $l = 1$; starting from a ground state the others are obtained for different positions of the isolated site. So, there are $L$ dimer coverings with extensive Hamming distances. Again, each dimer configuration is isolated with Hamming distances larger than 1 from the other coverings. Thus $\Sigma = \ln(L)/L$ and $s = \Sigma$.

- Case $L \equiv +1 \mod 3$:

  First, there are $L$ clusters each including two configurations with Hamming distance one. In one of these configurations we have a single cluster of size $l = 3$, which is replaced by two clusters of size $l = 1$ in the other configuration. Note that a cluster of size $l = 3$ has exactly the same energy as two clusters of size $l = 1$. i.e. $-\ln(3+1) = -2\ln(1+1)$. The other dimer coverings are isolated configurations. They are obtained by separating the two clusters of size $l = 1$ and placing each of them between two adjacent dimers. More precisely, the number of these configurations is $X = \sum_{i=1}^{L/10}[L\mathbb{I}(5i < L/2) + \frac{L}{2}\mathbb{I}(5i = L/2)]$. The indicator function $\mathbb{I}(C)$ is one if the condition $C$ is satisfied, otherwise it is zero. Therefore, here we have $\Sigma = \ln(L + X)/L$ and $s = \ln(2L + X)/L$.

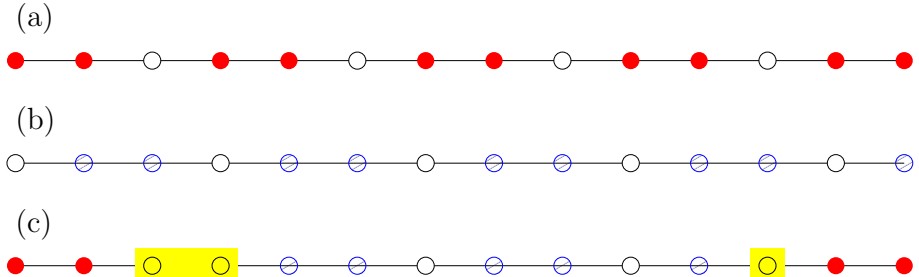

Figure 11: Instability of the ground states of a closed chain at finite temperatures. The free energy barrier to go from the ground state (a) to (b) is small and any finite temperature can destroy such an order. Panels (a) and (b) show a section of two different ground states which are related by a translation. An excited state is obtained in panel (c) by replacing part of state (a) with the same part from state (b). The excitation energy comes from the highlighted domain walls and is independent of the size of the region.

In all cases the entropy density and complexity approach zero in the thermodynamic limit. Moreover, despite the presence of extensively distant ground states, we do not observe a finite-temperature phase transition in this problem. As Fig. 11 schematically displays, this is because the domain walls which separate two different phases of the system have a finite energy cost. Thus, the extensive contribution of their entropy can easily destroy the ordered phases at any finite temperature.

## G  MF results for discrete Laplace matrix

In this section, we provide a detailed exposition of the mean field theory applied to the discrete Laplace scenario. In general, when computing any two-point function associated with a circulant matrix $\mathbf{A}$ of size $L \times L$ within the mean field approximation (refer to Eqs. D.2 and D.7 for the specific case where $\Delta_2 = \Delta_3 = 0$), one must evaluate the following determinants:

$$\det \mathbb{S}_{\mathrm{MF}} = \det(\mathbf{A} + \Delta_1 \mathbf{I}) \det\left(-\mathbf{A}^T - \Delta_1 \mathbf{I}\right), \tag{G.1a}$$

$$\det \mathbb{S}_{\mathrm{MF}(1j)(1j)} = \det(\mathbf{A}_1 + \Delta_1 \mathbf{I}_1) \det\left(-\mathbf{A}^T - \Delta_1 \mathbf{I}\right), \tag{G.1b}$$

$$\det \mathbb{S}_{\mathrm{MF}(1j)(2j)} = \det(\mathbf{A} + \Delta_1 \mathbf{I}) \det\left(-\mathbf{A}_1^T - \Delta_1 \mathbf{I}_1\right), \tag{G.1c}$$

where $\mathbf{A}_1$ and $\mathbf{I}_1$ are $\mathbf{A}$ and $\mathbf{I}$ respectively with first row and column removed. These relations tell us that

$$
\begin{aligned}
2\Delta_1 = \Delta_{11}^{\psi} - \Delta_{22}^{\psi} &= \frac{\det \mathbb{S}_{\mathrm{MF}(1r)(1r)}}{\det \mathbb{S}_{\mathrm{MF}}} - \frac{\det \mathbb{S}_{\mathrm{MF}(2r)(2r)}}{\det \mathbb{S}_{\mathrm{MF}}} \\
&= \frac{\det(\mathbf{A}_1 + \Delta_1 \mathbf{I}_1)}{\det(\mathbf{A} + \Delta_1 \mathbf{I})} - \frac{\det\left(-\mathbf{A}_1^T - \Delta_1 \mathbf{I}_1\right)}{\det\left(-\mathbf{A}^T - \Delta_1 \mathbf{I}\right)} = 2\frac{\det(\mathbf{A}_1 + \Delta_1 \mathbf{I}_1)}{\det(\mathbf{A} + \Delta_1 \mathbf{I})} .
\end{aligned}
\tag{G.2}
$$

Hence, the mean field theory applied to the discrete Laplace problem necessitates calculating the determinants of both $\mathbf{A}_{\mathrm{DL}} + \Delta_1 \mathbf{I}$ and $\mathbf{A}_{\mathrm{DL},1} + \Delta_1 \mathbf{I}_1$, as demonstrated in Equation 164. These

matrices are

$$\mathbf{A}_{\text{DL}} + \Delta_1 \mathbf{I} = \begin{bmatrix} 2+\Delta_1 & -1 & 0 & \dots & 0 & -1 \\ -1 & 2+\Delta_1 & -1 & \dots & 0 & 0 \\ 0 & -1 & 2+\Delta_1 & \dots & 0 & 0 \\ . & . & & \dots & 0 & 0 \\ . & & . & \dots & 2 & -1 \\ -1 & 0 & 0 & \dots & -1 & 2+\Delta_1 \end{bmatrix}_{L \times L}, \tag{G.3}$$

which, after removing the first row and column becomes

$$\mathbf{A}_{\text{DL},1} + \Delta_1 \mathbf{I}_1 = \begin{bmatrix} 2+\Delta_1 & -1 & 0 & \dots & 0 & 0 \\ -1 & 2+\Delta_1 & -1 & \dots & 0 & 0 \\ 0 & -1 & 2+\Delta_1 & \dots & 0 & 0 \\ . & . & & \dots & 0 & 0 \\ . & & . & \dots & 2+\Delta_1 & -1 \\ 0 & 0 & 0 & \dots & -1 & 2+\Delta_1 \end{bmatrix}_{(L-1)\times(L-1)}. \tag{G.4}$$

Before calculating these determinants, let us start with calculating the determinant of $\mathbf{A}_{\text{DL}}$ and $\mathbf{A}_{\text{DL},1}$ as a guidance of the other determinants. It is easily obtained using the properties of circulant matrices (see the previous section), giving rise to $\det \mathbf{A}_{\text{DL}} = 4^L \prod_{k=1}^{L} \sin^2 \frac{\pi k}{L}$. The determinant of $\mathbf{A}_{\text{DL},1}$ is simply calculated using a recursion relation. Denoting $\det \mathbf{A}_{\text{DL},1}$ by $D_0^{(L)}$, one easily shows by inspection that

$$D_0^{(L)} = 2D_0^{(L-1)} - D_0^{(L-2)}, \tag{G.5}$$

where $D_0^{(L-1)}$ ($D_0^{(L-2)}$) is the determinant of the $\mathbf{A}_{\text{DL},1}$ with first (first and second) row(s) and column(s) removed. Note that $D_0^{(1)} = 2$ and $D_0^{(2)} = 3$. One can easily solve this recursion relation by trying $D_0^{(L)} = a_{L-1}2^{L-1} + a_{L-2}2^{L-2} + \dots + 2a_1 + a_0$ where $a_i$'s should be determined, given that $a_0 = 2$ and $a_1 = \frac{1}{2}$. One shows that the solution corresponds to the relation $a_L = \frac{1}{2}a_{L-1}$, giving rise to $D_0^{(L)} = L + 1$.

Back to the problem of interest, i.e. the determinants in Eq. G.2, and considering the case $\mathbf{A} = \mathbf{A}_{\text{DL}}$ one easily finds that

$$\det(\mathbf{A}_{\text{DL}} + \Delta_1 \mathbf{I}) = \prod_{k=1}^{L} \left( 4\sin^2 \frac{\pi k}{L} + \Delta_1 \right). \tag{G.6}$$

Using the Euler-Maclaurin formula we find the following relation in the thermodynamic limit

$$\begin{aligned}
\lim_{L\to\infty} \ln \prod_{k=1}^{L} \left( 4\sin^2 \frac{\pi k}{L} + \Delta_1 \right) &= \lim_{L\to\infty} \sum_{k=1}^{L} \ln \left( 4\sin^2 \frac{\pi k}{L} + \Delta_1 \right) \\
&\to \int_0^L dk \ln \left( 4\sin^2 \frac{\pi k}{L} + \Delta_1 \right) + \mathcal{O}(1) \\
&= \frac{L}{\pi} \int_0^{\pi} dx \ln \left( 4\sin^2 x + \Delta_1 \right) + \mathcal{O}(1) \\
&= L \ln \frac{\Delta_1 + 2 + \sqrt{\Delta_1(\Delta_1 + 4)}}{2} + \mathcal{O}(1).
\end{aligned} \tag{G.7}$$

For $D^{(L)} \equiv \det(\mathbf{A}_{\text{DL},1} + \Delta_1 \mathbf{I}_1)$, similar to the Eq. G.5 we have

$$D^{(L)} = \zeta D^{(L-1)} - D^{(L-2)}, \tag{G.8}$$

where $\zeta \equiv 2 + \Delta_1$. To find a closed form for G.8, we note that:

$$
\begin{aligned}
D^{(1)} &= \zeta\,, \\
D^{(2)} &= \zeta^2 - 1\,, \\
D^{(3)} &= \zeta^3 - 2\zeta\,, \\
D^{(4)} &= \zeta^4 - 3\zeta^2 + 1\,, \\
D^{(5)} &= \zeta^5 - 4\zeta^3 + 3\zeta\,, \\
D^{(6)} &= \zeta^6 - 5\zeta^4 + 6\zeta^2 - 1\,, \\
D^{(7)} &= \zeta^7 - 6\zeta^5 + 10\zeta^3 - 4\zeta\,, \\
D^{(8)} &= \zeta^8 - 7\zeta^6 + 15\zeta^4 - 10\zeta^2 + 1\,, \\
D^{(9)} &= \zeta^9 - 8\zeta^7 + 21\zeta^5 - 20\zeta^3 + 5\zeta\,, \dots\,,
\end{aligned}
\tag{G.9}
$$

and in general

$$
\begin{aligned}
D^{(L)} &= \zeta^L - \binom{L-1}{L-2}\zeta^{L-2} + \binom{L-2}{L-4}\zeta^{L-4} - \binom{L-3}{L-6}\zeta^{L-6} + \binom{L-4}{L-8}\zeta^{L-8} + \dots \\
&= \sum_{m=0}^{\mathrm{int}\left[\frac{L}{2}\right]} (-1)^m \binom{L-m}{L-2m} \zeta^{L-2m} \\
&= \zeta^L {}_2F_1\left[-\frac{L-1}{2}, -\frac{L}{2}, -L, \frac{4}{\zeta^2}\right].
\end{aligned}
\tag{G.10}
$$

Note also that we can find the leading term in the thermodynamic limit ($L \to \infty$) using the Eq. G.8:

$$
\frac{D^{(L)}}{D^{(L-1)}} = (2 + \Delta_1) - \frac{1}{\frac{D^{(L-1)}}{D^{(L-2)}}}\,.
\tag{G.11}
$$

In the thermodynamic limit we have $x \equiv \frac{D^{(L)}}{D^{(L-1)}} \simeq \frac{D^{(L-1)}}{D^{(L-2)}}$, so that

$$
x = 2 + \Delta_1 - \frac{1}{x} \quad \rightarrow \quad x = \frac{\zeta}{2} + \frac{1}{2}\sqrt{\zeta^2 - 4}.
\tag{G.12}
$$

Now, by utilizing this thermodynamic solution, it becomes straightforward to examine the following trial form as the leading term in the thermodynamic limit (note that the size of $\mathbf{A}_{\mathrm{DL},1}$ is $(L-1) \times (L-1)$)

$$
D^{(L)} = \alpha(\zeta) \times x^{L-1}\,,
\tag{G.13}
$$

$\alpha(\zeta)$ being a proportionality function independent of $L$. Therefore, we have

$$
D^{(L\to\infty)}_{\text{leading term}} = \alpha(\zeta)\left(\frac{\zeta}{2} + \frac{1}{2}\sqrt{\zeta^2 - 4}\right)^{L-1}.
\tag{G.14}
$$

Comparing this with Eq. G.10 we find that $\lim_{L\to\infty} \alpha(\zeta) = 1$, and also

$$
\begin{aligned}
\alpha(\zeta) &= \lim_{L\to\infty} \left(\frac{2}{\zeta + \sqrt{\zeta^2 - 4}}\right)^{L-1} \zeta^L {}_2F_1\left[-\frac{L-1}{2}, -\frac{L}{2}, -L, \frac{4}{\zeta^2}\right] \\
&\simeq 1 + c_0 \frac{e^{-\gamma_1 \zeta}}{(\zeta - 2)^{\gamma_2}}\,,
\end{aligned}
\tag{G.15}
$$

where the second line is obtained by fitting the function in the interval $]2,3]$, and $c_0 \simeq \frac{4}{5}$, $\gamma_1 \simeq \frac{1}{2}$, and $\gamma_2 \simeq 0.6$. All in all we have

$$D^{(L \to \infty)}_{\text{leading term}} \simeq \left(1 + c_0 \frac{e^{-\gamma_1 \zeta}}{(\zeta-2)^{\gamma_2}}\right) \left(\frac{\zeta + \sqrt{\zeta^2 - 4}}{2}\right)^{L-1}. \qquad (G.16)$$

Therefore for large enough $L$ values, Eq. 164 gives us

$$\zeta - 2 = \alpha(\zeta) \left(\frac{\zeta + \sqrt{\zeta^2 - 4}}{2}\right)^{L-1} \left[\prod_{k=1}^{L}\left(4\sin^2 \frac{\pi k}{L} + \zeta - 2\right)\right]^{-1}, \qquad (G.17)$$

which, transforms to the following form using the Eq. G.7

$$\zeta - 2 = \alpha(\zeta) \frac{\left(\frac{\zeta + \sqrt{\zeta^2 - 4}}{2}\right)^{L-1}}{\left(\frac{\zeta + \sqrt{\zeta^2 - 4}}{2}\right)^{L}}, \quad \text{so that} \quad \alpha(\zeta) = \frac{1}{2}(\zeta - 2)\left(\zeta + \sqrt{\zeta^2 - 4}\right). \qquad (G.18)$$

# H  Stability of the MF solution in the dual space

In the dual space, the MF equations are similar to the ordinary one, except we need to interchange $\Delta$ functions by $\Delta' = u\Delta$, and $\mathbf{A}$ by $\mathbf{N}$ given in Eq. 40. Here we adopt the notation of section 6.4, i.e. we define $\Delta_j^{(1)'} \equiv \Delta_j^{(\bar{1}1)'} = \Delta_j^{(\bar{2}2)'}$ and $\Delta_j^{(3)'} \equiv \Delta_j^{(\bar{1}\bar{2})'} = \Delta_j^{(12)'}$, $y_j' \equiv \left(y_j^{(1)'}, y_j^{(3)'}\right)$, where $y_j^{(1)'} \equiv \Delta_j^{(1)'}$ and $y_j^{(3)'} \equiv i\Delta_j^{(3)'}$. $\mathcal{B} = \mathcal{B}_u \equiv \frac{2}{u}\mathbb{I}_{2l}$, where $u$ is the Sourlas transformation parameter. We set $\Delta_j^{(2)'}$ to zero like section 6.4. Then we use the identity

$$\frac{1}{2}J_\phi^T \mathcal{B}_u^{-1} J_\phi = u \sum_j \bar{\phi}_j^{(1)} \phi_j^{(1)} \bar{\phi}_j^{(2)} \phi_j^{(2)}, \qquad (H.1)$$

as an alternative expression for the four-interaction part of the effective action in the dual space. In this equation $(\bar{\phi}_j^{(r)}, \phi_j^{(r)})$ is a Grassmann couple, and $J_\phi \equiv (J_1^\phi, ..., J_l^\phi)$, $J_j^\phi \equiv (J_j^{(1)}, J_j^{(2)}, J_j^{(3)})$ given in Eq. 113 but this time for $\bar{\phi}$ and $\phi$, so that $\left\langle J_j^{(k)}\right\rangle = 2y_j^{(k)'}$. Using the identity Eq. 116 we find

$$M^{(2)}(\mathbf{A}) = c_l \int_{\phi\bar{\phi}} \exp\left[\bar{\phi}\mathbb{N}\phi + u\bar{\phi}_r^{(1)}\phi_r^{(1)}\bar{\phi}_r^{(2)}\phi_r^{(2)}\right] = [\det \mathcal{B}_u]^{\frac{1}{2}} \int D\{y\} e^{-\mathcal{F}_u^{(2)}(\mathbf{A}, y')}, \qquad (H.2)$$

where

$$\mathcal{F}_u^{(2)}(\mathbf{A}, y') \equiv -\ln\left[\mathcal{M}_u^{(2)}(\mathbf{A}, y')\right] \equiv -\ln\left[c_l \int_{\bar{\phi}\phi} e^{\mathfrak{L}_u(\bar{\phi}, \phi, y', J_\phi)}\right], \qquad (H.3)$$

and

$$\mathfrak{L}_u(\bar{\phi}, \phi, y', J_\phi) \equiv \bar{\phi}\mathbb{N}\phi - \frac{1}{2}y'^T \mathcal{B}_u y' + y'^T J_\phi, \qquad (H.4)$$

is a variant of $\mathfrak{L}$ defined in Eq. 114. We derive the dual MF equations by minimizing $\mathcal{F}_u^{(2)}(\mathbf{A}, y')$ with respect to all $y_i'$'s as follows:

$$\left.\partial_{y_j^{(k)'}} \mathcal{F}_u^{(2)}(\mathbf{A}, y')\right|_{y'=\bar{y}'} = 0, \qquad (H.5)$$

or equivalently

$$\partial_{y_j^{(k)'}} \ln\left[e^{\frac{1}{2}y'^T \mathcal{B}_u y'} \mathcal{M}_u^{(2)}(\mathbf{A}, y')\right] = \sum_{k'} \mathcal{B}_{u,kk'} \bar{y}'_j^{(k')}. \tag{H.6}$$

Analogous to the direct space, we define $F_{\mathrm{MF},u}^{(2)}(\mathbf{A}) \equiv \mathcal{F}_u^{(2)}(\mathbf{A}, \bar{y}')$. Expanding $\mathcal{F}_u^{(2)}(\mathbf{A}, y)$ around $\bar{y}'$ to the second order, and doing the Gaussian integral we find that

$$M_{\mathrm{SP}}^{(2)}(\mathbf{A}) \approx \frac{e^{-F_{\mathrm{MF},u}^{(2)}(\mathbf{A})}}{\sqrt{\det \mathcal{C}_u}}, \tag{H.7}$$

where $\mathcal{C}_u \equiv \mathcal{B}_u^{-1} \beta_u$, and $\beta_{u,kk'}^{(j)} \equiv \partial_{y_j'^{(k)}} \partial_{y_j'^{(k')}} \mathcal{F}_u^{(2)}(\mathbf{A}, y')|_{y'=\bar{y}'}$. Now, representing the elements of $\mathcal{C}_u$ by $c_{u,ij}^r$, one finds

$$c_{u,kk'}^{(j)} = \delta_{kk'} - h_{u,kk'}^{(j)}, \tag{H.8}$$

where

$$h_{u,kk'}^{(j)} = \frac{u}{2}\left[\left\langle J_j^{(k)} J_j^{(k')}\right\rangle_u - \left\langle J_j^{(k)}\right\rangle_u \left\langle J_j^{(k')}\right\rangle_u\right], \tag{H.9}$$

is the correlation function of the currents in the dual space. In this equation $\langle\rangle_u$ shows an expectation value with respect to the dual effective action $\mathfrak{L}_u$. Repeating the same calculations as Sec. 6.4, we find that

$$-h_{u,11}^{(j)} = h_{u,22}^{(j)} = \frac{1}{u}\left[\left(\Delta_j^{(1)'}\right)^2 + \left(\Delta_j^{(3)'}\right)^2\right], \qquad h_{u,12}^r = h_{u,21}^r = \frac{2i}{u}\Delta_j^{(1)'}\Delta_j^{(3)'}. \tag{H.10}$$

Here $\Delta_j^{(1)'} = u\left\langle \bar{\phi}_j^{(1)} \phi_j^{(1)}\right\rangle$ and $\Delta_j^{(3)'} = u\left\langle \phi_j^{(1)} \phi_j^{(2)}\right\rangle$ are the MF solutions of Eq. H.6. For the case the $\Delta_j^{(k)'}$ is independent of $j$, i.e. $\Delta_j^{(k)'} \equiv \Delta_k'$ $k = 1, 3$, we find that

$$M_{\mathrm{SP}}^{(2)}(\mathbf{A}) \approx \frac{e^{-F_{\mathrm{MF},u}^{(2)}(\mathbf{A})}}{(1 - \frac{1}{u^2}(\Delta_1'^2 - \Delta_3'^2)^2)^{\frac{l}{2}}}. \tag{H.11}$$

This equation can be written in terms of free energy density $f_{\mathrm{SP}}^u \equiv F_{\mathrm{SP}}^u/l$ as follows:

$$f_{\mathrm{SP}}^u / f_{\mathrm{MF}}^u = 1 + \frac{1}{2f_{\mathrm{MF}}^u}\ln\left[1 - \frac{1}{u^2}(\Delta_1'^2 - \Delta_3'^2)^2\right]. \tag{H.12}$$

# I  Jensen's inequality for Berezin integrals

A convex function $(\phi(x))$ is defined as a function that satisfies the following identity

$$\phi(t x_1 + (1-t)x_0) \leq t\phi(x_1) + (1-t)\phi(x_0), \tag{I.1}$$

where $0 \leq t \leq 1$ is the interpolation (real) parameter between $x_0$ and $x_1$. The above relation guarantees the following consequences

$$\phi(x_1) \geq \phi(x_0) + \frac{d\phi}{dt}\Big|_{t=0}, \qquad \phi(x_0) \geq \phi(x_1) - \frac{d\phi}{dt}\Big|_{t=1}. \tag{I.2}$$

One can also show that for a convex function

$$\phi\left(\int g(x)f(x)dx\right) \leq \int \phi(g(x))f(x)dx, \tag{I.3}$$

where $f(x) \geq 0$ is a weight function. An important example is $\phi(X) = \exp X$, resulting to

$$\left\langle e^{g(x)} \right\rangle \geq e^{\langle g(X) \rangle}, \tag{I.4}$$

where $\langle ... \rangle \equiv \int ... f(x)\mathrm{d}x$. This identity is valid in any dimension for the real integrals. For our case where we deal with the Berezin integrals we use the following construction: We define

$$M_0 \equiv M_{\mathrm{MF}}^{(2)}(\mathbf{A}) = \int_{\bar{\chi}\chi} e^{\mathcal{S}_0\{\bar{\chi},\chi\}}, \qquad M_1 \equiv M^{(2)}(\mathbf{A}) = \int_{\bar{\chi}\chi} e^{\mathcal{S}_{\mathrm{A}}^{(2)}\{\bar{\chi},\chi\}}, \tag{I.5}$$

where $\mathcal{S}_0\{\bar{\chi},\chi\} \equiv S_{\mathrm{MF}}\{\bar{\chi},\chi\} - \sum_r \Delta_4(r)$. We also define an interpolation between these two actions as follows

$$\mathcal{S}_t\{\bar{\chi},\chi\} \equiv t\mathcal{S}_{\mathrm{A}}^{(2)}\{\bar{\chi},\chi\} + (1-t)\mathcal{S}_0\{\bar{\chi},\chi\}, \tag{I.6}$$

and correspondingly

$$M_t \equiv \int_{\bar{\chi},\chi} e^{\mathcal{S}_t\{\bar{\chi},\chi\}}, \qquad \langle ... \rangle_t \equiv \frac{\int_{\bar{\chi},\chi} ... e^{\mathcal{S}_t\{\bar{\chi},\chi\}}}{M_t}. \tag{I.7}$$

Then one checks

$$\begin{aligned}
\frac{\mathrm{d}\log M_t}{\mathrm{d}t} &= \left\langle \mathcal{S}_{\mathrm{A}}^{(2)}\{\bar{\chi},\chi\} - \mathcal{S}_0\{\bar{\chi},\chi\} \right\rangle_t, \\
\frac{\mathrm{d}^2\log M_t}{\mathrm{d}t^2} &= \left\langle \left(\mathcal{S}_{\mathrm{A}}^{(2)}\{\bar{\chi},\chi\} - \mathcal{S}_0\{\bar{\chi},\chi\}\right)^2 \right\rangle_t - \left\langle \mathcal{S}_{\mathrm{A}}^{(2)}\{\bar{\chi},\chi\} - \mathcal{S}_0\{\bar{\chi},\chi\} \right\rangle_t^2 \geq 0,
\end{aligned} \tag{I.8}$$

which tells us that $\log M_t$ is a convex function. Then Eq. I.2 gives

$$\log M_1 \geq \log M_0 + \left. \frac{\mathrm{d}\log M_t}{\mathrm{d}t} \right|_{t=0}, \qquad M_1 \geq M_0 e^{\left\langle \mathcal{S}_{\mathrm{A}}^{(2)}\{\bar{\chi},\chi\} - \mathcal{S}_0\{\bar{\chi},\chi\} \right\rangle_0}. \tag{I.9}$$

This, when combined with Eq. 140 leads to Eq. 141.

## J Analysis of $\mathcal{M}$ and $\mathcal{F}$ for the TFI chain

In this section, we examine $\mathcal{F}$ and $\mathcal{M}$ as defined in Eq.119 both in the direct space and in the dual space (Eq. H.3). By conducting this analysis, insights can be gained into the reliability of the mean-field (MF) solutions with respect to external parameters and the $u$ parameter in the dual space. Our focus is on the Ising model, which exhibits a complex structure. Throughout this section, we assume $\Delta_2 = 0$, and $\Delta_1$ and $\Delta_3$ are real. In this case the $\Delta$ space is pretty large, i.e. $2l$-dimensional, see Eq. 119 where $\Delta$ functions have space indices. To make the space smaller, we consider the subspace corresponding to $\Delta_i^r = \Delta_i^{r'}$ for all $r, r' \in [1, l]$ and $i = 1, 3$. This subspace corresponds to the restriction implied by the periodicity of the system, making $\Delta$ functions independent of space. This gives a two-dimensional landscape which is considered in the following.

Figure 12 illustrates $\mathcal{M}_u^{(2)}(\Delta_1, \Delta_3)$ as a function of $h$. Since $\mathbf{F}$ is an antisymmetric matrix for the Ising model, one anticipates that the mean-field (MF) solutions are solely dependent on $\Delta_4$ as defined in Eq. 88 (refer to the reasoning leading to Eq. 102). For instance, in the case of $h = 0$, two hyperbolas emerge where $\Delta_1 = \pm\sqrt{\Delta_4 - \Delta_3^2}$ and $\mathcal{M}_u^{(2)}$ reaches its maximum. As $h$ increases, these hyperbolas converge due to the decreasing nature of $\Delta_4$ (the gap between them), ultimately disappearing at $h = 1$ in the thermodynamic limit, as observed in the $h = 1$ case in Fig. 102.

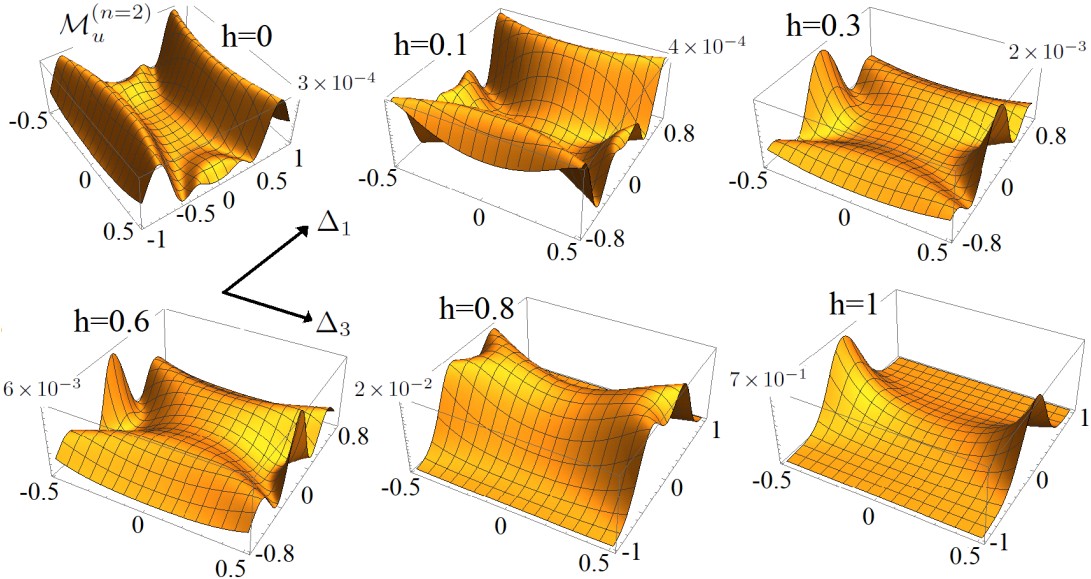

Figure 12: The SP integrand in the direct space $\mathcal{M}^{(2)}_{\mathrm{MF}}$ given in Eq. 119 for the Ising model $L = 10$, in terms of $\Delta_1$ and $\Delta_3$ for various $h$ values. The stationary points for a continuous line which is locus of stationary points with zero slope. It forms two hyperbolas according to the relation $\Delta_1^2 - \Delta_3^2 = \Delta_4$, where $\Delta_4$ is the solution of mean field consistency relations.

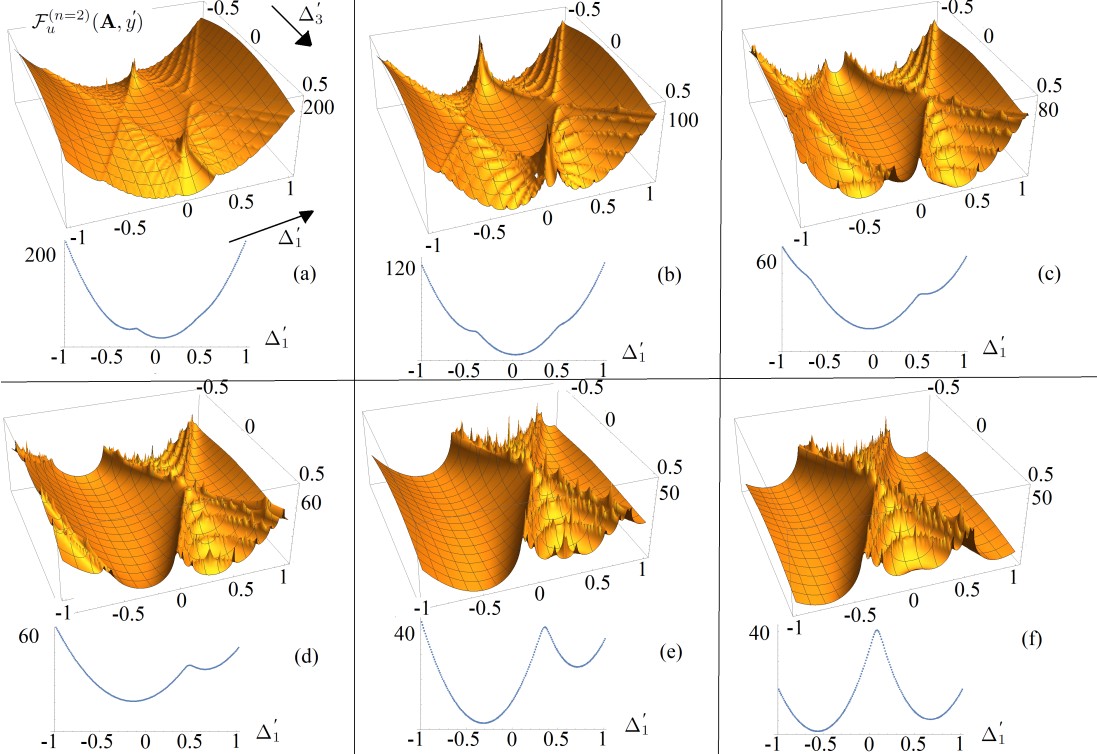

Figure 13: $\mathcal{F}^{(2)}_u$ in the dual space given in Eq. H.3 for the Ising model with $h = 0$, $L = 10$, and $m = -\sqrt{\sqrt{u} - u}$ in terms of $\Delta_1$ and $\Delta_3$ for various rates of $u$: (a) $u = 0.1$, (b) $u = 0.2$, (c) $u = 0.35$, (d) $u = 0.5$, (e) $u = 0.75$, and (f) $u = 0.99$. We see that for $0.5 \lesssim u$ a second peak appears which makes the MF approximation questionable.

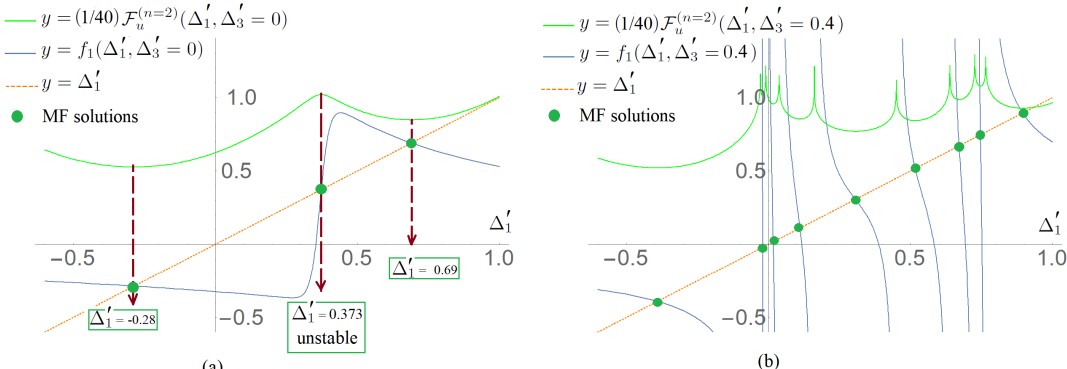

Figure 14: $\mathcal{F}_u^{(2)}(\Delta_1, \Delta_3)$ (Eq. H.3) for the Ising model with $h = 0$, $u = 0.7$, $L = 30$, and $m = -\sqrt{\sqrt{u} - u}$ in terms of $\Delta_1'$ for (a) $\Delta_3' = 0$ and (b) $\Delta_3' = 0.4$. In each figure $y' = \Delta_1'$ and $y = f_1(\Delta_1', \Delta_3')$ (defined as the right-hand-side of the first relation of Eq. 107) have also been shown, the crossing point of which is the MF solutions. We see that the crossing points are actually the extremum (stationary) points of the free energy $\mathcal{F}_u^{(2)}$. For (a), $\Delta_1' = 0.373$ is an unstable solution in the sense that it represents a local maximum for $\mathcal{F}_u^{(2)}$, which is unstable towards the two other solutions.

To decipher the solution structure, let's examine $\Delta_3 = 0$, where two peaks for $\mathcal{M}_u^{(2)}$ emerge, representing the primary contribution of the integrand 118: $\Delta_1^{\text{peak}} = \pm\Delta_4$. These peaks merge as $h \to 1$. In the other hand, as shown in Fig. 6d, the MF results become highly consistent with the exact numerical results as $h \to 1$. This leads us to the conclusion that when $\mathcal{M}_u^{(2)}$ exhibits a single peak, the MF results become more reliable.

To evaluate the universality of this finding, we undertake a parallel analysis in the dual space. In Fig.13, we present $\mathcal{F}_u^{(2)}$ for the Ising model with $L = 10$, $h = 0$, and various values of $u$. Notably, there are distinct paths where $\mathcal{M}$ becomes zero, leading to the divergence of $\mathcal{F}_u^{(2)}$ and the emergence of a complex energy landscape with multiple minima, representing mean-field (MF) solutions. To elucidate the solution structure, we plot $\mathcal{F}_u^{(2)}(\Delta_1', \Delta_3' = 0)$ in terms of $\Delta_1$ beneath each graph. For $\Delta_3' = 0$, the configuration of minima undergoes changes with varying $u$. As $u$ increases, a second peak emerges, gaining strength with continued increases in $u$. Around $0.5 \lesssim u$, the second peak becomes substantial, coinciding with a decrease in the validity of the MF results (refer to Fig.7, where it is evident that the MF results in the dual space are valid and stable for $u \lesssim 0.5$). Once again, when the second peak becomes comparable to the first, the reliability of the MF solutions diminishes. This is the reason why the direct MF results are better for higher $h$ values.

The structure of the MF solutions for a fixed $\Delta_3'$ is depicted in Fig.14, revealing a multitude of solutions for $\Delta_3' = 0.4$. Here, the minimum of $\mathcal{F}_u^{(2)}$ coincides with the MF solutions as per Eq.107, specifically, $\Delta_1' = f_1(\Delta_1', \Delta_3')$, where $f_1(\Delta_1', \Delta_3')$ corresponds to the right-hand side of the first relation in this equation.

In addition to the criteria outlined above, the most robust criterion we have identified for assessing the reliability of MF results is the stability analysis, which was outlined and employed in Sections 7.2 and 8. In the subsequent discussion, we provide results pertaining to the stability of MF results for the Ising model in the dual space. The Fig. 15 shows the slope (Fig. 15a) and the width (Fig. 15b) of $R_2$ for $h = 0$. This figure displays both the direct and dual MF results, along with the EN results. The corresponding stability tests are also presented in Fig. 15d for various magnetic fields. A good consistency is observed between the "best $u$ intervals" (based on the fitness to the EN results) and the most "stable" solutions (identified by a shaded area in blue using the criterion $f_{\text{SP}}^u / f_{\text{MF}}^u > 0.99$). Fig. 15c shows the best $u$ values that we used in Fig. 6 in terms of $h$.

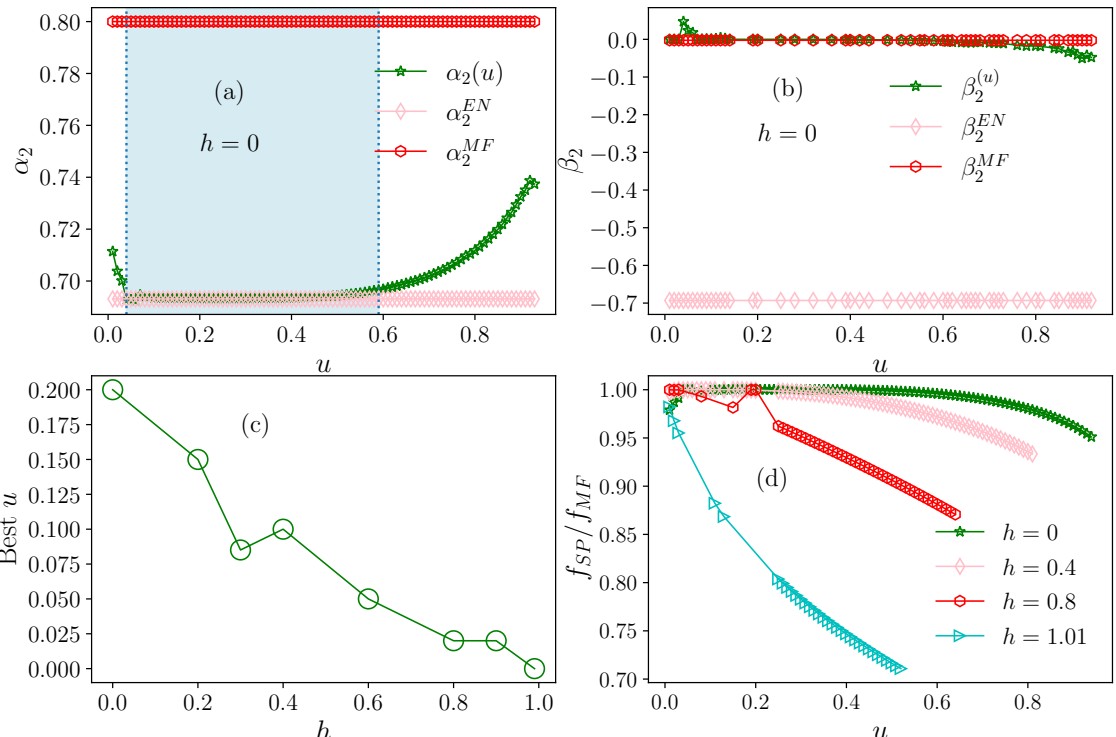

Figure 15: MF approximation of (a) $\alpha_2$ and (b) $\beta_2$ for the Ising model for $h = 0$ in terms of $u$ (compared with the exact result). The blue area in (a) shows the region with $f_{\text{SP}}/f_{\text{MF}} > 0.99$. (c) The best $u$ value that was used in Fig. 6d. Note that for $h > 1$ the best choice is the direct MF, which does not correspond to any $u$. (d) shows $f_{\text{SP}}/f_{\text{MF}}$ for various $h$ values representing the stability of the MF solution.

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
