# Peer review of "A field theory representation of sum of powers of principal minors and physical applications"

_SciPost Physics Core, doi:SciPost Phys. Core 8, 051 (2025)_

## Round 1 · Referee Report · Anonymous (Referee 1) · 2025-5-14

Strengths

1- Thorough theoretical examination of sums of powers of principal minors, with motivations in quantum lattice models 2- Mean field approximation gives good results

Weaknesses

1- No exact results except for the discrete Laplacian

Report

In this manuscript, the authors investigate the problem of computing the sum of powers of principal minors (SPPM) of a given matrix. The main motivation for studying this comes from the evaluation of Shannon-Renyi entropies in free fermionic quantum Hamiltonians, and related problems.

The main achievement is a Grassman integral representation for the SPPM, to which a mean field treatment can be applied, leading to reasonable approximations for large matrix sizes (long chains in the quantum context). This approximation correctly predicts the well known quantum phase transition occurring in the transverse field Ising model.

Overall the paper is quite technical and long, with also many appendices. Most computations are clearly presented. The results obtained by the authors also represent a small but worthy progress on an otherwise difficult problem. For this reason I recommend publication in Scipost Physics, provided the following questions are addressed.

Requested changes

1- It might be worth stating right after (1) that the SPPM for n=1 is simply related to the characteristic polynomial.

2- Page 3, second paragraph: 'We have managed to calculate this quantity analytically'. Technically this holds only at mean field level, so it is not an analytical formula for the entropy.

3- Page 4, top left. The sentence ending with 'those sites that there is a fermion' reads awkwardly.

4- Page 5, top left: 'DPP can be defined easily using the concept of formation probabilites'. For quadratic fermionic Hamiltonians.

5- Page 6, left 'that we also classify them' remove them.

6- The notation in equation (22) is not consistent for n=1 with that in (21), which is confusing.

7- I find the discussion in section VII. A to be a bit confusing. Where exactly is the analytical solution for large $L$ SPPM?

8- In particle conserving quadratic fermionic Hamiltonians, the ground state has a fixed number of particles. In the formalism of the present paper, this would mean summing over all minors of a given size. Could the author comment on this case.

Recommendation

Ask for minor revision

---

## Round 2 · Author Response

We thank the referee for his/her constructive comments, and detailed analysis of the manuscript, which encouraged us to revise the paper. We specifically thank the referee for stating that “I recommend publication in Scipost Physics” In the following we have addressed the questions raised by the referee and highlighted the answers and the changes in the manuscript in blue.

Comments of the reviewer:

[1] It might be worth stating right after (1) that the SPPM for n=1 is simply related to the characteristic polynomial.
We thank the reviewer for comments. Although it has been mentioned in Eq. 22c, we state it also right after Eq. 1.

[2] Page 3, second paragraph: 'We have managed to calculate this quantity analytically'. Technically this holds only at mean field level, so it is not an analytical formula for the entropy.
We changed it to “… we have managed to calculate this quantity analytically at the mean field level for the first time”.

[3] Page 4, top left. The sentence ending with 'those sites that there is a fermion' reads awkwardly.
We corrected this sentence.

[4] Page 5, top left: 'DPP can be defined easily using the concept of formation probabilites'. For quadratic fermionic Hamiltonians.
Corrected.

[5] Page 6, left 'that we also classify them' remove them.
Corrected.

[6] The notation in equation (22) is not consistent for n=1 with that in (21), which is confusing.
The notation is correct; please note that in Eq. (22) we write M^(n=1)(A) for a given matrix A. In Eq. (21) we have M^(n=2)(B) for a matrix B which comes from the Hubbard problem.

[7] I find the discussion in section VII. A to be a bit confusing. Where exactly is the analytical solution for large L SPPM?
The exact analytic expression for the partition function G(n,μ) (in the thermodynamic limit) is provided in Eq. (154). Its generalization G(n,μ,λl) with Lagrange multipliers λl is given in Eq. (161). The partition function is enough to obtain for instance the average energy and entropy in Eqs. (156) and (157).
A sentence is added to the beginning of this sub-section.

[8] In particle conserving quadratic fermionic Hamiltonians, the ground state has a fixed number of particles. In the formalism of the present paper, this would mean summing over all minors of a given size. Could the author comment on this case.
In the conclusion, we incorporated an additional sentence and reference, explicitly noting both the significance of this problem and our ongoing efforts to resolve the computational methodology.

---

## Round 2 · List of Changes

1- We processed all the comments given by the referee, highlighted in blue. 2- In the SEC. VIIA we added a description for the analytical results, as requested by the referee.

---

## Editorial Decision

published